# On Linear Mode Connectivity
# of Mixture-of-Experts Architectures

**Viet-Hoang Tran**[*][†]
Department of Mathematics
National University of Singapore
`hoang.tranviet@u.nus.edu`

**Van-Hoan Trinh**[*]
Department of Mathematics
Technical University of Munich
`vanhoan.trinh@tum.de`

**Khanh Vinh Bui**[*]
Independent Researcher
Ho Chi Minh City, Vietnam
`khanhvinhbui0512@gmail.com`

**Tan M. Nguyen**
Department of Mathematics
National University of Singapore
`tanmn@nus.edu.sg`

## Abstract

Linear Mode Connectivity (LMC) is a notable phenomenon in the loss landscapes of neural networks, wherein independently trained models have been observed to be connected–up to permutation symmetries–by linear paths in parameter space along which the loss remains consistently low. This observation challenges classical views of non-convex optimization and has implications for model ensembling, generalization, and our understanding of neural loss geometry. Inspired by recent studies on LMC in standard neural networks, we systematically investigate this phenomenon within Mixture-of-Experts (MoE) architectures–a class of models known for their scalability and computational efficiency, which combine traditional neural networks–referred to as experts–through a learnable gating mechanism. We begin by conducting a comprehensive analysis of both dense and sparse gating regimes, demonstrating that the symmetries inherent to MoE architectures are fully characterized by permutations acting on both the expert components and the gating function. Building on these foundational findings, we propose a matching algorithm that enables alignment between independently trained MoEs, thereby facilitating the discovery of LMC. Finally, we empirically validate the presence of LMC using our proposed algorithm across diverse MoE configurations–including dense, sparse, and shared-expert variants–under a wide range of model settings and datasets of varying scales and modalities. Our results confirm the existence of LMC in MoE architectures and offer fundamental insights into the functional landscape and optimization dynamics of deep learning models. The code is publicly available at `https://github.com/MLResearchX/lmc-moe`.

## 1 Introduction

Despite the high-dimensional and non-convex nature of deep neural network (DNN) training, stochastic optimization methods–such as stochastic gradient descent (SGD) and its variants–consistently find solutions that generalize well. This empirical success contrasts with the theoretical complexity of the loss landscape, which contains many local minima. A growing body of work has uncovered a surprising phenomenon known as *mode connectivity*, wherein minima found by independent SGD runs can often be connected by continuous low-loss paths.

---

[*]Co-first author
[†]Corresponding author

39th Conference on Neural Information Processing Systems (NeurIPS 2025).

**Mode Connectivity.** Related work on mode connectivity includes [34, 45, 67, 85, 61, 76, 91, 94]. Empirical studies demonstrate low-loss connections between independently trained models on MNIST and CIFAR-10 [31, 33, 21], and have shown that nearly any two solutions can be linked by a low-error curve [32]. The implications of mode connectivity extend beyond theoretical curiosity. It provides insight into why weight-space ensembling techniques, such as Stochastic Weight Averaging, yield improved generalization [39, 65, 89]. Moreover, it has proven useful in studying adversarial robustness [93], generalization theory [63, 44, 55], and the geometry of loss landscapes [35, 87, 56]. A more restrictive and analytically convenient variant of this phenomenon is *linear mode connectivity* (LMC), where two models are connected by a linear interpolation in parameter space along which the loss remains low [30, 24].

**Permutation Invariance.** A central challenge in observing LMC stems from the *permutation invariance* of neural networks: permuting neurons within a hidden layer leaves the network function unchanged [7, 23, 29, 9, 60]. Consequently, two functionally equivalent models may appear disconnected in parameter space unless one is suitably permuted. To address this, recent work considers LMC *up to permutation*, wherein a low-loss linear path exists after aligning hidden units [71, 2, 36]. Optimal Transport (OT) [86, 82, 78, 79, 80] methods have been proposed to compute soft matchings between neurons [71, 72], enabling applications such as model fusion in federated learning. Empirically, OT-based alignment achieves near-zero error barrier LMC between independently trained ResNets on CIFAR-10 [2], with connectivity improving with width and degrading with depth. Theoretically, dropout-stable networks have been shown to exhibit mode connectivity [48, 70], and [24] demonstrate that LMC up to permutation can already arise at initialization, especially in the NTK regime [41]. [28] further provide formal guarantees for LMC under OT alignment. These insights support the *convexity conjecture* [25], which posits that the SGD solution set is approximately convex once symmetries are accounted for. This view is strengthened by [68], who propose *simultaneous linear connectivity*, where a single model aligns linearly with multiple others. Additional studies explore the geometry of the solution space [3, 90] and identify star-shaped regions conducive to LMC [73].

**Functional Equivalence.** Prior work on LMC has primarily focused on feedforward and convolutional architectures, owing to the well-characterized permutation symmetries in these models [11, 62, 13, 77, 88]. In contrast, the symmetry structures of more modern architectures–such as Transformers [84, 19, 12] and Mixture-of-Experts [40, 69, 52, 26]–remain relatively underexplored. The study of these symmetries falls under the broader concept of *functional equivalence*, which aims to characterize when different parameter configurations yield identical input–output behavior [37, 15, 27, 50, 5, 6]. Recent work has extended this analysis to attention-based models, with [83, 46] examining symmetries in attention mechanisms, particularly permutations of heads and group actions induced by the general linear group. A rigorous understanding of functional equivalence is essential for uncovering and formalizing LMC in such complex, structured architectures.

**Permutation Alignment Methods.** These methods align parameter permutations to establish LMC. [25] proposed a simulated annealing-based algorithm. [72] employed Optimal Transport, while [4] utilized the Wasserstein Barycenter. [2] introduced three methods: activation matching (using intermediate activations), weight matching (in parameter space), and the Straight-Through Estimator (which minimizes interpolation midpoint loss via gradients); all are based on solving the Linear Assignment Problem [49, 42, 17]. [36] developed Sinkhorn re-basin, a differentiable method that improves alignment but struggles with residual connections due to layer-independent optimization.

**Contribution.** Motivated by this line of work, we extend the investigation of LMC to Mixture-of-Experts (MoE) architectures, which are related to traditional neural networks in that they aggregate outputs of multiple subnetworks via a gating mechanism. The paper is organized as follows:

1. In Section 3, we introduce the concept of the weight space of MoE architectures and define a group action on this space that preserves the functional behavior of MoE models.

2. In Section 4, we present two core results concerning functional equivalence in MoE models. We demonstrate that the proposed group action characterizes *all* inherent symmetries of the MoE gating mechanism, with rigorous theoretical justification.

3. In Section 5, we first observe that permutation invariance alone is sufficient to induce LMC. Building on this insight, we develop a Weight Matching algorithm that enables alignment between independently trained MoEs, thereby facilitating the discovery of LMC.

4. In Section 6, we provide empirical evidence of LMC across a wide range of MoE configurations–including dense, sparse, and shared-expert variants–evaluated under diverse model settings and across datasets of varying scales and modalities. Additionally, we assess the effectiveness of our proposed expert-matching algorithms, and conduct ablation studies to analyze LMC at different layers within deep MoE models.

Section 2 offers background on LMC and MoE architectures. Table of notation, along with theoretical foundations, experimental details, and additional content, is provided in the Appendix.

## 2 Preliminaries

This section provides an overview of Linear Mode Connectivity and Mixture-of-Experts architectures.

### 2.1 Linear Mode Connectivity

Let $f(\cdot; \theta)$ be a function parameterized by $\theta \in \Theta$, where $\Theta$ denotes the weight space. Given a non-negative loss function $\mathcal{L}(\theta)$, we seek to minimize $\mathcal{L}(\theta)$ over $\Theta$. The notion of the *loss barrier* was originally introduced by [30] and later formalized by [24] as follows: for $\theta_A, \theta_B \in \Theta$, define

$$B(\theta_A, \theta_B) = \sup_{t \in [0,1]} \left[ \mathcal{L}(t\theta_A + (1-t)\theta_B) - (t\mathcal{L}(\theta_A) + (1-t)\mathcal{L}(\theta_B)) \right]. \tag{1}$$

$\theta_A$ and $\theta_B$ are said to be *linearly mode connected* if their loss barrier is negligible, i.e., $B(\theta_A, \theta_B) \approx 0$. In many architectures, the function $f(\cdot; \theta)$ is invariant under certain permutations of the weight space–namely, for a permutation $g$ and all $\theta \in \Theta$, we have $f(\cdot; \theta) = f(\cdot; g\theta)$. Therefore, $\theta_A$ and $\theta_B$ are said to be *linearly mode connected up to permutation* if there exists a permutation $g$ such that $B(\theta_A, g\theta_B) \approx 0$. Any such permutation $g$ is referred to as a *winning permutation* [24].

### 2.2 Mixture-of-Experts Architectures

A Mixture-of-Experts (MoE) is a neural network architecture that combines the outputs of multiple models into a single prediction through a gating mechanism that assigns input-dependent weights. The two most commonly used gating strategies are *dense* and *sparse* gating. While we provide a brief overview of these concepts here, a comprehensive and formal treatment is deferred to Appendix A.

**Expert.** Let $d$ denote the input dimension. We define an *Expert* as a function $\mathcal{E}(\cdot; \theta) \colon \mathbb{R}^d \to \mathbb{R}^d$, parameterized by $\theta \in \mathbb{R}^e$, where $e \in \mathbb{N}$ represents the total number of trainable parameters of $\mathcal{E}$. In this work, we consider each expert $\mathcal{E}(\cdot; \theta)$ to be a feedforward neural network with ReLU activation functions. Unless stated otherwise, all experts are assumed to share the same architecture.

**Mixture-of-Experts with dense gating.** Given $n$ denoting the number of experts, we define *Mixture-of-Experts with dense gating* as a function $\mathcal{D} \colon \mathbb{R}^d \to \mathbb{R}^d$ such that

$$\mathcal{D}\big(x; \{W_i, b_i, \theta_i\}_{i=1}^n\big) = \sum_{i=1}^n \operatorname{softmax}_i\big(s_1(x), \ldots, s_n(x)\big)\mathcal{E}(x; \theta_i), \tag{2}$$

where $s_i(x) = W_i x + b_i$ determines the contribution of each expert to the final output. Here, $\theta_i$ are the parameters of the $i^{\text{th}}$ expert. The function $s = (s_1, \ldots, s_n)$ is called the *gating score*, and is parameterized as $s(\cdot; \{W_i, b_i\}_{i=1}^n)$ with $(W_i, b_i) \in \mathbb{R}^d \times \mathbb{R}$ are the corresponding *gating* parameters.

**Mixture-of-Experts with sparse gating.** Given a positive integer $k \leq n$ denoting the number of activated experts, define the Top-$k$ map by Top-$k(z) = \{i_1, \ldots, i_k\}$ for $z = (z_1, \ldots, z_n) \in \mathbb{R}^n$, where $i_1, \ldots, i_k$ are the indices corresponding to the $k$ largest components of $x$. In the event of ties, we select smaller indices first. We define a *Mixture-of-Experts with sparse gating* (SMoE) as a function $\mathcal{S} \colon \mathbb{R}^d \to \mathbb{R}^d$ as follows. For $x \in \mathbb{R}^d$, let $T(x) = \text{Top-}k(s(x))$, then define:

$$\mathcal{S}\big(x; \{W_i, b_i, \theta_i\}_{i=1}^n\big) = \sum_{i \in T(x)} \operatorname{softmax}_i\left(\big(s_i(x)\big)_{i \in T(x)}\right) \cdot \mathcal{E}(x; \theta_i) \tag{3}$$

In other words, the Top-$k$ selects the $k$ highest-scoring experts used to compute the output.

# 3 Group Action on Weight Space of Mixture-of-Experts

In this section, we introduce the formal notion of the weight space associated with MoE models. Furthermore, we define a group action on this space that preserves the functionality of MoE. A complete and rigorous exposition of these definitions and their implications is provided in Appendix A.

## 3.1 Weight Space of Mixture-of-Experts

The map $\mathcal{D}$ is parameterized by:

$$\phi = (W_i, b_i, \theta_i)_{i=1,\ldots,n} \in \Phi(n) \coloneqq (\mathbb{R}^d \times \mathbb{R} \times \Theta)^n = (\mathbb{R}^d \times \mathbb{R} \times \mathbb{R}^e)^n. \tag{4}$$

Here, $\Phi(n)$ is called the *weight space* of a Mixture-of-$n$-Experts. Varying the number of experts leads to a general Mixture-of-Experts weight space that spans across expert sets of different sizes, denoted by $\Phi = \sqcup_{n>0}\Phi(n)$. Note that the weight space of $\mathcal{E}$ coincides with that of $\mathcal{S}$, since the map Top-$k$ does not introduce any new trainable parameters.

## 3.2 Group Action on Weight Space of Mixture-of-Experts

We define the group $G(n)$ as the direct product $G(n) = \mathbb{R}^d \times \mathbb{R} \times \mathrm{S}_n$ of the groups $\mathbb{R}^d$, $\mathbb{R}$ with addition, and the permutation group $\mathrm{S}_n$. Each element $g \in G(n)$ is of the form $g = (c_W, c_b, \tau)$, where $c_W \in \mathbb{R}^d, c_b \in \mathbb{R}$ and $\tau \in \mathrm{S}_n$. The group $G(n)$ acts on the weight space $\Phi(n)$ as follows. For $g \in G(n)$ and $\phi \in \Phi(n)$ presented as in Equation (4), define:

$$g\phi \coloneqq \big(W_{\tau(i)} + c_W, b_{\tau(i)} + c_b, \theta_{\tau(i)}\big)_{i=1,\ldots,n} \in \Phi(n). \tag{5}$$

The result below establishes that this group action preserves the MoE map.

**Proposition 3.1** (Weight space invariance of Mixture-of-Experts)**.** *The MoE function $\mathcal{D}$ is $G(n)$-invariant under the action of $G(n)$ on its weight space $\Phi(n)$, i.e., $\mathcal{D}(\cdot; \phi) = \mathcal{D}(\cdot; g\phi)$.*

A proof of Proposition 3.1 is presented in Proposition A.3. An analogous invariance result holds in the case of the SMoE function $\mathcal{S}$. However, since the Top-$k$ selection map is generally discontinuous–primarily due to tie cases in the gating scores–additional conditions are required to ensure the validity of the invariance result. To address this, we focus on a subset of $\mathbb{R}^d$ where the Top-$k$ scores are unambiguously defined. Specifically, for $\{W_i, b_i\}_{i=1}^n \in (\mathbb{R}^d \times \mathbb{R})^n$, we define:

$$\Omega\big(\{W_i, b_i\}_{i=1}^n\big) \coloneqq \big\{x \in \mathbb{R}^d : (W_i x + b_i)_{i=1}^n \text{ are pairwise distinct}\big\}. \tag{6}$$

The SMoE function $\mathcal{S}$ exhibits well-behaved properties on this domain. First, consider the case where the pairs $\{W_i, b_i\}_{i=1}^n$ are not pairwise distinct. In this case, the set $\Omega(\{W_i, b_i\}_{i=1}^n)$ is clearly empty. However, when the pairs $\{W_i, b_i\}_{i=1}^n$ are pairwise distinct, the set $\Omega(\{W_i, b_i\}_{i=1}^n)$ becomes an open and dense subset of $\mathbb{R}^d$. Moreover, the SMoE function $\mathcal{S}$ is continuous on $\Omega(\{W_i, b_i\}_{i=1}^n)$. These properties are formally established in Propositions A.1 and A.2. We now show that the invariance property of the SMoE map holds under restriction to this domain.

**Proposition 3.2** (Weight space invariance of Sparse Mixture-of-Experts)**.** *Given the SMoE function $\mathcal{S}$, as defined in Equation (3). Assume that $\{W_i, b_i\}$ are pairwise distinct for $i = 1, \ldots, n$. Then, the set $\Omega(\{W_i, b_i\}_{i=1}^n)$ is invariant under the group action of $G(n)$, i.e., for $g = (c_W, c_b, \tau) \in G(n)$, we have $\Omega(\{W_i, b_i\}_{i=1}^n) = \Omega(\{W_{\tau(i)} + c_W, b_{\tau(i)} + c_b\}_{i=1}^n)$. Moreover, the function $\mathcal{S}$, restricted to $\Omega(\{W_i, b_i\}_{i=1}^n)$, is invariance under the action of $G(n)$ on its weight space $\Phi(n)$, i.e. $\mathcal{S}(\cdot; \phi) = \mathcal{S}(\cdot; g\phi)$ on $\Omega(\{W_i, b_i\}_{i=1}^n)$.*

A proof of Proposition 3.2 is presented in Proposition A.4.

**Remark 3.3.** The permutation invariance of the summation operator and the translation invariance of the softmax function are the two primary sources of invariance for the functions $\mathcal{D}$ and $\mathcal{S}$, as stated in Propositions 3.1 and 3.2. In the case of SMoE, these invariance properties are additionally upheld by the permutation and translation invariance of the Top-$k$ operator.

# 4 Symmetries and Functional Equivalence in Mixture-of-Experts

In the remainder of this section, we let $\phi \in \Phi(n)$ and $\phi' \in \Phi(n')$ denote the parameters of two Mixture-of-Experts models with $n$ and $n'$ experts, respectively:

$$\phi = (W_i, b_i, \theta_i)_{i=1,\ldots,n} \in \Phi(n), \quad \text{and} \quad \phi' = (W_i', b_i', \theta_i')_{i=1,\ldots,n'} \in \Phi(n'). \tag{7}$$

## 4.1 Functional Equivalence in Mixture-of-Experts

We aim to characterize the conditions under which distinct parameter sets yield functionally equivalent MoE models. The central result of this section is that *the symmetries of the gating mechanism are fully characterized by the group action of* $G(n)$, as defined in Equation (5). Given the fundamentally different structural and analytical characteristics of the two gating mechanisms, we treat each case independently in the analysis. In addition, we introduce a set of assumptions, the motivations for which are discussed in detail in the following subsection. We begin with the case of dense gating.

**Theorem 4.1** (Functional equivalence in Mixture-of-Experts with Dense Gating). *Suppose* $\phi, \phi'$ *define the same MoE function, i.e.,* $\mathcal{D}(\cdot; \phi) = \mathcal{D}(\cdot; \phi')$. *Assume that the following conditions hold:*

1. *Both* $\{\mathcal{E}(\cdot; \theta_i)\}_{i=1}^n$ *and* $\{\mathcal{E}(\cdot; \theta_i')\}_{i=1}^{n'}$ *consist of pairwise distinct functions;*

2. *Both* $\{W_i - W_j\}_{1 \le i < j \le n}$ *and* $\{W_i' - W_j'\}_{1 \le i < j \le n'}$ *consist of pairwise distinct vectors in* $\mathbb{R}^d$.

*Then* $n = n'$, *and there exists* $g = (c_W, c_b, \tau) \in G(n)$ *such that for all* $i = 1, \ldots, n$, *we have* $W_i' = W_{\tau(i)} + c_W$, $b_i' = b_{\tau(i)} + c_b$, *and* $\mathcal{E}(\cdot; \theta_i') = \mathcal{E}(\cdot; \theta_{\tau(i)})$ *on* $\mathbb{R}^d$.

For the case of sparse gating, we require the notion of the *strongly distinct* property. Specifically, two functions $f$ and $g$ defined on $\mathbb{R}^d$ are said to be *strongly distinct* if the set $\{x \in \mathbb{R}^d : f(x) \ne g(x)\}$ is dense in $\mathbb{R}^d$. For instance, distinct polynomials are strongly distinct, whereas distinct ReLU networks are not strongly distinct in general. A formal definition of this property, along with illustrative examples, is provided in Definition C.1 and Example C.2.

We now state a result that parallels Theorem 4.1, adapted to the sparse gating regime with $k > 1$, and established under a set of assumptions that are stronger than those previously required.

**Theorem 4.2** (Functional equivalence in Mixture-of-Experts with Sparse Gating). *Suppose* $\phi, \phi'$ *define the same SMoE function, i.e.,* $\mathcal{S}(\cdot; \phi) = \mathcal{S}(\cdot; \phi')$. *Assume that the following conditions hold:*

1. *Both* $\{\mathcal{E}(\cdot; \theta_i)\}_{i=1}^n$ *and* $\{\mathcal{E}(\cdot; \theta_i')\}_{i=1}^{n'}$ *consist of pairwise strongly distinct functions;*

2. *Both* $\{W_{i-1} - W_i\}_{i=2}^n$ *and* $\{W_{i-1}' - W_i'\}_{i=2}^{n'}$ *are linearly independent subsets of* $\mathbb{R}^d$.

*Then* $n = n'$, *and there exists* $g = (c_W, c_b, \tau) \in G(n)$ *such that for all* $i = 1, \ldots, n$, *we have* $W_i' = W_{\tau(i)} + c_W$, $b_i' = b_{\tau(i)} + c_b$, *and* $\mathcal{E}(x; \theta_i') = \mathcal{E}(x; \theta_{\tau(i)})$ *for all* $x \in \Omega(\{W_i, b_i\}_{i=1}^n)$ *such that* $\tau(i) \in \text{Top-}k((W_j x + b_j)_{j=1}^n)$.

The proofs of Theorems 4.1 and 4.2 are provided in Appendix B and Appendix C, respectively. Both arguments build upon two essential ingredients: a result establishing the linear independence of exponential functions, as formulated in Lemma B.3, and a key observation on the piecewise affine structure of ReLU networks, discussed in Appendix B.1.

**Remark 4.3.** While Theorem 4.2 shares a conceptual parallel with Theorem 4.1, it is important to underscore that *the sparse gating case presents significantly deeper mathematical difficulties*. The core source of complexity lies in the discontinuous behavior of the Top-$k$ operator, which induces nontrivial discontinuities in the gating function, thereby disrupting smoothness and complicating the functional analysis required to establish equivalence. As a result, standard analytical techniques used in the dense setting become insufficient, necessitating more delicate arguments to account for the combinatorial and piecewise structure inherent to sparse gating.

Theorems 4.1 and 4.2 provide a rigorous characterization of functional equivalence in both dense and sparse MoE models, with particular attention given to the role and structure of the gating mechanism. Nonetheless, it is important to note that these results do not account for all potential symmetries inherent in the MoE and SMoE mappings defined in Equations (2) and (3). Notably, further symmetries may exist within the expert networks, especially when they are implemented as ReLU neural networks. Given that this work is primarily concerned with the MoE architecture itself, our analysis is restricted to the behavior of the gating mechanism. Consequently, we treat the experts as black-box functions and abstract away from their internal parameterizations.

## 4.2 Necessity and Implications of Technical Assumptions

At a conceptual level, symmetry analysis aims to identify *universal invariances*–those that persist regardless of specific parameter values–while explicitly excluding *singular symmetries*, which emerge only under special, degenerate configurations of the model parameters. The assumptions introduced in our results are designed precisely to rule out such singularities, which do not represent inherent structural invariances of the architecture but instead arise from pathological or measure-zero subsets of parameter space. We provide a brief overview of the motivation behind these assumptions here; a more detailed justification, along with illustrative examples, is given in Remarks B.6 and C.8.

**The case of dense gating.** Assumption 1 is introduced to eliminate degenerate scenarios in which two experts compute identical functions and receive identical gating scores–situations where permuting the corresponding experts has no effect on the model's output. By excluding linear dependencies among the gating weight vectors, Assumption 2 ensures that expert activations remain distinguishable–thereby preventing the emergence of non-structural, symmetry-like artifacts.

**The case of sparse gating.** Theorem 4.2 relies on a stronger set of assumptions than those required in Theorem 4.1. This added strength is necessary due to a key feature of sparse gating: an expert's behavior on inputs where it is not selected is unconstrained, meaning that the expert can act arbitrarily outside its region of activation. Consequently, different sets of expert functions may result in the same overall model output, provided their outputs agree where they are active.

**The case of $k = 1$.** In the particular case where $k = 1$, the SMoE architecture employs a Top-1 gating mechanism that activates only the expert associated with the highest gating score. Consequently, the softmax output reduces to a one-hot vector, with a single component equal to 1. Under this regime, the SMoE map exhibits additional nontrivial symmetries, specifically invariance under the multiplicative group $\mathbb{R}_{>0}$. That is, for any positive scalar $c > 0$, the following identity holds:

$$\mathcal{S}\big(x; \{W_i, b_i, \theta_i\}_{i=1}^n\big) = \mathcal{S}\big(x; \{cW_i, cb_i, \theta_i\}_{i=1}^n\big). \tag{8}$$

This invariance holds because the argmax used for expert selection is unaffected by positive scaling, i.e., $\mathrm{argmax}_{i=1,\dots,n} (W_i x + b_i) = \mathrm{argmax}_{i=1,\dots,n} (cW_i x + cb_i)$, for all $x \in \Omega(\{W_i, b_i\}_{i=1}^n)$. Additionally, since only a single expert contributes to the output, the model lacks any form of explicit aggregation across experts. Due to the analytical challenges posed by these additional invariances, we do not consider $k = 1$ in our primary results and instead defer its investigation to future work.

# 5 Linear Mode Connectivity of Mixture-of-Experts Architectures

This section outlines the theoretical foundations of LMC in MoE architectures. We first show that permutation invariance suffices to explain the existence of LMC in MoEs, then propose an algorithm to identify such permutations, which we later employ to empirically assess LMC in MoE models.

## 5.1 Permutation Invariance Sufficiency in Linear Mode Connectivity of Mixture-of-Experts

Theorems 4.1 and 4.2 show that the group action as in Equation (5) suffices to uncover LMC between two MoE models. Each group element $g = (c_W, c_b, \tau) \in G(n)$ consists of two components: a translation term $(c_W, c_b) \in \mathbb{R}^d \times \mathbb{R}$ applied to the gating function, and a permutation $\tau$ that reorders both the experts and their associated gating scores. Observe that the translation component does not affect the barrier loss defined in Equation (1). Indeed, for any $h = (c_W, c_b, \mathrm{id}_n) \in G(n)$–with $\mathrm{id}_n$ denoting the identity permutation in $S_n$–the loss barrier remains unchanged:

$$B(\phi_A, \phi_B) = B(\phi_A, h\phi_B). \tag{9}$$

This invariance, established in Proposition D.1, indicates that only the permutation component $\tau$ influences LMC behavior. Combined with prior work showing that permutation symmetries are sufficient to induce LMC in standard neural networks [24, 28], this suggests that analyzing permutations alone suffices to capture LMC in MoE models.

## 5.2 Permutation Alignment Algorithm for Mixture-of-Experts

We propose the Permutation Alignment Algorithm to align MoEs. Inspired by the data-independent Weight Matching algorithm [3], our method operates in the parameter space of two MoEs, enabling efficient permutation alignment tailored to MoE architectures in large-scale settings.

Table 1: Configurations for experiments analyzing LMC in Transformers with dense MoE replacing the FFN in the *first* layer (refer to Table 5 for SMoE and DeepSeekMoE). For each dataset, number of layers, and number of experts, we visualize LMC curves for the *three* model pairs among *three* models in the referenced figures. Datasets denoted as $A \to B$ indicate pretraining on dataset $A$, fine-tuning on dataset $B$, and evaluating on $B$.

| Dataset | Layers | Experts | Figure | Dataset | Layers | Experts | Figure |
|---|---|---|---|---|---|---|---|
| MNIST | 1 | [2, 4] | [5, 6] | ImageNet-21k→CIFAR-10 | 12 | [2, 4, 6] | [18, 19, 20] |
|  | 2 | [2, 4] | [7, 8] | ImageNet-21k→CIFAR-100 | 12 | [2, 4, 6] | [21, 22, 23] |
| CIFAR-10 | 2 | [2, 4, 6] | [9, 10, 11] | ImageNet-1k | 12 | [2, 4, 6, 8] | [24, 25, 26, 27] |
|  | 6 | [2, 4, 6] | [12, 13, 14] | WikiText103 | 12 | [2, 4, 6, 8] | [28, 29, 30, 31] |
| CIFAR-100 | 6 | [2, 4, 6] | [15, 16, 17] | One Billion Word | 12 | [2, 4, 6, 8] | [32, 33, 34, 35] |

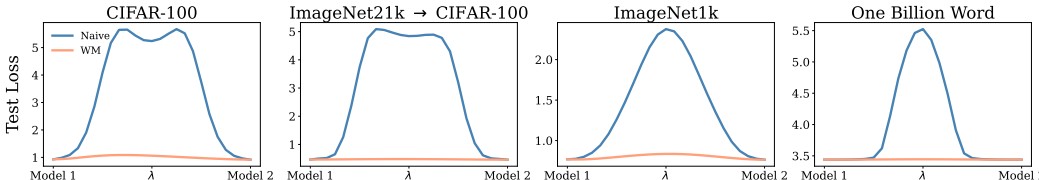

Figure 1: LMC curves for ViT (subplots 1-3) and GPT-2 (subplot 4) with a 4-expert MoE replacement at the *first* Transformer layer, on CIFAR-100, ImageNet21k→CIFAR-100, ImageNet-1k, and One Billion Word datasets, respectively. Plots show consistent low-loss linear interpolation paths between fine-tuned models, indicating strong linear mode connectivity.

**Proposed Algorithm.** In MoEs, equivalent functionality can arise from different expert orderings and internal neuron configurations. To address this, we propose a two-stage alignment method that first reorders experts to match their roles, then aligns the internal weights of corresponding experts. Both steps are formulated as Linear Assignment Problems (LAPs) [10]. An LAP assigns $N$ tasks to $N$ agents using a cost matrix $C \in \mathbb{R}^{N \times N}$, where each entry $C_{i,j}$ represents the cost of assigning task $i$ to agent $j$. The objective is to find a permutation $\pi$ that minimizes the total cost: $\min_{\pi \in S_N} \sum_{i=1}^{N} C_{i,\pi(i)}$. This is efficiently solved using the Hungarian algorithm in $O(N^3)$ time [49, 42, 17].

Suppose we have $\phi = (W_i, b_i, \theta_i)_{i=1,\dots,n}$ and $\phi' = (W_i', b_i', \theta_i')_{i=1,\dots,n}$ representing the parameters of two distinct MoEs with $n$ experts. In practice, each expert is implemented as an MLP with one hidden layer [26, 52, 22]. We denote the parameters of expert $i$ in $\phi$ and expert $j$ in $\phi'$ as $\theta_i = \{A_i, u_i, B_i, v_i\}$ and $\theta_j' = \{A_j', u_j', B_j', v_j'\}$, respectively, where $A_i \in \mathbb{R}^{h \times d}$, $u_i \in \mathbb{R}^h$, $B_i \in \mathbb{R}^{d \times h}$, $v_i \in \mathbb{R}^d$, with $h$ as the hidden dimension. Denote $\widetilde{A}_i = [A_i, u_i] \in \mathbb{R}^{h \times (d+1)}$, $\widetilde{B}_i = [B_i, v_i] \in \mathbb{R}^{d \times (h+1)}$, and similarly for $\widetilde{A}_j'$ and $\widetilde{B}_j'$, as concatenated matrices.

**Matching Order of Experts.** In MoEs, experts and their gating can be reordered without affecting functionality. We propose two methods to align the expert order between $\phi$ and $\phi'$, using LAP.

*Method 1 - Gating Weights.* To address the translation-invariance of the softmax function in the gating, we center the weights and biases of the gating. For $\phi$, these are $\widehat{W}_i = W_i - \frac{1}{n} \sum_{m=1}^{n} W_m$ and $\hat{b}_i = b_i - \frac{1}{n} \sum_{m=1}^{n} b_m$ per expert $i$, and similarly for $\phi'$. The *cost matrix with respect to gating weights* is defined as $C = \{C_{i,j}\}_{1 \leq i \leq n, 1 \leq j \leq n} \in \mathbb{R}^{n \times n}$, where

$$C_{i,j} = \left( \|\widehat{W}_i - \widehat{W}_j'\|_2^2 + (\hat{b}_i - \hat{b}_j')^2 \right)^{\frac{1}{2}}, \tag{10}$$

measuring dissimilarity between centered gating parameters of expert $i$ in $\phi$ and expert $j$ in $\phi'$.

*Method 2 - Internal Weights.* Experts exhibit permutation invariance in hidden neuron orderings. We use Gram matrices for a permutation-invariant representation (see Appendix D.2). The *cost matrix*

*with respect to expert internal weights* is defined as $C = \{C_{i,j}\}_{1 \leq i \leq n, 1 \leq j \leq n} \in \mathbb{R}^{n \times n}$, where

$$C_{i,j} = \left( \left\| (\widetilde{A}_i)^\top \widetilde{A}_i - (\widetilde{A}'_j)^\top \widetilde{A}'_j \right\|_F^2 + \left\| \widetilde{B}_i (\widetilde{B}_i)^\top - \widetilde{B}'_j (\widetilde{B}'_j)^\top \right\|_F^2 \right)^{\frac{1}{2}}, \quad (11)$$

quantifying dissimilarity between weight representations of expert $i$ in $\phi$ and expert $j$ in $\phi'$.

For both methods, the optimal expert ordering $\tau$ is determined by solving the LAP: $\tau = \arg\min_{\pi \in S_n} \sum_{i=1}^n C_{i,\pi(i)}$ in $O(n^3)$ time.

**Aligning Expert Internal Weights.** After finding the permutation $\tau$ to align experts between $\phi$ and $\phi'$, we align the internal weights of each matched expert pair $(i, \tau(i))$ using established MLP alignment methods [2, 36, 25, 72], specifically the Weight Matching algorithm [3]. This LAP-based algorithm aligns two MLPs with one hidden layer of size $h$ in $O(h^3)$ time, ensuring efficient scaling. Applying this to each pair $(i, \tau(i))$ guarantees consistent neuron ordering across aligned experts.

### 5.3 Weight Matching Algorithm

We propose a Weight Matching Algorithm designed to align MoEs by addressing permutation invariance in expert ordering and internal parameters, as detailed in Algorithm 1. The algorithm employs gate-based and expert-based methods to optimally permute experts, with both performing comparably well in loss barrier evaluations during model interpolation (see Section 6.3). Operating solely in parameter space, the algorithm avoids data-dependent computational overhead, with a complexity of $O(n^3 + nh^3)$, ensuring scalability for large-scale MoE alignment tasks.

---

**Algorithm 1** Weight Matching for Mixture-of-Experts

---

    **Input:** MoE model weights $\phi = (W_i, b_i, \theta_i)_{i=1,\dots,n}$, $\phi' = (W'_i, b'_i, \theta'_i)_{i=1,\dots,n}$
    **Output:** Permutation $\tau$ for experts, and permutations $\{P_i\}_{i=1}^n$ for hidden units
    % Step 1: Match experts' order using two methods
    **for** method in {gate, expert} **do**
        Compute cost matrix $C$
        Solve LAP to obtain expert permutation $\tau_{\text{method}}$
    **end for**
    % The two candidate expert orderings $\tau_{\text{gate}}$ and $\tau_{\text{expert}}$ are obtained
    % Step 2: Align internal weights of matched expert pairs
    **for** method in {gate, expert} **do**
        **for** $i = 1$ to $n$ **do**
            Compute $P_i$ by applying Weight Matching to $\theta_i$ and $\theta'_{\tau_{\text{method}}(i)}$
        **end for**
    **end for**
    **return** $\left( \tau_{\text{gate}}, (\{P_i\}_{i=1}^n)_{\text{gate}} \right), \left( \tau_{\text{expert}}, (\{P_i\}_{i=1}^n)_{\text{expert}} \right)$

---

## 6 Experiments

This section provides empirical validation of LMC across *three* MoE variants, including dense MoE, sparse MoE (SMoE), and DeepSeekMoE, detailed in Section 6.1. By analyzing interpolation paths between independently fine-tuned models–each subject to permutation invariance–low-loss connections are consistently observed, indicating shared solution basins in the optimization landscape.

### 6.1 Experimental Setup

**Experimental Design.** We investigate LMC by replacing the Feedforward Network (FFN) in the Transformer [84] layer with a randomly initialized MoE, based on empirical evidence from Section 6.4, Appendices E, G.2, and G.3 indicating lower perturbation sensitivity when replacing deeper FFN layers with MoEs. Only the MoE parameters are fine-tuned using multiple random seeds for each experiment. LMC is evaluated by linearly interpolating between all checkpoint pairs, measuring model performance on the *test* set at 25 evenly spaced points along the interpolation

Table 2: Comparison of the two expert matching methods for 12-layer ViT-MoE models with 4 experts on CIFAR-10 and CIFAR-100, showing Rank (out of 24 permutations) and $\hat{L} = \frac{L_{\text{method}} - L_{\text{top1}}}{L_{\text{naive}} - L_{\text{top1}}} \times 10^2$ across layer placements, averaged over 10 checkpoint pairs. Please refer to Tables 8 and 9 for SMoE and DeepSeekMoE variants.

| Dataset | Layer replaced | Expert Weight Matching | | Gate Weight Matching | | Dataset | Layer replaced | Expert Weight Matching | | Gate Weight Matching | |
|---|---|---|---|---|---|---|---|---|---|---|---|
| | | Rank ↓ | $\hat{L}$ ↓ | Rank ↓ | $\hat{L}$ ↓ | | | Rank ↓ | $\hat{L}$ ↓ | Rank ↓ | $\hat{L}$ ↓ |
| CIFAR-10 | 1 | $2.50 \pm 1.50$ | $2.12 \pm 0.42$ | $3.00 \pm 1.00$ | $3.42 \pm 0.55$ | CIFAR-100 | 1 | $2.80 \pm 0.40$ | $3.17 \pm 0.25$ | $2.90 \pm 0.70$ | $2.73 \pm 1.03$ |
| | 4 | $2.10 \pm 0.50$ | $1.04 \pm 0.32$ | $2.60 \pm 0.92$ | $1.54 \pm 0.44$ | | 4 | $3.60 \pm 1.20$ | $1.15 \pm 0.55$ | $2.70 \pm 1.00$ | $2.03 \pm 0.93$ |
| | 8 | $2.70 \pm 0.46$ | $0.60 \pm 0.27$ | $2.90 \pm 0.30$ | $0.74 \pm 0.17$ | | 8 | $3.30 \pm 0.78$ | $0.67 \pm 0.14$ | $3.20 \pm 0.87$ | $1.13 \pm 0.55$ |
| | 12 | $4.60 \pm 2.00$ | $0.13 \pm 0.05$ | $3.80 \pm 1.66$ | $0.09 \pm 0.03$ | | 12 | $3.40 \pm 0.92$ | $0.07 \pm 0.03$ | $4.20 \pm 0.89$ | $0.11 \pm 0.04$ |

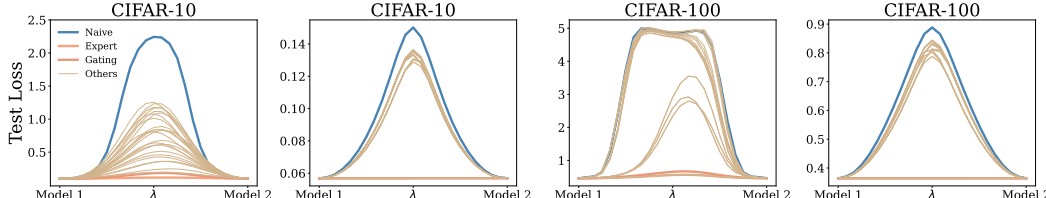

Figure 2: Loss curves for 12-layer ViT-MoE models with a 4-expert MoE replacement in either the first layer (subplots 1, 3) or the last layer (subplots 2, 4), on CIFAR-10 and CIFAR-100. The curves compare two Expert Order Matching methods across 24 permutations, with Weight Matching applied post-reordering to all permutations. Corresponding accuracy metrics are presented in Figure 96.

path. We examine three MoE variants: dense MoE [40, 43], SMoE with top-2 routing ($k = 2$) [26], and DeepSeekMoE with top-2 and one shared expert ($k = 2, s = 1$) (see Appendix D.3 for formal formulations) [54].

**Datasets and Models.** We use ViT [20] for image classification (MNIST [51], CIFAR-10/100 [47], ImageNet [18]) and GPT-2 [64] for language modeling (WikiText103 [59] and One Billion Word [14]). Hyperparameters such as batch size, optimizer, number of experts, and hidden size are fixed, while the learning rate is tuned per setting. Full details are provided in Appendix F.

## 6.2 Linear Mode Connectivity Verification

**Experimental Objective.** We analyze LMC in pretrained Transformer models where the FFN of a Transformer layer is replaced with an MoE. Our study focuses on two configurations: replacing the FFN in the *first*, *last* and *all* Transformer layers. For each configuration, we fine-tune *three* independently initialized models. We vary the number of experts (e.g., 2, 4, 6, 8, 16) and the number of Transformer layers to assess LMC under different architectural scales. LMC is evaluated by linearly interpolating between all model pairs using our proposed algorithm (Algorithm 1).

**Results.** Table 1 shows results for first-layer MoE replacement, with full details for three MoE variants in Table 5. The results for both the last-layer and all-layer configurations are provided in Appendix G.2 and G.3. LMC persists in all cases, but is less pronounced in the last layer due to greater stability and reduced influence on convergence. Early layers strongly shape the solution basin, making first-layer changes more revealing of connectivity. This depth-dependent effect is also demonstrated in Section 6.4 and Appendix E. Figure 1 presents representative LMC curves across our testing cases.

## 6.3 Expert Order Matching Method

**Experimental Objective.** We evaluate two Expert Order Matching methods (Step 1, Algorithm 1)– Expert Weight Matching and Gate Weight Matching–using 12-layer ViT models with 4-expert MoE replacements. The methods are assessed by ranking the selected permutation among all 24 possible expert permutations after Weight Matching (Step 2), with MoE integrations at layers 1, 4, 8, or 12 on CIFAR-10 and CIFAR-100, repeated five times. We report the rank and a scaled metric

Table 3: Ablation Study on varying the number of Transformer layers and MoE layer replacement in ViT for CIFAR-10. The ratios of metrics (loss barrier and AUC, accuracy barrier and AUC) compare Algorithm 1 to naive interpolation. For all MoE variants, kindly refer to Tables 11, 12, and 13.

| Layers | Layer replaced | Loss Barrier ↓ | Loss AUC ↓ | Accuracy Barrier ↓ | Accuracy AUC ↓ | Layers | Layer replaced | Loss Barrier ↓ | Loss AUC ↓ | Accuracy Barrier ↓ | Accuracy AUC ↓ |
|---|---|---|---|---|---|---|---|---|---|---|---|
| 2 | 1 | $8.54 \pm 1.53$ | $8.73 \pm 1.68$ | $8.39 \pm 1.42$ | $8.01 \pm 1.39$ | 6 | 1 | $4.16 \pm 1.47$ | $4.96 \pm 2.31$ | $3.64 \pm 1.00$ | $4.13 \pm 1.55$ |
|  | 2 | $9.33 \pm 2.04$ | $8.94 \pm 1.92$ | $9.10 \pm 2.19$ | $8.83 \pm 2.22$ |  | 2 | $6.78 \pm 4.80$ | $7.71 \pm 3.09$ | $7.27 \pm 5.73$ | $8.98 \pm 2.94$ |
| 4 | 1 | $8.29 \pm 1.63$ | $8.08 \pm 1.59$ | $7.41 \pm 1.45$ | $6.43 \pm 1.09$ |  | 3 | $9.69 \pm 4.92$ | $8.62 \pm 5.46$ | $9.03 \pm 3.31$ | $8.14 \pm 3.57$ |
|  | 2 | $10.19 \pm 3.43$ | $10.19 \pm 3.16$ | $10.21 \pm 5.26$ | $9.08 \pm 5.33$ |  | 4 | $10.83 \pm 5.61$ | $12.01 \pm 6.42$ | $8.92 \pm 2.21$ | $10.00 \pm 7.30$ |
|  | 3 | $8.50 \pm 1.82$ | $7.83 \pm 1.89$ | $9.82 \pm 2.58$ | $8.70 \pm 2.58$ |  | 5 | $9.56 \pm 4.52$ | $11.72 \pm 5.59$ | $8.72 \pm 2.89$ | $11.01 \pm 3.83$ |
|  | 4 | $5.12 \pm 0.67$ | $4.60 \pm 0.65$ | $4.17 \pm 1.68$ | $4.58 \pm 1.01$ |  | 6 | $6.90 \pm 3.54$ | $9.67 \pm 6.59$ | $7.16 \pm 3.98$ | $9.69 \pm 3.11$ |

$\hat{L} = \frac{L_{\text{method}} - L_{\text{top1}}}{L_{\text{naive}} - L_{\text{top1}}} \times 10^2$, averaged over ten checkpoint pairs, where $L_{\text{method}}$, $L_{\text{top1}}$, and $L_{\text{naive}}$ denote loss barriers of our methods, the best permutation, and the naive interpolation, respectively.

**Results.** Table 2 shows both methods achieve low ranks and near-zero $\hat{L}$, indicating near-optimal matching. Full results are presented in Tables 8 (loss) and 9 (accuracy). Figure 2 visualizes the representative performance of our methods across all 24 permutations, highlighting the importance of correct expert order matching, as poor ordering can yield performance close to the naive interpolation. To further validate the importance of expert alignment, we extend our analysis to the large-scale One Billion Word dataset. As shown in Table 10, the proposed Total Weight Matching consistently yields lower loss barriers compared to the Skipping-Expert-Order Matching baseline across all MoE variants, confirming that incorrect expert ordering substantially hampers interpolation smoothness. These results demonstrate that accurate Expert-Order Matching is crucial for revealing true Linear Mode Connectivity between MoE models, as even minor permutation mismatches can disrupt the underlying shared representation space.

## 6.4 Ablation Study on Number of Layers

This study investigates the impact of the number of Transformer layers on LMC in MoE models on CIFAR-10. We evaluate models with 2, 4, and 6 layers, incorporating MoE at all possible layer positions. Employing Algorithm 1, we compute ratios for four metrics–loss barrier, loss Area Under the Curve (AUC, relative to the straight line connecting the metrics of the two endpoint models), accuracy barrier, and accuracy AUC, relative to linear interpolation–across five repeated experiments, evaluating ten model pairs per configuration. Table 3 reports results for MoE, showing significant reduction ratios across all configurations. Corresponding results for SMoE and DeepSeekMoE are provided in Table 11, with detailed loss and accuracy metrics presented in Tables 12 and 13.

## 7 Conclusion

This paper presents a systematic analysis of LMC in MoE models. We show that functional equivalence via permutation symmetries is sufficient to construct low-loss linear paths–when they exist–between independently trained models. To enable this, we introduce an algorithm for identifying such permutations. Empirical evaluations confirm the presence of LMC across a wide range of MoE configurations–including dense, sparse, and shared-expert variants–under diverse hyperparameter settings and across datasets of varying scales and modalities. While our method does not provide theoretical bounds on the loss barrier, this remains a common limitation across prior LMC studies. Overall, our work offers a principled foundation for extending LMC analysis to other architectures, such as Transformers and State Space Models.

## Acknowledgments and Disclosure of Funding

This research / project is supported by the National Research Foundation Singapore under the AI Singapore Programme (AISG Award No: AISG2-TC-2023-012-SGIL). This research / project is supported by the Ministry of Education, Singapore, under the Academic Research Fund Tier 1 (FY2023) (A-8002040-00-00, A-8002039-00-00). This research / project is also supported by the NUS Presidential Young Professorship Award (A-0009807-01-00) and the NUS Artificial Intelligence Institute–Seed Funding (A-8003062-00-00).

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

## Table of Notation

| | |
|---|---|
| $\mathbb{R}^d$ | $d$-dimensional Euclidean space |
| $\mathbb{S}^{d-1}$ | $(d-1)$-dimensional hypersphere |
| $\|\cdot\|_2$ | Euclidean norm |
| $\|\cdot\|_F$ | Frobenius norm |
| $\mathcal{E}$ | Expert function |
| $\mathcal{D}$ | Mixture-of-Experts with Dense gating |
| $\mathcal{S}$ | Mixture-of-Experts with Sparse gating |
| Top-$k(\cdot)$ | The Top-$k$ map |
| $\theta, \theta'$ | parameter of Expert functions |
| $\Theta$ | weight space of Expert functions |
| $\phi, \phi'$ | parameter of Mixture-of-Experts |
| $\Phi(n)$ | weight space of Mixture-of-$n$-Experts |
| $\mathsf{S}_n$ | permutation group of order $n$ |
| $\tau$ | element of permutation group |
| $c_W, c_b$ | elements of additive group of Euclidean space |
| $G, G(n)$ | group act on weight space of Mixture-of-$n$-Experts |
| $\Omega, \Omega_1, \Omega_2$ | sets with specific purposes |
| $B$ | barrier loss |
| $C, C_{ij}$ | cost matrix and its entries |
| $A, B, u, v$ | weight and bias of Expert functions |
| $P$ | permutation matrix |
| $\partial$ | boundary of a set in a topological space |
| $\sigma$, ReLU | rectifier activation function |
| $\mathcal{H}$ | space of holomorphic functions |
| $\mathcal{F}$ | space of meromorphic functions |
| $\mathbb{C}[x], \mathbb{C}[x_1, \ldots, x_n]$ | polynomial ring in complex variables |
| $\mathbb{C}(x), \mathbb{C}(x_1, \ldots, x_n)$ | field of rational functions |
| $p_i, r_i$ | polynomial |

# Appendix of "On Linear Mode Connectivity of Mixture-of-Experts Architectures"

**Table of Contents**

# A  On the Weight Spaces of Mixture-of-Experts Architecture

Denote the input token dimension as $d$.

## A.1  Weight Space of Mixture-of-Experts

**Expert functions.** We study expert functions $\mathcal{E} \colon \mathbb{R}^d \to \mathbb{R}^d$, which are realized by standard feedforward neural networks employing the ReLU activation function. Specifically, each expert map $\mathcal{E}(\cdot; \theta)$ is represented as a composition of affine transformations and ReLU nonlinearities:

$$\mathcal{E}(x; \theta) = f_L \circ \sigma \circ f_{L-1} \circ \cdots \circ \sigma \circ f_1(x), \tag{12}$$

where each affine layer is defined by

$$f_i(x) = xW_i + b_i, \quad \text{for } i = 1, \ldots, L. \tag{13}$$

Here, $\sigma$ denotes the ReLU activation function, applied component-wise, and $\theta = \{W_i, b_i\}_{i=1}^{L}$ represents the full set of learnable parameters. The weight matrix $W_i \in \mathbb{R}^{n_{i-1} \times n_i}$ and bias vector $b_i \in \mathbb{R}^{n_i}$ parameterize the $i$-th layer, where $n_0 = n_L = d$ is the input and output dimensions, respectively. We have

$$\theta \in \prod_{i=1}^{L} (\mathbb{R}^{n_{i-1} \times n_i} \times \mathbb{R}^{n_i}) = \mathbb{R}^e, \quad \text{where} \quad e = \sum_{i=1}^{L} (n_{i-1} n_i + n_i). \tag{14}$$

Define the *weight space of expert functions* as:

$$\Theta = \prod_{i=1}^{L} \left( \mathbb{R}^{n_{i-1} \times n_i} \times \mathbb{R}^{n_i} \right) = \mathbb{R}^e. \tag{15}$$

**Dense Mixture-of-Experts.** Given a positive integer $n$ representing the number of experts, a *Mixture-of-Experts with Dense gating* (MoE) is defined as the function: $\mathcal{D} \colon \mathbb{R}^d \to \mathbb{R}^d$ defined by

$$\mathcal{D}\left(x; \{W_i, b_i, \theta_i\}_{i=1}^{n}\right) = \sum_{i=1}^{n} \text{softmax}_i \left(\{W_i x + b_i\}_{i=1}^{n}\right) \cdot \mathcal{E}(x; \theta_i). \tag{16}$$

The function $\mathcal{D}$ is parameterized as $\mathcal{D}(x; \phi)$ where

$$\phi = \left((W_i, b_i), \theta_i\right)_{i=1,\ldots,n} \in \left((\mathbb{R}^D \times \mathbb{R}) \times \Theta\right)^n. \tag{17}$$

Denote the *weight space of a Mixture of $n$ Experts* as

$$\Phi(n) = \left((\mathbb{R}^D \times \mathbb{R}) \times \Theta\right)^n. \tag{18}$$

Varying the number of experts leads to a Mixture-of-Experts weight space that spans across expert sets of different sizes, denoted by

$$\Phi = \bigsqcup_{n=1}^{\infty} \Phi(n) = \bigsqcup_{n=1}^{\infty} \left((\mathbb{R}^D \times \mathbb{R}) \times \Theta\right)^n. \tag{19}$$

**Sparse Mixture-of-Experts.** Given a positive integer $k \le n$, the Top-$k$ map is defined by: for any vector $z = (z_1, \ldots, z_n) \in \mathbb{R}^n$,

$$\text{Top-}k(z) = \{i_1, \ldots, i_k\}, \tag{20}$$

where $i_1, \ldots, i_k$ are the indices corresponding to the $k$ largest components of $x$. In the event of ties, we select smaller indices first. Using this, a *Mixture-of-Experts with Sparse gating* (SMoE) is the function $\mathcal{S} \colon \mathbb{R}^d \to \mathbb{R}^d$ defined by

$$\mathcal{S}\left(x; \{W_i, b_i, \theta_i\}_{i=1}^{n}\right) = \sum_{i \in T(x)} \text{softmax}_i \left(\{W_i x + b_i\}_{i \in T(x)}\right) \cdot \mathcal{E}(x; \theta_i). \tag{21}$$

where

$$T(x) = T\left(x; \{W_i, b_i\}_{i=1}^n\right) = \text{Top-}k\left(\left(W_i x + b_i\right)_{i=1}^n\right). \tag{22}$$

The weight space for the Sparse Mixture-of-Experts is identical to that of the Dense Mixture-of-Experts, as the inclusion of the Top-$k$ operator does not introduce any additional trainable parameters. However, the SMoE function $\mathcal{S}$ is generally discontinuous, primarily due to the non-smooth nature of the Top-$k$ selection–particularly in scenarios where multiple experts share identical gating scores (i.e., ties occur). To circumvent this issue and ensure well-defined behavior, we restrict our analysis to a subset of the input space $\mathbb{R}^d$ where the top $K$ gating scores are uniquely determined without ambiguity. In particular, for $\{W_i, b_i\}_{i=1}^n \in (\mathbb{R}^d \times \mathbb{R})^n$, we define the subset of $\mathbb{R}^d$ such that the gating scores of all experts are pairwise distinct, i.e.,

$$\Omega\left(\{W_i, b_i\}_{i=1}^n\right) = \left\{x \in \mathbb{R}^d : \ (W_i x + b_i)_{i=1}^n \ \text{are pairwise distinct}\right\}. \tag{23}$$

We provide one result characterizing the structure of this domain, and another result describing the behavior of the SMoE map when it is restricted to inputs within this domain.

**Proposition A.1.** *If $\{W_i, b_i\}$ are pairwise distinct for $i = 1, \ldots, n$, then $\Omega\left(\{W_i, b_i\}_{i=1}^n\right)$ is an open and dense subset of $\mathbb{R}^d$.*

*Proof.* We have

$$\begin{aligned}
\Omega\left(\{W_i, b_i\}_{i=1}^n\right) &= \left\{x \in \mathbb{R}^d : \ W_i x + b_i \ \text{is pairwise distinct for all } i = 1, \ldots, n\right\} \\
&= \bigcap_{1 \le i < j \le n} \left\{x \in \mathbb{R}^d : \ W_i x + b_i \ne W_j x + b_j\right\} \\
&= \bigcap_{1 \le i < j \le n} \left(\mathbb{R}^d \setminus \left\{x \in \mathbb{R}^d : \ W_i x + b_i = W_j x + b_j\right\}\right).
\end{aligned} \tag{24}$$

Note that, the set

$$\left\{x \in \mathbb{R}^d : \ W_i x + b_i = W_j x + b_j\right\} = \left\{x \in \mathbb{R}^d : \ (W_i - W_j)\, x = b_j - b_i\right\}, \tag{25}$$

is either a hyperplane (when $W_i \ne W_j$) or empty (when $W_i = W_j$ but $b_i \ne b_j$). In both scenarios, its complement in $\mathbb{R}^d$ forms an open and dense subset. By Equation (24), and using the fact that a finite intersection of open and dense subsets in $\mathbb{R}^d$ remains open and dense, it follows that the set $\Omega\left(W_i, b_{i\,i=1}^n\right)$ is itself open and dense in $\mathbb{R}^d$. $\qquad\square$

**Proposition A.2.** *If $\{W_i, b_i\}$ are pairwise distinct for $i = 1, \ldots, n$, then the SMoE map $\mathcal{S}$, as given in Equation (21), is continuous over the domain $\Omega\left(\{W_i, b_i\}_{i=1}^n\right)$.*

*Proof.* Let $x \in \Omega\left(\{W_i, b_i\}_{i=1}^n\right)$. By the definition of this set, there exists an open neighborhood $U \subset \mathbb{R}^d$ containing $x$ such that $U \subset \Omega\left(\{W_i, b_i\}_{i=1}^n\right)$ and

$$\text{Top-}k\left((W_i x + b_i)_{i=1}^n\right) = \text{Top-}k\left((W_i y + b_i)_{i=1}^n\right) \tag{26}$$

for all $y \in U$. This condition ensures that the sparse gating mechanism defined in Equation (21) remains fixed within $U$, and consequently, the SMoE map is continuous on the domain $\Omega\left(\{W_i, b_i\}_{i=1}^n\right)$. $\qquad\square$

Propositions A.1 and A.2 serve as essential building blocks in the proof of Theorem C.4.

## A.2 Group Action on Weight Spaces of Mixture-of-Experts

We define the group $G = G(n)$ as

$$G(n) = \left(\mathbb{R}^d \times \mathbb{R}\right) \times \mathrm{S}_n, \tag{27}$$

which is the direct product of the additive groups $\mathbb{R}^d$ and $\mathbb{R}$, and the symmetric group $\mathrm{S}_n$ on $n$ elements. Each element $g \in G(n)$ can be written in the form

$$g = (c_W, c_b, \tau), \quad \text{where } c_W \in \mathbb{R}^d, \ c_b \in \mathbb{R}, \text{ and } \tau \in \mathrm{S}_n. \tag{28}$$

The group $G(n)$ acts on the weight space $\Phi(n)$ as follows: for $g = (c_W, c_b, \tau) \in G(n)$ and $\phi \in \Phi(n)$ given as in Equation (17), the action is defined by

$$g\phi := \left(W_{\tau(i)} + c_W,\ b_{\tau(i)} + c_b,\ \theta_{\tau(i)}\right)_{i=1,\dots,n} \in \Phi(n). \tag{29}$$

This group action preserves the functionality of both the MoE and SMoE maps. The invariance arises from two key properties: the permutation invariance of the summation operator, and the translation invariance of the softmax function. We begin by establishing a result that characterizes the invariance of MoE maps under this group action.

**Proposition A.3** (Weight space invariance of Mixture-of-Experts). *The MoE map $\mathcal{D}$ is $G(n)$-invariance under the action of $G(n)$ on its weight space, i.e.,*

$$\mathcal{D}(\cdot; \phi) = \mathcal{D}(\cdot; g\phi). \tag{30}$$

*Proof.* Given $g = (c_W, c_b, \tau) \in G(n)$. For all $x \in \mathbb{R}^d$, we have

$$
\begin{aligned}
\mathcal{D}(x; g\theta) &= \sum_{i=1}^{n} \mathrm{softmax}_i \left( \left\{ \left(W_{\tau(i)} + c_W\right) x + \left(b_{\tau(i)} + c_b\right) \right\}_{i=1}^{n} \right) \cdot \mathcal{E}\left(x; \theta_{\tau(i)}\right) \\
&= \sum_{i=1}^{n} \mathrm{softmax}_i \left( \left\{ W_{\tau(i)} x + b_{\tau(i)} \right\}_{i=1}^{n} \right) \cdot \mathcal{E}\left(x; \theta_{\tau(i)}\right) \\
&= \sum_{i=1}^{n} \mathrm{softmax}_i \left( \left\{ W_i x + b_i \right\}_{i=1}^{n} \right) \mathcal{E}\left(x; \theta_i\right) \\
&= \mathcal{D}(x; \theta).
\end{aligned}
\tag{31}
$$

Thus, the proposition is proven. $\square$

The analysis of the SMoE architecture requires additional assumptions due to the inherent discontinuity of the Top-$k$ selection operator. In what follows, we show that the SMoE map, when restricted to an appropriate subset of its domain, remains invariant under the group action of $G(n)$.

**Proposition A.4** (Weight space invariance of Sparse Mixture-of-Experts). *Given the SMoE map as defined in Equation (21), assume that the pairs $\{W_i, b_i\}$ are pairwise distinct for $i = 1, \dots, n$. Then, the set $\Omega\left(\{W_i, b_i\}_{i=1}^{n}\right)$ is invariant under the group action of $G(n)$, i.e.,*

$$\Omega\left(\{W_i, b_i\}_{i=1}^{n}\right) = \Omega\left(g\{W_i, b_i\}_{i=1}^{n}\right). \tag{32}$$

*Moreover, the SMoE map, restricted to*

$$\Omega\left(\{W_i, b_i\}_{i=1}^{n}\right), \tag{33}$$

*is $G(n)$-invariance under the action of $G(n)$ on its weight space, i.e.,*

$$\mathcal{S}(\cdot; \theta) = \mathcal{S}(\cdot; g\theta) \quad \text{on} \quad \Omega\left(\{W_i, b_i\}_{i=1}^{n}\right). \tag{34}$$

*Proof.* Let $g = (c_W, c_b, \tau) \in G(n)$. We first verify that the group action preserves the set $\Omega\left(\{W_i, b_i\}_{i=1}^{n}\right)$. Indeed,

$$
\begin{aligned}
&\Omega\left(\{W_i, b_i\}_{i=1}^{n}\right) \\
&= \left\{ x \in \mathbb{R}^d :\ W_i x + b_i \ \text{is pairwise distinct for all } i = 1, \dots, n \right\} \\
&= \left\{ x \in \mathbb{R}^d :\ \left(W_{\tau(i)} + c_W\right) x + \left(b_{\tau(i)} + c_b\right) \ \text{is pairwise distinct for all } i = 1, \dots, n \right\} \\
&= \Omega\left(g\{W_i, b_i\}_{i=1}^{n}\right).
\end{aligned}
\tag{35}
$$

Let

$$gT(x) = \text{Top-}k\left(\left(\left(W_{\tau(i)} + c_W\right) x + \left(b_{\tau(i)} + c_b\right)\right)_{i=1}^{n}\right). \tag{36}$$

For all $x \in \Omega\left(\{W_i, b_i\}_{i=1}^{n}\right)$, we have $gT(x) = \tau(T(x))$. The desired invariance result for the SMoE map then follows by applying the same reasoning as in Proposition A.3. $\square$

**Remark A.5.** While the group action on Mixture-of-Experts (MoE) models is formally introduced in Equation (22), it does not fully capture all symmetries inherent to these architectures. In particular, each expert network possesses internal neuron permutations that preserve its functional behavior–a well-documented property in the literature on neural networks [7, 23, 29, 9, 60, 11, 62, 13, 77, 81, 88, 37, 15, 27, 50, 5, 6]. Nonetheless, since the distinguishing feature of MoE models is their input-dependent gating function, our focus centers on the symmetries associated with the gate itself, treating expert-level invariances as part of a well-established theoretical foundation.

## B    Results on Mixture-of-Experts with Dense Gating

### B.1    A Structural Property of Neural Networks with ReLU Activation

A *polytope* is a geometric entity bounded by flat surfaces, which can be either finite (bounded) or infinite (unbounded) in extent. We define the notion of *local affineness* as follows: a function $f : \mathbb{R}^d \to \mathbb{R}^D$ is said to be locally affine if there exists a partition of $\mathbb{R}^d$ into a finite set of polytopes such that on each polytope, the function $f$ agrees with an affine transformation from $\mathbb{R}^d$ to $\mathbb{R}^D$.

**Remark B.1.** While the term *local affineness* may take on different interpretations in other contexts, its usage here is unambiguous within the scope of this work.

We examine the local affineness property in ReLU neural networks. Consider a feedforward neural network $f : \mathbb{R}^{n_0} \to \mathbb{R}^{n_L}$ constructed from affine maps interleaved with ReLU activations, given by

$$f = f_L \circ \sigma \circ f_{L-1} \circ \cdots \circ \sigma \circ f_1,$$

where each map $f_i : \mathbb{R}^{d_{i-1}} \to \mathbb{R}^{d_i}$ is affine and has the form $f_i(x) = W_i x + b_i$, and $\sigma$ denotes the ReLU function applied elementwise.

The combination of these affine transformations and ReLU activations partitions the domain $\mathbb{R}^{d_0}$ into finitely many convex polytopes. On each such region, the activation status of the ReLU units–whether they transmit their input or suppress it to zero–remains fixed. Consequently, the network reduces to a simpler form in which each ReLU behaves either as the identity or the zero function. Given that both ReLU and affine maps are piecewise linear and closed under composition, the overall function $f$ is affine within each activation region.

Hence, the network satisfies the following property:

$$f(x) = A_i x + b_i, \quad \text{for all } x \in P_i, \tag{37}$$

where the domain is partitioned into polytopes $\{P_i\}_{i=1}^m$, and $A_i$, $b_i$ define the affine transformation active within region $P_i$.

Let $\partial P_i$ denote the boundary of the polytope $P_i$. Then the set

$$\mathbb{R}^{d_0} \setminus \bigcup_{i=1}^m \partial P_i \tag{38}$$

is open and dense in $\mathbb{R}^{d_0}$; in other words, the union of the interiors of the polytopes covers a set that is both open and dense.

Now consider a finite collection of ReLU networks $\{f^{(k)}\}_{k=1}^n$. Since the intersection of finitely many open dense subsets is itself open and dense, there exists a subset $\Omega \subset \mathbb{R}^{d_0}$ such that, for every point $x \in \Omega$, there is a neighborhood around $x$ on which all functions $f^{(k)}$ act as affine maps.

### B.2    A Technical Lemma on Holomorphic Functions in $\mathbb{C}^n$

A function $f : \mathbb{C}^n \to \mathbb{C}$ is said to be *holomorphic on* $\mathbb{C}^n$ if it is complex differentiable at every point in $\mathbb{C}^n$. A function is called *meromorphic on* $\mathbb{C}^n$ if it can be locally written as a quotient of two holomorphic functions, where the denominator is not identically zero. The collection of all holomorphic functions on $\mathbb{C}^n$ forms an integral domain, denoted by $\mathcal{H}$, while the set of meromorphic functions on $\mathbb{C}^n$ forms a field, denoted by $\mathcal{F}$. In particular, $\mathcal{F}$ is the field of fractions of the integral domain $\mathcal{H}$. Let $\mathbb{C}[x] = \mathbb{C}[x_1, \ldots, x_n]$ denote the polynomial ring in $n$ complex variables, and let $\mathbb{C}(x) = \mathbb{C}(x_1, \ldots, x_n)$ denote the corresponding field of rational functions. Then $\mathbb{C}[x]$ is a subring of $\mathcal{H}$ and remains an integral domain, while $\mathbb{C}(x)$ is a subfield of $\mathcal{F}$ and represents the field of fractions of $\mathbb{C}[x]$.

**Remark B.2.** For any polynomial $p \in \mathbb{C}[x]$, the exponential $e^p$ defines a holomorphic function on $\mathbb{C}^n$, i.e. $e^p \in \mathcal{D}$.

Since $\mathbb{C}(x) \subset \mathcal{F}$, we can view $\mathcal{F}$ as a vector space over $\mathbb{C}(x)$. The following lemma addresses the linear independence of exponential functions of polynomials in this vector space setting.

**Lemma B.3.** *Let $p_1, \ldots, p_N$ be polynomials in $\mathbb{C}[x]$ such that $p_i - p_j$ is nonconstant whenever $i \neq j$. Then the functions $e^{p_1}, \ldots, e^{p_N}$, viewed as elements of $\mathcal{F}$, are linearly independent over $\mathbb{C}(x)$.*

*Proof.* We proceed by induction on $N$. The base case $N = 1$ is immediate, since $e^p \neq 0$ for any polynomial $p$. Assume the result holds for any smaller number of exponentials. Consider polynomials $r_1, \ldots, r_N \in \mathbb{C}[x]$ such that

$$r_1 \cdot e^{p_1} + \cdots + r_N \cdot e^{p_N} = 0. \tag{39}$$

We aim to prove that all $r_i = 0$. Suppose not; then at least one $r_i$ is nonzero. Without loss of generality, assume $r_N \neq 0$. Dividing through by $r_N \cdot e^{p_N}$ gives

$$\frac{r_1}{r_N} \cdot e^{p_1 - p_N} + \cdots + \frac{r_{N-1}}{r_N} \cdot e^{p_{N-1} - p_N} + 1 = 0. \tag{40}$$

Differentiating both sides with respect to $x_i$ for each $i = 1, \ldots, n$, we obtain:

$$\sum_{j=1}^{N-1} \left( \frac{\partial}{\partial x_i} \left( \frac{r_j}{r_N} \right) + \frac{r_j}{r_N} \cdot \frac{\partial}{\partial x_i} (p_j - p_N) \right) \cdot e^{p_j - p_N} = 0. \tag{41}$$

Note that each term

$$\frac{\partial}{\partial x_i} \left( \frac{r_j}{r_N} \right) + \frac{r_j}{r_N} \cdot \frac{\partial}{\partial x_i} (p_j - p_N) \tag{42}$$

belongs to $\mathbb{C}(x)$. Now, the polynomials $p_1 - p_N, \ldots, p_{N-1} - p_N$ are pairwise distinct and nonconstant. Thus, by the induction hypothesis, the exponentials $e^{p_j - p_N}$ are linearly independent over $\mathbb{C}(x)$ for $j = 1, \ldots, N - 1$. Therefore, from equation (41), we must have

$$\frac{\partial}{\partial x_i} \left( \frac{r_j}{r_N} \right) + \frac{r_j}{r_N} \cdot \frac{\partial}{\partial x_i} (p_j - p_N) = 0, \tag{43}$$

for all $i = 1, \ldots, n$ and $j = 1, \ldots, N - 1$, which implies

$$\frac{\partial}{\partial x_i} \left( \frac{r_j}{r_N} \cdot e^{p_j - p_N} \right) = 0. \tag{44}$$

Hence, for each $j = 1, \ldots, N - 1$, the function

$$\frac{r_j}{r_N} \cdot e^{p_j - p_N} = c_j \in \mathbb{C} \tag{45}$$

must be constant. If $c_j \neq 0$, then both $r_j \neq 0$ and $e^{p_j - p_N} = \frac{c_j r_N}{r_j}$ must hold. But this forces $e^{p_j - p_N}$ to be constant, which contradicts the assumption that $p_j - p_N$ is nonconstant. Thus, $c_j = 0$, which implies $r_j = 0$ for all $j = 1, \ldots, N - 1$. Plugging back into equation (40) gives a contradiction, as $1 \neq 0$. Therefore, all $r_i = 0$, completing the proof. $\qquad \square$

**Remark B.4.** This lemma plays a key role and will be applied repeatedly in the proofs of Theorem B.5 and Theorem C.4.

## B.3   Functional Equivalence in Mixture-of-Experts with Dense Gating

The following result establishes the equivalence between two sets of weights that define the same MoE map. Certain assumptions are introduced for technical reasons, and their justification is provided in Remark B.6.

**Theorem B.5** (Functional equivalence in Mixture-of-Experts with Dense Gating)**.** *Suppose $\phi, \phi'$ define the same MoE function, i.e. $\mathcal{D}(\cdot; \phi) = \mathcal{D}(\cdot; \phi')$. Assume that the following conditions hold:*

  1. *Both $\{\mathcal{E}(\cdot; \theta_i)\}_{i=1}^{n}$ and $\{\mathcal{E}(\cdot; \theta_i')\}_{i=1}^{n'}$ consist of pairwise distinct functions;*

2. Both $\{W_i - W_j\}_{1 \le i < j \le n}$ and $\{W'_i - W'_j\}_{1 \le i < j < n'}$ consist of pairwise distinct vector of $\mathbb{R}^d$.

Then $n = n'$, and there exists $g = (c_W, c_b, \tau) \in G(n)$ such that for all $i = 1, \dots, n$, we have $W'_i = W_{\tau(i)} + c_W$, $b'_i = b_{\tau(i)} + c_b$, and $\mathcal{E}(\cdot; \theta'_i) = \mathcal{E}(\cdot; \theta_{\tau(i)})$ on $\mathbb{R}^d$.

*Proof.* To enhance clarity, we begin by outlining the main steps of the proof at a high level:

1. We first expand the equality $\mathcal{D}(\cdot; \phi) = \mathcal{D}(\cdot; \phi')$ and introduce simplified notation to streamline the subsequent derivations.

2. We then observe that each expert function can be locally characterized as an affine map.

3. Next, we prove that $n = n'$ and establish the existence of a permutation $\tau$ and a transformation $c_W$ with the required properties.

4. We proceed to verify the equivalence between the two sets of experts.

5. Finally, we show that a transformation $c_b$ satisfying the necessary conditions can be constructed.

We now proceed with the detailed derivations and justifications corresponding to each of the five steps above.

**Step 1.**   Given that $\mathcal{D}(\cdot; \phi) = \mathcal{D}(\cdot; \phi')$, we have

$$\sum_{i=1}^{n} \text{softmax}_i \left( \{W_i x + b_i\}_{i=1}^{n} \right) \cdot \mathcal{E}(x; \theta_i) = \sum_{i=1}^{n'} \text{softmax}_i \left( \{W'_i x + b'_i\}_{i=1}^{n'} \right) \cdot \mathcal{E}(x; \theta'_i), \quad (46)$$

for all $x \in \mathbb{R}^d$. Define

$$\mathcal{E}_i(\cdot) = \mathcal{E}(\cdot; \theta_i), \quad \text{and} \quad \mathcal{E}'_i(\cdot) = \mathcal{E}(\cdot; \theta'_i). \quad (47)$$

By expanding the softmax terms in Equation (46), we obtain

$$\sum_{i=1}^{n} \frac{e^{W_i x + b_i}}{\sum_{j=1}^{n} e^{W_j x + b_j}} \cdot \mathcal{E}_i(x) = \sum_{i=1}^{n'} \frac{e^{W'_i x + b'_i}}{\sum_{j=1}^{n'} e^{W'_j x + b'_j}} \cdot \mathcal{E}'_i(x). \quad (48)$$

Multiplying both sides by the respective denominators yields

$$\left( \sum_{j=1}^{n'} e^{W'_j x + b'_j} \right) \cdot \left( \sum_{i=1}^{n} e^{W_i x + b_i} \cdot \mathcal{E}_i(x) \right) = \left( \sum_{j=1}^{n} e^{W_j x + b_j} \right) \cdot \left( \sum_{i=1}^{n'} e^{W'_i x + b'_i} \cdot \mathcal{E}'_i(x) \right), \quad (49)$$

which can be rewritten as

$$\sum_{i=1}^{n} \sum_{j=1}^{n'} e^{(W_i + W'_j)x + (b_i + b'_j)} \cdot \left( \mathcal{E}_i(x) - \mathcal{E}'_j(x) \right) = 0. \quad (50)$$

**Step 2.**   Since each function $\mathcal{E}_i$ and $\mathcal{E}'_j$ is locally affine, it follows from the result in Appendix B.1 that there exists a dense open subset $\Omega \subset \mathbb{R}^d$ such that: for any point $a \in \Omega$, one can find an open neighborhood $U \subset \Omega$ containing $a$ on which all functions $\mathcal{E}_i$ and $\mathcal{E}'_j$ are affine. Consequently, each of these functions agrees with a polynomial on $U$. That is, there exists a family of open sets $\{U_k\}_{k \in I}$ covering $\Omega$, so that

$$\Omega = \bigcup_{k \in I} U_k, \quad (51)$$

and for each set $U = U_k$ in this cover, there exist polynomials $p_{U,i}, p'_{U,j} \in \mathbb{R}[x]$ such that

$$\mathcal{E}_i(x) = p_{U,i}(x), \quad \text{and} \quad \mathcal{E}'_j(x) = p'_{U,j}(x) \quad \text{for all } x \in U. \quad (52)$$

Substituting into Equation (50) yields:

$$\sum_{i=1}^{n}\sum_{j=1}^{n'} e^{(W_i+W_j')x+(b_i+b_j')} \cdot \left(p_{U,i}(x) - p_{U,j}'(x)\right) = 0 \quad \text{for all } x \in U. \tag{53}$$

Observe that the expression on the left-hand side above defines a holomorphic function. By the Identity Theorem for Holomorphic Functions (see [1, 66, 16, 74]), we conclude that:

$$\sum_{i=1}^{n}\sum_{j=1}^{n'} e^{(W_i+W_j')x+(b_i+b_j')} \cdot \left(p_{U,i}(x) - p_{U,j}'(x)\right) = 0 \quad \text{for all } x \in \mathbb{C}^d. \tag{54}$$

**Step 3.** According to Assumption 2, the sets $\{W_i\}_{i=1}^n$ and $\{W_j'\}_{j=1}^{n'}$ are composed of mutually distinct elements. As a result, there exists a direction

$$\alpha \in \mathbb{S}^{d-1} = \{x \in \mathbb{R}^d \,:\, \|x\|_2 = 1\}, \tag{55}$$

such that the projected values $\{W_i\alpha\}_{i=1}^n$ and $\{W_j'\alpha\}_{j=1}^{n'}$ are comprised of $n$ and $n'$ distinct real numbers, respectively. Without loss of generality, we may relabel the indices so that:

$$W_1\alpha < W_2\alpha < \ldots < W_n\alpha \quad \text{and} \quad W_1'\alpha < W_2'\alpha < \ldots < W_{n'}'\alpha. \tag{56}$$

Furthermore, observe that the problem setting and the preceding equations are invariant under translations of the form $W_j' \mapsto W_j' + c_W$ for a constant vector $c_W \in \mathbb{R}^d$. Hence, we can assume without loss of generality that $W_1 = W_1'$. Under this assumption, we aim to demonstrate that $n = n'$ and $W_i = W_i'$ for all $i = 1, \ldots, n$. Toward that goal, we begin by showing that $W_i = W_i'$ for each $i = 1, \ldots, \min\{n, n'\}$ using mathematical induction.

*Base case.* By assumption, we have $W_1 = W_1'$, so the base case holds trivially.

*Auxiliary step for induction.* For every index pair $(i, j) \neq (1, 1)$, we have the inequality:

$$W_1\alpha + W_1'\alpha < W_i\alpha + W_j'\alpha. \tag{57}$$

Thus, the sum $W_1 + W_1'$ differs from $W_i + W_j'$ for all $(i, j) \neq (1, 1)$. Applying Equation (54) in conjunction with Lemma B.3, we conclude that

$$p_{U,1} = p_{U,1}'. \tag{58}$$

*Inductive step.* Assume that $W_i = W_i'$ holds for all $1 \leq i < k$, where $k$ is an integer such that $1 < k \leq \min\{n, n'\}$. Suppose, for contradiction, that $W_k \neq W_k'$. We analyze the terms $W_1 + W_k'$ and $W_k + W_1'$. Under our assumption, these two quantities must differ. Without loss of generality, we may assume that

$$W_1\alpha + W_k'\alpha \leq W_k\alpha + W_1'\alpha. \tag{59}$$

- For all index pairs $(i, j)$ with $i \geq k$, we have

$$W_1\alpha + W_k'\alpha \leq W_k\alpha + W_1'\alpha \leq W_i\alpha + W_j'\alpha. \tag{60}$$

  Equality holds if and only if $(i, j) = (k, 1)$. Moreover, since $W_1 + W_k'$ and $W_k + W_1'$ are distinct, it follows that $W_1 + W_k'$ differs from $W_i + W_j'$ for all $(i, j)$ with $i \geq k$.

- For all $(i, j)$ with $j \geq k$, we have

$$W_1\alpha + W_k'\alpha \leq W_i\alpha + W_j'\alpha. \tag{61}$$

  Equality occurs only when $(i, j) = (1, k)$. Therefore, $W_1 + W_k'$ is distinct from $W_i + W_j'$ for all $(i, j) \neq (1, k)$ with $j \geq k$.

- For all $(i, j)$ such that $i, j < k$, we claim that $W_1 + W_k'$ does not equal $W_i + W_j'$. Suppose, for contradiction, that

$$W_1 + W_k' = W_i + W_j' \tag{62}$$

for some pair $(i, j)$ with $i, j < k$. Then by the induction hypothesis,

$$W_1' + W_k' = W_i' + W_j'. \tag{63}$$

Rearranging terms, we obtain

$$W_1' - W_j' = W_i' - W_k', \tag{64}$$

which contradicts the assumption that all such differences are pairwise distinct, since $(1, j) \neq (i, k)$.

These observations collectively show that $W_1 + W_k'$ differs from $W_i + W_j'$ for all $(i, j) \neq (1, k)$. Applying Equation (54) together with Lemma B.3, we conclude that

$$p_{U,1} = p_{U,k}'. \tag{65}$$

Additionally, from Equation (58), we already have

$$p_{U,1}' = p_{U,k}'. \tag{66}$$

This implies that $\mathcal{E}_1' = \mathcal{E}_k'$ on $U$. Since this holds for every open set $U$ in the covering $\{U_k\}_{k \in I}$, it follows that $\mathcal{E}_1' = \mathcal{E}_k'$ on $\Omega$. Because $\Omega$ is dense in $\mathbb{R}^d$ and each $\mathcal{E}_j'$ is continuous, we conclude that $\mathcal{E}_1' = \mathcal{E}_k'$ on all of $\mathbb{R}^d$. This contradicts the assumption that the $\mathcal{E}_j'$ functions are pairwise distinct. Therefore, our initial assumption must be false, and we conclude that $W_1 + W_k' = W_k + W_1'$, which implies $W_k = W_k'$.

*Conclusion.* By induction, we have established that $W_i = W_i'$ for all $i = 1, \ldots, \min\{n, n'\}$. It remains to show that $n = n'$. Suppose, for contradiction, that $n < n'$. Consider the sum $W_1 + W_{n'}'$. We claim that this quantity is distinct from every $W_i + W_j'$ with $(i, j) \neq (1, n')$. Assume otherwise, that

$$W_1 + W_{n'}' = W_i + W_j' \tag{67}$$

for some pair $(i, j) \neq (1, n')$. Using the inductive assumption that $W_i = W_i'$ for all $i \leq n$, we deduce

$$W_1' + W_{n'}' = W_i' + W_j', \tag{68}$$

which implies

$$W_1' - W_j' = W_i' - W_{n'}'. \tag{69}$$

This leads to a contradiction, as it violates the assumption that the differences $W_i' - W_j'$ are pairwise distinct. Hence, $W_1 + W_{n'}'$ must be distinct from all other sums $W_i + W_j'$ where $(i, j) \neq (1, n')$. Then, by Equation (54) and Lemma B.3, it follows that

$$p_{U,1} = p_{U,n'}'. \tag{70}$$

In addition, from Equation (58), we already know that

$$p_{U,1}' = p_{U,n'}'. \tag{71}$$

Consequently, we obtain $\mathcal{E}_1' = \mathcal{E}_{n'}'$ on $U$. Since this holds on every open set $U$ in the covering $\{U_k\}_{k \in I}$, it extends to $\Omega$, and by continuity, to all of $\mathbb{R}^d$. This contradicts the assumption that the expert functions $\mathcal{E}_j'$ are pairwise distinct. Therefore, our assumption must be false, and we conclude that $n = n'$. Finally, the reordering of indices and the translation applied to the set $\{W_j'\}_{j=1}^{n'}$ throughout the argument confirm the existence of a permutation $\tau \in S_n$ and a translation vector $c_W \in \mathbb{R}^d$.

**Step 4.** We now establish that $\mathcal{E}_i = \mathcal{E}_i'$ on $\mathbb{R}^d$ for each $i = 1, \ldots, n$. From **Step 3**, we already have that $n = n'$ and $W_i = W_i'$ for all indices in this range. Consider any pair $(i, j)$. If it holds that $W_i + W_j' = W_{i'} + W_{j'}'$, then $(i', j')$ must be either $(i, j)$ or $(j, i)$. In particular, this implies that $W_i + W_i'$ is distinct from $W_j + W_k'$ for all $(j, k) \neq (i, i)$. Invoking Equation (54) together with Lemma B.3, we obtain

$$p_{U,i} = p_{U,i}'. \tag{72}$$

This argument parallels that used in **Step 3**, and applying the same reasoning, we conclude that $\mathcal{E}_i = \mathcal{E}_i'$ on $\mathbb{R}^d$. Since this holds for all $i = 1, \ldots, n$, the result follows.

**Step 5.** It remains to prove the existence of a constant $c_b \in \mathbb{R}$ such that

$$b'_i = b_i + c_b \quad \text{for all } i = 1, \ldots, n. \tag{73}$$

Recall from **Step 4** that if $W_i + W'_j = W_{i'} + W'_{j'}$, then it must be that $(i', j') = (i, j)$ or $(j, i)$. Using this structural property, along with Equation (50), Lemma B.3, and the identity $\mathcal{E}_i = \mathcal{E}'_i$ established in **Step 4**, we derive the following equality:

$$e^{\left(W_i+W'_j\right)x+\left(b_i+b'_j\right)} \cdot \left(\mathcal{E}_i(x) - \mathcal{E}_j(x)\right) + e^{\left(W_j+W'_i\right)x+\left(b_j+b'_i\right)} \cdot \left(\mathcal{E}_j(x) - \mathcal{E}_i(x)\right) = 0, \tag{74}$$

valid for all index pairs $(i, j)$. Since $\mathcal{E}_i \neq \mathcal{E}_j$ whenever $i \neq j$, there exists a point $x_0 \in \mathbb{R}^d$ for which $\mathcal{E}_i(x_0) \neq \mathcal{E}_j(x_0)$. Plugging $x = x_0$ into Equation (74) and simplifying by factoring out the common nonzero difference, we find:

$$e^{b_i+b'_j} = e^{b_j+b'_i}, \tag{75}$$

which implies

$$b_i + b'_j = b_j + b'_i, \tag{76}$$

and hence

$$b_i - b'_i = b_j - b'_j. \tag{77}$$

This shows that the difference $b_i - b'_i$ remains constant for all $i$. Letting $c_b := b'_1 - b_1$, we obtain

$$b'_i = b_i + c_b \quad \text{for all } i = 1, \ldots, n. \tag{78}$$

This completes the proof of Theorem B.5.

$\square$

**Remark B.6** (Rationale behind the assumptions in Theorem B.5)**.** In modeling architectures, it is important that the symmetry group arises from the structure of the model itself rather than from specific, possibly degenerate, parameter choices. That is, the symmetry group should act globally and consistently across the entire weight space. This requirement motivates the following conditions in Theorem B.5:

1. Both $\{\mathcal{E}(\cdot; \theta_i)\}_{i=1}^{n}$ and $\{\mathcal{E}(\cdot; \theta'_i)\}_{i=1}^{n'}$ consist of pairwise distinct functions;

2. Both $\{W_i - W_j\}_{1 \leq i < j \leq n}$ and $\{W'_i - W'_j\}_{1 \leq i < j < n'}$ consist of pairwise distinct vector of $\mathbb{R}^d$.

We now elaborate on the purpose of these assumptions.

*Assumption 1.* If this condition is violated–for instance, if two experts compute the same function and are assigned identical gating values–then permuting these experts leaves the model output unchanged. Such permutations introduce additional, non-essential elements into the symmetry group, which we refer to as spurious symmetries. These do not correspond to genuine structural invariances but rather arise from degenerate parameter settings that represent singular points in the model's parameter space.

*Assumption 2.* Assumption 2 addresses a more nuanced issue: it rules out cases where linear dependencies among gating weight vectors may cause multiple experts to exhibit indistinguishable gating behavior. Although such situations may not be as immediately intuitive as those excluded by Assumption 1, they too can artificially enlarge the symmetry group beyond its intended form. To illustrate this, we present an explicit example. Let $d = 1$ and $n = n' = 3$, and consider parameter configurations $\phi$ and $\phi'$ such that:

- $W_1 = W'_1 = -1$, $W_2 = W'_2 = 0$, and $W_3 = W'_3 = 1$,

- The parameters $\theta_1, \theta_2, \theta_3, \theta'_1, \theta'_2, \theta'_3$ are chosen so that all six experts are constant functions. Let us define:

$$\begin{aligned}
\mathcal{E}(\cdot; \theta_1) &= A_1, & \mathcal{E}(\cdot; \theta'_1) &= A_2, \\
\mathcal{E}(\cdot; \theta_2) &= B_1, & \mathcal{E}(\cdot; \theta'_2) &= B_2, \\
\mathcal{E}(\cdot; \theta_3) &= C_1, & \mathcal{E}(\cdot; \theta'_3) &= C_2.
\end{aligned} \tag{79}$$

We now select the biases $b_i$, $b_i'$ and constants $A_j, B_j, C_j$ such that the models satisfy $\mathcal{D}(\cdot;\phi) = \mathcal{D}(\cdot;\phi')$, even though there exists no transformation as described in Theorem B.5 that maps $\phi$ to $\phi'$. Our goal is to ensure that

$$
\frac{e^{-x+b_1}}{e^{-x+b_1} + e^{b_2} + e^{x+b_3}} \cdot A_1 + \frac{e^{b_2}}{e^{-x+b_1} + e^{b_2} + e^{x+b_3}} \cdot B_1 + \frac{e^{x+b_3}}{e^{-x+b_1} + e^{b_2} + e^{x+b_3}} \cdot C_1
$$
$$
= \frac{e^{-x+b_1'}}{e^{-x+b_1'} + e^{b_2'} + e^{x+b_3'}} \cdot A_2 + \frac{e^{b_2'}}{e^{-x+b_1'} + e^{b_2'} + e^{x+b_3'}} \cdot B_2 + \frac{e^{x+b_3'}}{e^{-x+b_1'} + e^{b_2'} + e^{x+b_3'}} \cdot C_2. \quad (80)
$$

For convenience, we introduce the following shorthand:

$$
\begin{aligned}
e^{b_1} &= X_1, & e^{b_1'} &= X_2, \\
e^{b_2} &= Y_1, & e^{b_2'} &= Y_2, \\
e^{b_3} &= Z_1, & e^{b_3'} &= Z_2.
\end{aligned} \quad (81)
$$

With this notation, Equation (80) becomes:

$$
\frac{e^{-x}X_1}{e^{-x}X_1 + Y_1 + e^x Z_1} \cdot A_1 + \frac{Y_1}{e^{-x}X_1 + Y_1 + e^x Z_1} \cdot B_1 + \frac{e^x Z_1}{e^{-x}X_1 + Y_1 + e^x Z_1} \cdot C_1
$$
$$
= \frac{e^{-x}X_2}{e^{-x}X_2 + Y_2 + e^x Z_2} \cdot A_2 + \frac{Y_2}{e^{-x}X_2 + Y_2 + e^x Z_2} \cdot B_2 + \frac{e^x Z_2}{e^{-x}X_2 + Y_2 + e^x Z_2} \cdot C_2, \quad (82)
$$

which is equivalent to:

$$
\left(e^{-x}X_1 A_1 + Y_1 B_1 + e^x Z_1 C_1\right)\left(e^{-x}X_2 + Y_2 + e^x Z_2\right)
$$
$$
= \left(e^{-x}X_2 A_2 + Y_2 B_2 + e^x Z_2 C_2\right)\left(e^{-x}X_1 + Y_1 + e^x Z_1\right). \quad (83)
$$

By equating the coefficients of $e^{-2x}, e^{-x}, 1, e^x, e^{2x}$, we derive the following system:

$$
\begin{aligned}
e^{-2x} &: & X_1 X_2 A_1 &= X_1 X_2 A_2, \\
e^{2x} &: & Z_1 Z_2 C_1 &= Z_1 Z_2 C_2, \\
e^x &: & Y_1 Z_2 B_1 + Z_1 Y_2 C_1 &= Y_1 Z_2 C_2 + Z_1 Y_2 B_2, \\
e^{-x} &: & Y_1 X_2 B_1 + X_1 Y_2 A_1 &= Y_1 X_2 A_2 + X_1 Y_2 B_2, \\
1 &: & X_1 Z_2 A_1 + Z_1 X_2 C_1 + Y_1 Y_2 B_1 &= X_1 Z_2 C_2 + Z_1 X_2 A_2 + Y_1 Y_2 B_2.
\end{aligned} \quad (84)
$$

Setting $A_1 = A_2 = A$ and $C_1 = C_2 = C$ satisfies the equations involving $e^{-2x}$ and $e^{2x}$ automatically. Removing those, we simplify the system to:

$$
\begin{aligned}
e^x &: & Y_1 Z_2 B_1 + Z_1 Y_2 C &= Y_1 Z_2 C + Z_1 Y_2 B_2, \\
e^{-x} &: & Y_1 X_2 B_1 + X_1 Y_2 A &= Y_1 X_2 A + X_1 Y_2 B_2, \\
1 &: & X_1 Z_2 A + Z_1 X_2 C + Y_1 Y_2 B_1 &= X_1 Z_2 C + Z_1 X_2 A + Y_1 Y_2 B_2.
\end{aligned} \quad (85)
$$

Solving the equations for $A$ and $C$, assuming $Z_1 Y_2 \neq Y_1 Z_2$ and $X_1 Y_2 \neq Y_1 X_2$, gives:

$$
\begin{aligned}
A &= \frac{X_1 Y_2 B_2 - Y_1 X_2 B_1}{X_1 Y_2 - Y_1 X_2}, \\
C &= \frac{Z_1 Y_2 B_2 - Y_1 Z_2 B_1}{Z_1 Y_2 - Y_1 Z_2}.
\end{aligned} \quad (86)
$$

Substituting into the constant term equation in (85) gives:

$$
Y_1 Y_2 (B_1 - B_2) = (C - A)(X_1 Z_2 - Z_1 X_2). \quad (87)
$$

Now computing $A - C$:

$$
A - C = \frac{X_1 Y_2 B_2 - Y_1 X_2 B_1}{X_1 Y_2 - Y_1 X_2} - \frac{Z_1 Y_2 B_2 - Y_1 Z_2 B_1}{Z_1 Y_2 - Y_1 Z_2}
$$

$$= \frac{Y_1 Y_2 (B_1 - B_2)(X_1 Z_2 - Z_1 X_2)}{(X_1 Y_2 - Y_1 X_2)(Z_1 Y_2 - Y_1 Z_2)}. \tag{88}$$

Plugging into Equation (87), we find:

$$Y_1 Y_2 (B_1 - B_2) = -\frac{Y_1 Y_2 (B_1 - B_2)(X_1 Z_2 - Z_1 X_2)}{(X_1 Y_2 - Y_1 X_2)(Z_1 Y_2 - Y_1 Z_2)} (X_1 Z_2 - Z_1 X_2). \tag{89}$$

Assuming $B_1 \neq B_2$ and $Y_1 Y_2 \neq 0$, we divide both sides to obtain:

$$(X_1 Y_2 - Y_1 X_2)(Y_1 Z_2 - Z_1 Y_2) = (X_1 Z_2 - Z_1 X_2)^2. \tag{90}$$

While this equation can be solved in general, it suffices to exhibit a concrete solution. Consider:

$$\begin{aligned}
(X_1, X_2) &= (1, 2), \\
(Y_1, Y_2) &= (3, 5), \\
(Z_1, Z_2) &= (2, 3).
\end{aligned} \tag{91}$$

With these values, $B_1$ and $B_2$ may be freely chosen. These assignments define parameter configurations $\phi$ and $\phi'$ for which $\mathcal{D}(\cdot; \phi) = \mathcal{D}(\cdot; \phi')$, yet no transformation described in Theorem B.5 maps $\phi$ to $\phi'$.

## C   Results on Mixture-of-Experts with Sparse Gating

### C.1   Strongly Distinctness Property

The following definition formalizes the notion of *strong distinctness*, which will be used in the statement of Theorem C.4.

**Definition C.1** (Strongly distinct)**.** Two functions $f, g \colon X \to Y$ are said to be *strongly distinct* if the set $\{x \in X : f(x) \neq g(x)\}$ is dense in $X$.

**Example C.2.** We present several examples to illustrate the concept of strong distinctness.

- Two distinct polynomials on $\mathbb{R}^n$ or $\mathbb{C}^n$ are strongly distinct.

- Two distinct holomorphic functions are strongly distinct.

- In contrast, two distinct locally affine functions are not necessarily strongly distinct. For instance:

  – Let $f_1, f_2 \colon \mathbb{R} \to \mathbb{R}$ be defined as:

  $$f_1(x) = \begin{cases} 0 & \text{if } x < 0, \\ x & \text{if } x \geq 0, \end{cases} \qquad f_2(x) = 1. \tag{92}$$

  Then $f_1$ and $f_2$ are strongly distinct.

  – Let $g_1, g_2 \colon \mathbb{R} \to \mathbb{R}$ be defined as:

  $$g_1(x) = \begin{cases} 0 & \text{if } x < 0, \\ x & \text{if } x \geq 0, \end{cases} \qquad g_2(x) = 0. \tag{93}$$

  Here, $g_1$ and $g_2$ are distinct but not strongly distinct, since they coincide on $(-\infty, 0)$.

We now define a certain class of subsets of $\mathbb{R}^d$. Given

$$\{W_i, b_i\}_{i=1}^n \in \left( \mathbb{R}^d \times \mathbb{R} \right)^n, \tag{94}$$

we define the set

$$\Omega \left( \{W_i, b_i\}_{i=1}^n \right) := \left\{ x \in \mathbb{R}^d : \{W_i x + b_i\}_{i=1}^n \text{ are pairwise distinct} \right\}. \tag{95}$$

The following result provides a sufficient condition on the gating weights under which the Top-$k$ operator is capable of selecting any subset of $k$ experts.

**Proposition C.3.** *Suppose that the collection $\{W_i\}_{i=1}^n$ satisfies the condition that the set $\{W^{(G,i-1)} - W^{(G,i)}\}_{i=2}^n$ is linearly independent in $\mathbb{R}^d$. Then, for every subset $\mathcal{A} \subseteq \{1, \ldots, n\}$ of size $k$, there exists a point $x \in \Omega\left(\{W_i, b_i\}_{i=1}^n\right)$ such that*

$$\text{Top-}k\left((W_i x + b_i)_{i=1}^n\right) = \mathcal{A}. \tag{96}$$

*Proof.* Without loss of generality, assume $\mathcal{A} = \{1, \ldots, k\}$. To prove the existence of $x \in \Omega$ such that

$$\text{Top-}k\left((W_i x + b_i)_{i=1}^n\right) = \{1, \ldots, k\}, \tag{97}$$

it suffices to find $x \in \mathbb{R}^d$ such that

$$W_1 x + b_1 > W_2 x + b_2 > \cdots > W_n x + b_n. \tag{98}$$

We can strengthen this to require:

$$
\begin{aligned}
(W_1 x + b_1) - (W_2 x + b_2) &= 1, \\
(W_2 x + b_2) - (W_3 x + b_3) &= 1, \\
&\vdots \\
(W_{n-1} x + b_{n-1}) - (W_n x + b_n) &= 1.
\end{aligned}
\tag{99}
$$

This is equivalent to the following system of linear equations:

$$
\begin{aligned}
(W_1 - W_2) x &= 1 - (b_1 - b_2), \\
(W_2 - W_3) x &= 1 - (b_2 - b_3), \\
&\vdots \\
(W_{n-1} - W_n) x &= 1 - (b_{n-1} - b_n).
\end{aligned}
\tag{100}
$$

Since the vectors $\{W_{i-1} - W_i\}_{i=2}^n$ are linearly independent by assumption, this system has a unique solution. Hence, there exists $x \in \mathbb{R}^d$ satisfying Equation (100). $\qquad\square$

Proposition C.3 will be invoked in the proof of Theorem C.4. A justification for the linear independence assumption is provided in Remark C.8.

## C.2 Functional Equivalence in Mixture-of-Experts with Sparse Gating

We establish a functional equivalence theorem for the Sparse Mixture-of-Experts (SMoE) architecture, in parallel with the result for MoE given in Theorem B.5. However, our analysis is limited to the case $k > 1$, as the $k = 1$ setting leads to singularities that disrupt the general structure of equivalence. A detailed explanation for this exclusion is provided in Remark C.9.

**Theorem C.4** (Functional equivalence in Mixture-of-Experts with Sparse Gating). *Suppose $\phi, \phi'$ define the same SMoE function, i.e. $\mathcal{S}(\cdot; \phi) = \mathcal{S}(\cdot; \phi')$. Assume that the following conditions hold:*

1. *Both $\{\mathcal{E}(\cdot; \theta_i)\}_{i=1}^n$ and $\{\mathcal{E}(\cdot; \theta_i')\}_{i=1}^{n'}$ consist of pairwise strongly distinct functions;*

2. *Both $\{W_{i-1} - W_i\}_{i=2}^n$ and $\{W_{i-1}' - W_i'\}_{i=2}^{n'}$ are linear independent subsets of $\mathbb{R}^d$.*

*Then $n = n'$, and there exists $g = (c_W, c_b, \tau) \in G(n)$ such that for all $i = 1, \ldots, n$, we have $W_i' = W_{\tau(i)} + c_W$, $b_i' = b_{\tau(i)} + c_b$, and $\mathcal{E}(x; \theta_i') = \mathcal{E}(x; \theta_{\tau(i)})$ for all $x \in \Omega(\{W_i, b_i\}_{i=1}^n)$ such that $\tau(i) \in \text{Top-}k((W_i x + b_i)_{i=1}^n)$.*

Before presenting the proof of Theorem C.4, we begin with two remarks.

**Remark C.5.** Suppose $n = n'$ and there exist $\tau \in S_n$, $c_W \in \mathbb{R}^d$, and $c_b \in \mathbb{R}$ such that for every $i = 1, \ldots, n$, we have

$$W_i' = W_{\tau(i)} + c_W, \quad b_i' = b_{\tau(i)} + c_b. \tag{101}$$

Then the two sets $\Omega\left(\{W_i, b_i\}_{i=1}^n\right)$ and $\Omega\left(\{W_i', b_i'\}_{i=1}^n\right)$ coincide. Furthermore, for any $x$ in this set, the condition $\tau(i) \in \text{Top-}k((W_i x + b_i)_{i=1}^n)$ holds if and only if $i \in \text{Top-}k((W_i' x + b_i')_{i=1}^n)$.

**Remark C.6.** It is easy to verify that Assumption 2 in Theorem C.4 implies Assumption 2 in Theorem B.5.

*Proof.* To enhance clarity, we begin by outlining the main steps of the proof at a high level:

1. Formulate the identity $\mathcal{S}(\cdot; \phi) = \mathcal{S}(\cdot; \phi')$ explicitly and introduce simplified notation to streamline the presentation.

2. Partition the input space into regions where the Top-$k$ operator selects the same index set, and within which all expert functions are affine.

3. Demonstrate the desired equivalence for a fixed subset of experts. The core technique is to invoke the MoE equivalence result given in Theorem B.5.

4. Generalize the argument to establish the equivalence across all experts.

We now proceed with the detailed derivation and proofs for each of the outlined steps.

**Step 1.** Given that $\mathcal{S}(\cdot; \phi) = \mathcal{S}(\cdot; \phi')$, it follows that

$$
\sum_{i \in \text{Top-}k((W_i x + b_i)_{i=1}^n)} \text{softmax}_i \left( \{W_i x + b_i\}_{i \in \text{Top-}k((W_i x + b_i)_{i=1}^n)} \right) \cdot \mathcal{E}(x; \theta_i)
$$
$$
= \sum_{i \in \text{Top-}k((W_i' x + b_i')_{i=1}^{n'})} \text{softmax}_i \left( \{W_i' x + b_i'\}_{i \in \text{Top-}k((W_i' x + b_i')_{i=1}^{n'})} \right) \cdot \mathcal{E}(x; \theta_i'), \tag{102}
$$

for all $x \in \mathbb{R}^d$. To simplify notation, define

$$
\mathcal{E}_i(\cdot) = \mathcal{E}(\cdot; \theta_i), \quad \text{and} \quad \mathcal{E}_i'(\cdot) = \mathcal{E}(\cdot; \theta_i'). \tag{103}
$$

Using this notation, we can rewrite Equation (102) more compactly as:

$$
\sum_{i \in \text{Top-}k((W_i x + b_i)_{i=1}^n)} \text{softmax}_i \left( \{W_i x + b_i\}_{i \in \text{Top-}k((W_i x + b_i)_{i=1}^n)} \right) \cdot \mathcal{E}_i(x)
$$
$$
= \sum_{i \in \text{Top-}k((W_i' x + b_i')_{i=1}^{n'})} \text{softmax}_i \left( \{W_i' x + b_i'\}_{i \in \text{Top-}k((W_i' x + b_i')_{i=1}^{n'})} \right) \cdot \mathcal{E}_i'(x). \tag{104}
$$

**Step 2.** We begin by highlighting two key observations:

- Assumption 2 guarantees that the parameter pairs $\{W_i, b_i\}$ are pairwise distinct for $i = 1, \ldots, n$, and likewise, $\{W_i', b_i'\}$ are pairwise distinct for $i = 1, \ldots, n'$. By Proposition A.1, the set

$$
\Omega_1 = \Omega \left( \{W_i, b_i\}_{i=1}^n \right) \cap \Omega \left( \{W_i', b_i'\}_{i=1}^{n'} \right), \tag{105}
$$

is open and dense in $\mathbb{R}^d$. For any $x \in \Omega_1$, the values $\{W_i x + b_i\}$ and $\{W_i' x + b_i'\}$ are pairwise distinct. Moreover, for each $x \in \Omega_1$, there exists an open neighborhood around $x$ in which both Top-$k$ selections remain fixed.

- From the analysis in Appendix B.1, there exists a set $\Omega_2 \subset \mathbb{R}^d$, also open and dense, such that for every $x \in \Omega_2$, all expert functions $\mathcal{E}_i$ and $\mathcal{E}_j'$ are affine in a neighborhood of $x$.

Taking the intersection $\Omega = \Omega_1 \cap \Omega_2$, we obtain a set that is still open and dense. Within $\Omega$, both the Top-$k$ selections and the expert functions remain locally constant and affine, respectively. Consequently, there exists a collection of open sets $\{U_i\}_{i \in I}$ that cover $\Omega$, such that

$$
\Omega = \bigcup_{i \in I} U_i, \tag{106}
$$

and on each $U_i$, the expert functions $\mathcal{E}_i$ and $\mathcal{E}_j'$ are affine, and the Top-$k$ selections do not vary.

**Step 3.** Let $U$ be an arbitrary open set from the cover described in Equation (106). Without loss of generality, we may relabel the indices such that both Top-$k$ maps are constantly equal to $\{1, \ldots, k\}$ throughout $U$. Under this reindexing, Equation (104) simplifies to

$$\sum_{i=1}^{k} \text{softmax}_i \left( \{W_i x + b_i\}_{i=1}^{k} \right) \cdot \mathcal{E}_i(x)$$

$$= \sum_{i=1}^{k} \text{softmax}_i \left( \{W_i' x + b_i'\}_{i=1}^{k} \right) \cdot \mathcal{E}_i'(x) \quad \text{for all } x \in U. \tag{107}$$

According to Assumption 1, the expert functions $\mathcal{E}_i$ are strongly distinct, which implies that they remain distinct on the open set $U$. The same conclusion applies to the functions $\mathcal{E}_i'$. Therefore, the assumptions of Theorem B.5 are satisfied on $U$, and Equation (107) falls within its scope. As a consequence, there exist constants $c_W \in \mathbb{R}^d$ and $c_b \in \mathbb{R}$ such that, up to reindexing, we have for all $i = 1, \ldots, k$,

$$W_i' = W_i + c_W, \quad b_i' = b_i + c_b, \tag{108}$$

and furthermore, $\mathcal{E}_i = \mathcal{E}_i'$ on $U$.

**Step 4.** For any index $m \in \{3, 4, \ldots, n\}$, we invoke Proposition C.3 to select an open set $V_1$ from the cover in Equation (106) such that both indices $1$ and $k$ appear in $T(V_1)$. Restricting Equation (104) to $V_1$ and applying Theorem C.4, we conclude that there exist indices $1 \leq t_1, s_1 \leq n'$ such that

$$W_1 - W_m = W_{t_1}' - W_{s_1}'. \tag{109}$$

Applying the same reasoning with indices $2$ and $m$, we obtain $1 \leq t_2, s_2 \leq n'$ such that

$$W_2 - W_m = W_{t_2}' - W_{s_2}'. \tag{110}$$

Subtracting Equation (110) from Equation (109) yields

$$W_1' - W_2' = W_1 - W_2 = (W_1 - W_m) - (W_2 - W_m) = (W_{t_1}' - W_{s_1}') - (W_{t_2}' - W_{s_2}'). \tag{111}$$

Due to the linear independence guaranteed by Assumption 2, this identity can only hold if $t_1 = 1$, $t_2 = 2$, and $s_1 = s_2$. Denoting this shared index by $\tau(m)$, i.e., $\tau(m) = s_1 = s_2$, we then have

$$W_1 - W_m = W_1' - W_{\tau(m)}', \tag{112}$$

which implies

$$W_{\tau(m)}' - W_m = W_1' - W_1 = c_W. \tag{113}$$

Similarly, we deduce

$$b_{\tau(m)}' - b_m = b_1' - b_1 = c_b. \tag{114}$$

Since $m$ ranges over $\{3, 4, \ldots, n\}$, the corresponding values of $\tau(m)$ must be distinct. If not, suppose that $\tau(m) = \tau(m')$ for some $m \neq m'$, which would imply

$$W_m - W_{m'} = W_{\tau(m)}' - W_{\tau(m')}' = 0, \tag{115}$$

contradicting Assumption 3.

Applying a symmetric argument to the parameters of $\mathcal{S}(\cdot; \phi')$, we conclude that $n = n'$. Therefore, there exists a permutation $\tau$ of $\{1, \ldots, n\}$ such that

$$W_i' = W_{\tau(i)} + c_W, \quad b_i' = b_{\tau(i)} + c_b. \tag{116}$$

Finally, the analysis above shows that for any $x \in \Omega\left(\{W_i, b_i\}_{i=1}^{n}\right)$ with $\tau(i) \in \text{Top-}k((W_i x + b_i)_{i=1}^{n})$–i.e., when index $i$ is selected by the Top-$k$ operator in $\mathcal{S}$–we have

$$\mathcal{E}_{\tau(i)}(x) = \mathcal{E}_i'(x). \tag{117}$$

This completes the proof of Theorem C.4. $\qquad \square$

**Remark C.7.** While Theorem C.4 is conceptually analogous to Theorem B.5, it is crucial to recognize that establishing the result for SMoE involves substantially greater technical complexity. The main challenge arises from the Top-$k$ operator, which introduces discontinuities by dynamically changing the subset of active experts in a manner that depends intricately on the input. This input-dependent behavior complicates the analysis and makes the theoretical treatment significantly more delicate.

**Remark C.8** (Rationale behind the assumptions in Theorem C.4). We begin by restating the two assumptions made in Theorem C.4:

1. Both $\{\mathcal{E}(\cdot;\theta_i)\}_{i=1}^n$ and $\{\mathcal{E}(\cdot;\theta_i')\}_{i=1}^{n'}$ consist of pairwise strongly distinct functions;

2. Both $\{W_{i-1} - W_i\}_{i=2}^n$ and $\{W_{i-1}' - W_i'\}_{i=2}^{n'}$ are linear independent subsets of $\mathbb{R}^d$.

These assumptions are strictly stronger than those required in Theorem B.5. We now discuss their necessity and implications in greater detail.

*Assumption 1.* This condition arises primarily from the behavior of the Top-$k$ operator, which induces input-dependent expert selection. As a result, the functional contribution of an expert is restricted to regions where it is actively selected by the gating mechanism. Outside these regions, the expert can behave arbitrarily without influencing the output. Therefore, if the experts are merely pairwise distinct–rather than pairwise strongly distinct–it becomes possible for different sets of expert functions to yield identical overall behavior when restricted to their respective activation domains. This ambiguity underscores the necessity of strong distinctness to ensure functional identifiability in the SMoE setting.

*Assumption 2.* In practice, the number of experts $n$ is typically much smaller than the input (token) dimension $D$. As a result, the collections $\{W^{(G,i-1)} - W_i\}_{i=2}^n$ and $\{W'^{(G,i-1)} - W_i'\}_{i=2}^{n'}$ are generically linearly independent. However, when linear dependence arises, it can prevent certain expert pairs from ever being simultaneously selected by the gating mechanism across all possible inputs. This limitation introduces singular symmetries: different parameter configurations that yield functionally identical outputs but cannot be related via the equivalence structure defined in Theorem C.4.

To illustrate this phenomenon concretely, consider an example with $n = 4$ and $k = 2$, and let $\mathcal{E}_1, \mathcal{E}_2, \mathcal{E}_3, \mathcal{E}_4$ denote arbitrary expert functions. Define two SMoE models $\mathcal{S}_1$ and $\mathcal{S}_2$, whose gating logits are $(-2x, -x, x, 2x)$ and $(-3x, -2x, 2x, 3x)$, respectively. The resulting functions take the form:

$$\mathcal{S}_1(x) = \begin{cases} \text{softmax}_1(-2x, -x) \cdot \mathcal{E}_1(x) + \text{softmax}_2(-2x, -x) \cdot \mathcal{E}_2(x) & \text{if } x < 0, \\ \text{softmax}_1(x, 2x) \cdot \mathcal{E}_3(x) + \text{softmax}_2(x, 2x) \cdot \mathcal{E}_4(x) & \text{if } x > 0, \end{cases} \tag{118}$$

and

$$\mathcal{S}_2(x) = \begin{cases} \text{softmax}_1(-3x, -2x) \cdot \mathcal{E}_1(x) + \text{softmax}_2(-3x, -2x) \cdot \mathcal{E}_2(x) & \text{if } x < 0, \\ \text{softmax}_1(2x, 3x) \cdot \mathcal{E}_3(x) + \text{softmax}_2(2x, 3x) \cdot \mathcal{E}_4(x) & \text{if } x > 0. \end{cases} \tag{119}$$

It is easy to verify that $\mathcal{S}_1(x) = \mathcal{S}_2(x)$ for all $x \in \mathbb{R} \setminus \{0\}$, where the gating logits are pairwise distinct and the Top-$k$ selections remain constant. However, no transformation of the form specified in Theorem C.4 maps one configuration to the other. This demonstrates how singular symmetries can arise in the SMoE architecture, even when functional outputs coincide on an open dense subset of the input domain.

**Remark C.9** (The case of $k = 1$). In the special case where $k = 1$, the SMoE function in Equation (21) simplifies to

$$\mathcal{S}\left(x; \{W_i, b_i, \theta_i\}_{i=1}^n\right) = \mathcal{E}\left(x; \theta_i\right), \tag{120}$$

where the index $i$ is determined by

$$i = \underset{i=1,\ldots,n}{\operatorname{argmax}} \left(W_i x + b_i\right). \tag{121}$$

In this setting, the Top-1 gating mechanism selects only the expert with the highest score, and the softmax reduces to a one-hot distribution with a single nonzero entry equal to 1.

Beyond the standard $G(n)$ symmetry acting on the expert parameters, the SMoE architecture with $k = 1$ exhibits an additional, nontrivial invariance under the action of the multiplicative group $\mathbb{R}_{>0}$. Specifically, for any scalar $c > 0$, we have

$$S\left(x; \{W_i, b_i, \theta_i\}_{i=1}^n\right) = S\left(x; \{cW_i, cb_i, \theta_i\}_{i=1}^n\right), \tag{122}$$

as the argmax used to select the active expert remains invariant under uniform positive scaling:

$$\underset{i=1,\ldots,n}{\operatorname{argmax}} \left(W_i x + b_i\right) = \underset{i=1,\ldots,n}{\operatorname{argmax}} \left(cW_i x + cb_i\right), \tag{123}$$

for all $x \in \Omega\left(\{W_i, b_i\}_{i=1}^n\right)$. Furthermore, since only one expert is active at any given input, the architecture does not involve explicit interactions between experts. This leads to a more complex symmetry structure, including hidden and continuous transformations, which complicates theoretical analysis. For this reason, we exclude the case $k = 1$ from our main results and leave its investigation to future work.

# D  Technical Details for Sections 5 and 6

## D.1  Proof for the Sufficiency of Permutation Invariance in LMC of MoE

The following result establishes that translation transformations do not affect the barrier loss between two parameter configurations.

**Proposition D.1.** *Let* $h = (c_W, c_b, \operatorname{id}_n) \in G(n)$*, where* $\operatorname{id}_n \in \mathrm{S}_n$ *denotes the identity permutation. For any* $\phi_A, \phi_B \in \Phi(n)$*, the barrier loss remains invariant under translation:*

$$B(\phi_A, \phi_B) = B(\phi_A, h\phi_B). \tag{124}$$

*That is, the translation components do not contribute to the barrier loss as defined in Equation* (1).

*Proof.* Recall that

$$B(\phi_A, \phi_B) = \sup_{t \in [0,1]} \left[\mathcal{L}(t\phi_A + (1-t)\phi_B) - (t\mathcal{L}(\phi_A) + (1-t)\mathcal{L}(\phi_B))\right]. \tag{125}$$

For $t \in [0, 1]$, denote $h_t = (tc_W, tc_b, \operatorname{id}_n) \in G(n)$. For $\phi_A, \phi_B \in \Phi(n)$ such that

$$\phi_A = (W_i^A, b_i^A, \theta_i^A)_{i=1,\ldots,n} \tag{126}$$
$$\phi_B = (W_i^B, b_i^B, \theta_i^B)_{i=1,\ldots,n}, \tag{127}$$

we have:

$$
\begin{aligned}
& t\phi_A + (1-t)(h\phi_B) \\
&= t(W_i^A, b_i^A, \theta_i^A)_{i=1,\ldots,n} + (1-t)(W_i^B + c_W, b_i^B + c_b, \theta_i^B)_{i=1,\ldots,n} \\
&= \left(tW_i^A + (1-t)W_i^B + (1-t)c_W, tb_i^A + (1-t)b_i^B + (1-t)c_b, t\theta_i^A + (1-t)\theta_i^B\right)_{i=1,\ldots,n} \\
&= \left(h_{1-t}\right)\left(t\phi_A + (1-t)\phi_B\right). 
\end{aligned}
\tag{128}
$$

Since the group action of $G(n)$ on $\Phi(n)$ preserves the functionality of the MoE function $\mathcal{D}$, we have

$$\mathcal{D}\left(\cdot; \phi_B\right) = \mathcal{D}\left(\cdot; h\phi_B\right), \tag{129}$$

and

$$
\begin{aligned}
\mathcal{D}\left(\cdot; t\phi_A + (1-t)(h\phi_B)\right) &= \mathcal{D}\left(\cdot; \left(h_{1-t}\right)\left(t\phi_A + (1-t)\phi_B\right)\right) \\
&= \mathcal{D}\left(\cdot; t\phi_A + (1-t)\phi_B\right).
\end{aligned}
\tag{130}
$$

These observations lead to

$$\mathcal{L}(\phi_B) = \mathcal{L}(h\phi_B), \tag{131}$$
$$\mathcal{L}\left(t\phi_A + (1-t)(h\phi_B)\right) = \mathcal{L}\left(t\phi_A + (1-t)\phi_B\right). \tag{132}$$

Thus, for $t \in [0, 1]$, we have

$$B(\phi_A, h\phi_B)$$
$$= \sup_{t\in[0,1]} \left[ \mathcal{L}\big(t\phi_A + (1-t)(h\phi_B)\big) - \big(t\mathcal{L}(\phi_A) + (1-t)\mathcal{L}(h\phi_B)\big) \right]$$
$$= \sup_{t\in[0,1]} \left[ \mathcal{L}\big(t\phi_A + (1-t)\phi_B\big) - \big(t\mathcal{L}(\phi_A) + (1-t)\mathcal{L}(\phi_B)\big) \right]$$
$$= B(\phi_A, \phi_B). \tag{133}$$

Therefore, the proof is finished. $\square$

### D.2 Proof of the Permutation-Invariant Property for Equation (11)

**Proposition D.2.** *The cost function defined in Method 2 of Section 5.2 is permutation-invariant with respect to the hidden units of the experts in the MoE models.*

*Proof.* For each expert $i$ in the first Mixture-of-Experts (MoE) model $\phi$, let $A_i \in \mathbb{R}^{h \times d}$ and $u_i \in \mathbb{R}^h$ be the weight matrix and bias for the first layer, and $B_i \in \mathbb{R}^{d \times h}$ and $v_i \in \mathbb{R}^d$ be the weight matrix and bias for the second layer. Define the augmented matrices $\widetilde{A}_i = [A_i, u_i] \in \mathbb{R}^{h \times (d+1)}$ and $\widetilde{B}_i = [B_i, v_i] \in \mathbb{R}^{d \times (h+1)}$. Similarly, for each expert $j$ in the second MoE model $\phi'$, define $A'_j$, $u'_j$, $B'_j$, $v'_j$, and the augmented matrices $\widetilde{A}'_j$, $\widetilde{B}'_j$ accordingly. We define the Gram matrices $(\widetilde{A}_i)^\top \widetilde{A}_i \in \mathbb{R}^{(d+1) \times (d+1)}$ and $\widetilde{B}_i(\widetilde{B}_i)^\top \in \mathbb{R}^{d \times d}$ for each expert $i$ in $\phi$, and similarly for $\phi'$.

To establish permutation invariance, consider a permutation matrix $P \in \mathbb{R}^{h \times h}$ applied to the hidden units of expert $i$ in $\phi$. For the first layer, the permuted augmented weight matrix is $P\widetilde{A}_i$. For the second layer, the permutation affects the columns of $B_i$ within $\widetilde{B}_i = [B_i, v_i]$, so the transformed matrix is $\widetilde{B}_i \begin{bmatrix} P^\top & 0 \\ 0 & 1 \end{bmatrix}$, leaving the bias $v_i$ unchanged. We now verify the invariance of the Gram matrices under this permutation. For the first layer:

$$(P\widetilde{A}_i)^\top (P\widetilde{A}_i) = (\widetilde{A}_i)^\top P^\top P \widetilde{A}_i = (\widetilde{A}_i)^\top I \widetilde{A}_i = (\widetilde{A}_i)^\top \widetilde{A}_i, \tag{134}$$

since $P^\top P = I$, the $h \times h$ identity matrix. For the second layer:

$$\left( \widetilde{B}_i \begin{bmatrix} P^\top & 0 \\ 0 & 1 \end{bmatrix} \right) \left( \widetilde{B}_i \begin{bmatrix} P^\top & 0 \\ 0 & 1 \end{bmatrix} \right)^\top = \widetilde{B}_i \begin{bmatrix} P^\top & 0 \\ 0 & 1 \end{bmatrix} \begin{bmatrix} P & 0 \\ 0 & 1 \end{bmatrix} \widetilde{B}_i^\top$$
$$= \widetilde{B}_i \begin{bmatrix} P^\top P & 0 \\ 0 & 1 \end{bmatrix} \widetilde{B}_i^\top$$
$$= \widetilde{B}_i \begin{bmatrix} I & 0 \\ 0 & 1 \end{bmatrix} \widetilde{B}_i^\top$$
$$= \widetilde{B}_i \widetilde{B}_i^\top, \tag{135}$$

where $\begin{bmatrix} I & 0 \\ 0 & 1 \end{bmatrix}$ is the $(h+1) \times (h+1)$ identity matrix. This confirms that both $(\widetilde{A}_i)^\top \widetilde{A}_i$ and $\widetilde{B}_i(\widetilde{B}_i)^\top$ are invariant under permutations of the hidden units.

The cost matrix $C \in \mathbb{R}^{n \times n}$ for the Linear Assignment Problem (LAP) is $C = \{C_{i,j}\}_{1 \le i \le n, 1 \le j \le n}$, where

$$C_{i,j} = \left( \left\| (\widetilde{A}_i)^\top \widetilde{A}_i - (\widetilde{A}'_j)^\top \widetilde{A}'_j \right\|_F^2 + \left\| \widetilde{B}_i(\widetilde{B}_i)^\top - \widetilde{B}'_j(\widetilde{B}'_j)^\top \right\|_F^2 \right)^{\frac{1}{2}}. \tag{136}$$

Since the Gram matrices $(\widetilde{A}_i)^\top \widetilde{A}_i$ and $\widetilde{B}_i(\widetilde{B}_i)^\top$ (and their counterparts in $\phi'$) are permutation-invariant, their differences and the Frobenius norms of these differences are also invariant. Consequently, $C_{ij}$ remains unchanged under permutations of the hidden units in either $\phi$ or $\phi'$.

Thus, the cost function is permutation-invariant with respect to the hidden units of the experts, completing the proof. $\square$

### D.3 Formal formulation of DeepSeekMoE

For completeness, we present the notation and formulations of three variants of MoE models: Dense MoE, Sparse MoE (SMoE), and DeepSeekMoE.

**Expert.** Given $d$ denoting the input dimension. We define an *Expert* as a function $\mathcal{E}(\cdot;\theta)\colon \mathbb{R}^d \to \mathbb{R}^d$, parameterized by $\theta \in \mathbb{R}^e$ with $e \in \mathbb{N}$ is the total number of trainable parameters of $\mathcal{E}$. In this work, we consider each expert $\mathcal{E}(\cdot;\theta)$ to be a feedforward neural network with ReLU activation functions. Unless stated otherwise, all experts are assumed to share the same architecture.

**Mixture of Expert with dense gating.** Given $n$ denoting the number of experts, we define *Mixture-of-Experts with dense gating* as a function $\mathcal{D}\colon \mathbb{R}^d \to \mathbb{R}^d$ such that

$$\mathcal{D}\big(x; \{W_i, b_i, \theta_i\}_{i=1}^n\big) = \sum_{i=1}^n \text{softmax}_i\big(s_1(x), \ldots, s_n(x)\big)\mathcal{E}(x; \theta_i), \qquad (137)$$

where $s_i(x) = W_i x + b_i$ determines the contribution of each expert to the final output. Here, $\theta_i$ are the parameters of the $i^{\text{th}}$ expert. The function $s = (s_1, \ldots, s_n)$ is called the *gating score*, and is parameterized as $s(\cdot; \{W_i, b_i\}_{i=1}^n)$ with $(W_i, b_i) \in \mathbb{R}^d \times \mathbb{R}$ are the corresponding *gating* parameters.

**Mixture-of-Experts with sparse gating.** Given a positive integer $k \leq n$ denoting the number of activated experts, define the Top-$k$ map by $\text{Top-}k(z) = \{i_1, \ldots, i_k\}$ for $z = (z_1, \ldots, z_n) \in \mathbb{R}^n$, where $i_1, \ldots, i_k$ are the indices corresponding to the $k$ largest components of $x$. In the event of ties, we select smaller indices first. We define a *Mixture-of-Experts with sparse gating* (SMoE) as a function $\mathcal{S}\colon \mathbb{R}^d \to \mathbb{R}^d$ as follows. For $x \in \mathbb{R}^d$, let $T(x) = \text{Top-}k(s(x))$, then define:

$$\mathcal{S}\big(x; \{W_i, b_i, \theta_i\}_{i=1}^n\big) = \sum_{i \in T(x)} \text{softmax}_i\left(\big(s_i(x)\big)_{i \in T(x)}\right) \cdot \mathcal{E}(x; \theta_i) \qquad (138)$$

In other words, the Top-$k$ selects the $k$ highest-scoring experts used to compute the output.

**DeepseekMoE with shared and routed experts.** The DeepseekMoE architecture extends the sparse Mixture-of-Experts by incorporating both shared experts that are always active and routed experts that are sparsely activated based on the input. Given positive integers $n_s$ and $n_r$ denoting the number of shared and routed experts, respectively ($n = n_s + n_r$ as the total number of experts), and a positive integer $k_r \leq n_r$ denoting the number of activated routed experts, the DeepseekMoE layer is defined as a function $\mathcal{M}\colon \mathbb{R}^d \to \mathbb{R}^d$ such that

$$\mathcal{M}\big(x; \{\theta_i^{(s)}\}_{i=1}^{n_s}, \{W_i, b_i, \theta_i\}_{i=1}^{n_r}\big) = \sum_{i=1}^{n_s} \mathcal{E}_i^{(s)}(x; \theta_i^{(s)}) + \sum_{i=1}^{n_r} g_i(x)\mathcal{E}(x; \theta_i^{(r)}), \qquad (139)$$

where:

- $\mathcal{E}_i^{(s)}(\cdot; \theta_i^{(s)})\colon \mathbb{R}^d \to \mathbb{R}^d$ are the shared experts, parameterized by $\theta_i^{(s)}$,
- $\mathcal{E}_i^{(r)}(\cdot; \theta_i^{(r)})\colon \mathbb{R}^d \to \mathbb{R}^d$ are the routed experts, parameterized by $\theta_i^{(r)}$,
- The gating values $g_i(x)$ are computed as follows:

$$\tilde{s}(x) = \text{softmax}\big(s_1(x), \ldots, s_{n_r}(x)\big) \in \mathbb{R}^{n_r},$$

$$g_i(x) = \begin{cases} \tilde{s}_i(x), & \text{if } i \in \text{Top-}k_r(\tilde{s}(x)) \\ 0, & \text{otherwise} \end{cases}$$

This formulation allows the model to leverage a combination of always-active shared experts and input-dependent routed experts, enhancing efficiency and performance in large-scale neural networks.

## E  Impact of Feedforward Reinitialization on Pretrained Transformer Performance

We empirically investigate the sensitivity of pretrained Transformer models to targeted parameter reinitialization within their Feedforward Network (FFN) modules. The FFN, a critical component for

non-linear transformations and feature representation within each Transformer block, was selectively reinitialized layer by layer to assess its functional contribution and the robustness of the overall model architecture. This analysis aims to identify the extent to which performance is dependent on the learned parameters of individual FFNs and to potentially highlight layers that are more critical to maintaining task proficiency. To this end, experiments were conducted on two representative model-task pairs:

1. Image Classification using a Vision Transformer (ViT-Base, patch size 16, image resolution 224) pretrained on ImageNet.
2. Language Modeling using a GPT-2 model (12-layer, hidden size 768) pretrained on Wiki-Text103.

For each model, the parameters of the FFN subcomponent within a single Transformer block were reinitialized using a standard initialization scheme (e.g., variance scaling), while all other model parameters, including attention weights and the FFNs in other layers, were kept frozen from the pretrained state. Following reinitialization, the models were evaluated on their respective tasks without any subsequent fine-tuning. This approach isolates the immediate effect of disrupting a single FFN on pretrained performance.

Figures 3 and 4 illustrate the performance degradation observed when reinitializing the FFN at different layers for the ViT and GPT-2 models, respectively. A striking observation is the significantly larger performance drop (increase in loss, decrease in accuracy) when reinitializing the FFN in the first layer (Layer 0) compared to any subsequent layer. While reinitializing FFNs in layers beyond the first does result in performance degradation, this impact is markedly less severe than at Layer 0. The results further reveal a non-uniform sensitivity profile among these subsequent layers, with FFN reinitialization in middle layers tending to induce a more pronounced performance decrement than in later layers. This pattern suggests a unique functional importance or sensitivity associated with the initial layer's FFN, while the network's architecture, particularly the presence of skip connections, appears to enable a greater degree of resilience to disruption in FFNs at deeper layers.

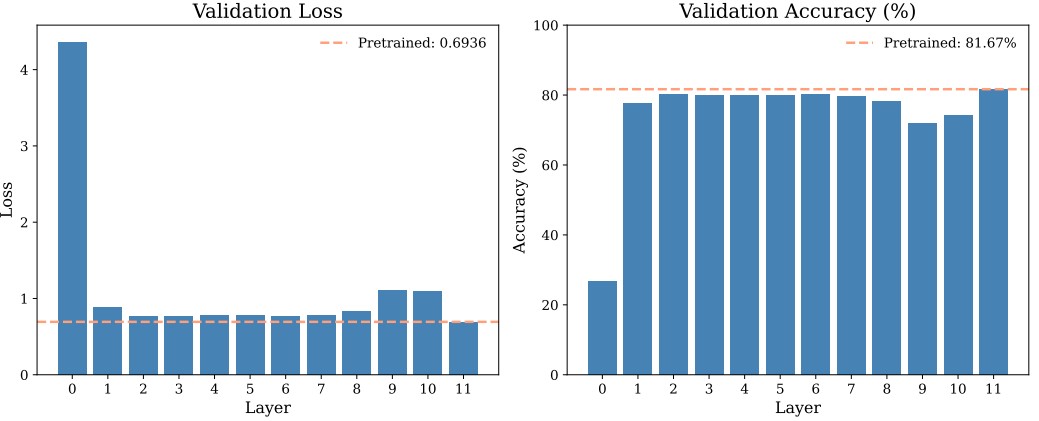

Figure 3: Performance degradation in ViT-Base on ImageNet due to FFN reinitialization at different layers.

A key observation from these experiments, particularly for deeper and more complex models with skip connections, is that reinitializing a single FFN (except potentially the very first one) often does not lead to a catastrophic performance collapse. This phenomenon can be partially attributed to the presence of skip connections, which facilitate the unimpeded flow of information across layers, allowing features learned by other parts of the network to bypass the disrupted FFN. Furthermore, standard weight initialization schemes typically employ relatively small scales centered around zero, meaning the reinitialized FFN parameters introduce a perturbation that is initially small and balanced compared to the potentially large and specialized weights learned during pretraining. This combination of architectural properties (skip connections) and initialization characteristics helps the model maintain functionality, suggesting that the reinitialized state may remain within or close to the original optimization basin found during pretraining. Previous work [53] has highlighted how skip

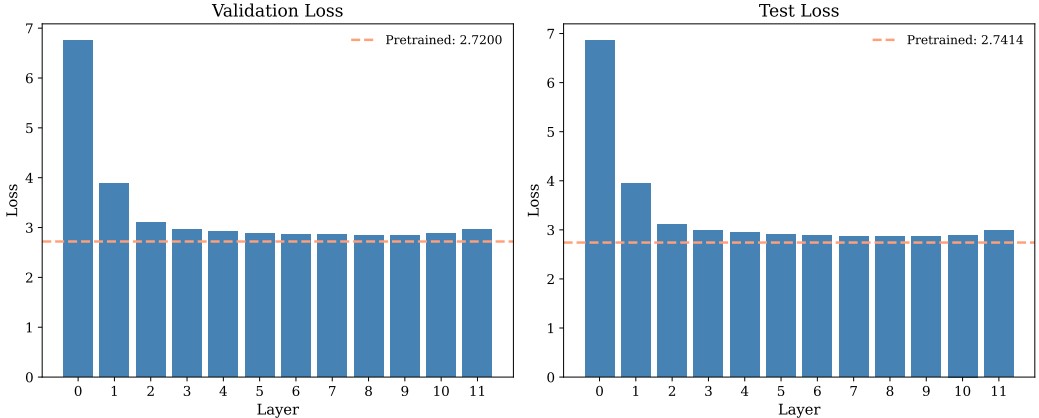

Figure 4: Effect of FFN reinitialization on GPT-2 perplexity across layers on WikiText103.

connections contribute to flatter minima and improve training stability in deep networks, which aligns with the observation that reinitialization causes less disruption than might intuitively be expected.

To further investigate the generality of this observation, we also conducted FFN reinitialization experiments on tasks commonly used for evaluating model robustness and optimization landscape properties, including text classification tasks (IMDB Review [57], AGNEWS [92], DBPedia [8]) and additional language modeling tasks (EnWik8 [38], Penn Treebank (Word Level) [58]). Table 4 presents a summary of the performance changes on these datasets after reinitializing an arbitrary single FFN layer compared to the original pretrained performance, and also shows performance after subsequent fine-tuning.

Table 4: Effect of single FFN reinitialization on various tasks. "Pretrained" indicates performance of the original model. "Reinitialize" shows performance after reinitializing a single FFN layer. "Finetune" shows performance after fine-tuning the reinitialized model. "Naive Loss Barrier" refers to the loss barrier observed between two fine-tuned models initialized with different random seeds and trained with different batch orders.

| Dataset | Pretrained Loss | Pretrained Accuracy | Reinitialize Loss | Reinitialize Accuracy | Finetune Loss | Finetune Accuracy | Naive Loss Barrier |
|---|---|---|---|---|---|---|---|
| IMDB Review | 0.3724 | 87.50 | 0.6452 | 68.75 | 0.4284 | 87.50 | 0.0019 |
| AGNEWS | 0.2849 | 90.50 | 0.6006 | 76.64 | 0.2986 | 90.73 | 0.0007 |
| DBPedia | 0.1896 | 93.75 | 0.3017 | 87.50 | 0.2036 | 94.47 | 0.0003 |
| EnWik8 | 0.9684 | - | 1.3053 | - | 0.9506 | - | 0.0082 |
| Penn Treebank | 4.5246 | - | 6.9699 | - | 4.4639 | - | 0.0454 |

As shown in Table 4, while reinitializing a single FFN does lead to performance degradation on these tasks (e.g., loss increases by approximately 0.1 to 0.3), the magnitude of this change is relatively modest compared to the significant loss increases observed in experiments involving more widespread reinitialization or on larger tasks like GPT-2 on WikiText103 or ViT on ImageNet (where full model reinitialization can increase loss by several units). The relatively small performance drop suggests that for these specific datasets and this focused reinitialization strategy, the model's state remains close to its original pretrained optimum. Consequently, metrics used to assess mode connectivity between the original and reinitialized states, such as the Naive Loss Barrier (shown in Table 4), yield very low values. This indicates that a simple linear path in parameter space between the original and reinitialized single-FFN models exhibits low loss values. However, due to the minor performance impact of the reinitialization itself, this low barrier value may primarily reflect the fact that the reinitialized state is already highly functional and located within the same or a very accessible optimization basin, rather than necessarily implying a broadly flat landscape between drastically different high-performing modes. Therefore, while consistent with observations of flat minima facilitated by skip connections [53], these specific reinitialization experiments on these datasets may not provide deep insights into complex mode connectivity far from the original optimum.

In summary, these findings highlight the heterogeneous functional roles of FFNs across different layers in Transformer models and demonstrate varying degrees of sensitivity to parameter reinitialization. The relative robustness observed for many layers, especially in deep models, is consistent with the structural benefits of skip connections and the properties of standard initialization. This layer-wise sensitivity analysis contributes to a better understanding of Transformer architecture and has potential implications for model compression techniques (e.g., identifying less critical FFNs), efficient fine-tuning strategies, and the design of conditional computation mechanisms like Mixture-of-Experts (MoE) routing.

## F   Experimental Details and Hyperparameters

We conduct a comprehensive evaluation of Linear Mode Connectivity (LMC) across a broad suite of vision and language modeling benchmarks. For vision tasks, our study includes MNIST, CIFAR-10, CIFAR-100, ImageNet-1k, as well as transfer learning scenarios from ImageNet-21k to CIFAR-10 and CIFAR-100. For language modeling, we utilize WikiText103 and the One Billion Word dataset. All experiments are performed using pretrained Transformer-based architectures, wherein all original model parameters are frozen and only the inserted Mixture-of-Experts (MoE) layers are subject to fine-tuning.

For vision tasks, we adopt Vision Transformer (ViT) backbones, while GPT-2 serves as the backbone for language modeling. Model configurations–including architecture depth, width, and tokenization context–are adjusted per dataset to reflect task-specific complexity and scale. Each expert module is architecturally aligned with the original feedforward network (MLP) layers of the pretrained model. To promote stable convergence, learning rates are independently tuned for each setting within the range of $10^{-4}$ to $10^{-1}$.

**MNIST** We utilize the original MNIST grayscale images, each resized into a grid of non-overlapping patches with a patch size of 7. A lightweight Vision Transformer (ViT) model is employed, configured with an embedding dimension of 32 and a depth of 1 to 2 Transformer layers, depending on the specific setup. Each self-attention layer uses 4 attention heads to process the patch embeddings. Dropout is applied at a rate of 0.0 and the activation function throughout the network is `gelu`. The model is optimized using stochastic gradient descent (SGD) during both the pretraining and fine-tuning stages.

**CIFAR-10.** We utilize the original CIFAR-10 images, each divided into non-overlapping patches with a patch size of 4. A Vision Transformer (ViT) model is employed, configured with an embedding dimension of 128, 8 self-attention heads, and a depth ranging from 2 to 6 Transformer layers. Dropout is applied at a rate of 0.0, and the activation function throughout the network is `gelu`. The model is optimized using the Adam optimizer during pretraining and stochastic gradient descent (SGD) during fine-tuning.

**CIFAR-100.** We resize CIFAR-100 images to $224 \times 224$ and divide them into non-overlapping patches with a patch size of 16. A Vision Transformer (ViT) model is employed, configured with an embedding dimension of 384, 6 self-attention heads, and a depth ranging from 6 to 12 Transformer layers. Dropout is applied at a rate of 0.0, and the activation function throughout the network is `gelu`. The model is optimized using the Adam optimizer during pretraining and stochastic gradient descent (SGD) during fine-tuning.

**Imagenet21k→CIFAR10.**  We adopt the ViT-Small-Patch16-224 model [75], pretrained on ImageNet-21k and subsequently fine-tuned on CIFAR-10. The model consists of 12 layers, a hidden size of 384, an MLP size of 1536, and 6 attention heads, resulting in approximately 22.2M parameters. It employs a patch size and stride of 16. Dropout is disabled (set to 0.0), and the activation function is `gelu`. Stochastic Gradient Descent (SGD) is employed during fine-tuning.

**Imagenet21k→CIFAR100.**  We adopt the ViT-Small-Patch16-224 model [75], pretrained on ImageNet-21k and subsequently fine-tuned on CIFAR-100. The model consists of 12 layers, a hidden size of 384, an MLP size of 1536, and 6 attention heads, resulting in approximately 22.2M parameters. It employs a patch size and stride of 16. The attention probability dropout and hidden layer dropout are set to 0.0, and the activation function is `gelu`. The SGD optimizer is employed during fine-tuning.

**ImageNet-1k.** We use the ViT-Base-Patch16-224 model [20], consisting of 12 Transformer layers, each with a hidden size of 768, an MLP hidden dimension of 3072, and 12 self-attention heads. The total parameter count is approximately 86M. The encoder employs a patch size and stride of 16, and the `gelu` activation function is applied throughout. Both the attention dropout and hidden dropout rates are set to 0.0. The model is pretrained for 300 epochs with a linear warm-up of 10,000 steps, followed by cosine decay scheduling. Fine-tuning is performed for 30 epochs without warm-up. Optimization is carried out using the Adam optimizer during both pretraining and MoE-layer fine-tuning stages with identical hyperparameter settings.

**WikiText103.** We adopt a GPT-2 model architecture consisting of 12 layers with 12 attention heads and an embedding dimension of 768. The context length and maximum position embeddings are set to 1024. The dropout rate is set to 0.1, and the activation function is `gelu`. The vocabulary size is 50,257. The model uses the standard GPT-2 initialization and layer normalization settings. The Adam optimizer is employed during both pretraining and fine-tuning.

**One Billion Word.** Similar to WikiText103, the GPT-2 model comprises 12 transformer layers, 12 attention heads, and an embedding dimension of 768, but with a reduced context and positional length of 256. The dropout rate and activation function remain identical to those in the WikiText103 setup. The vocabulary size is expanded to 793470 to capture the dataset's linguistic diversity. The model is first pretrained for 500000 steps using the Adam optimizer, followed by a fine-tuning phase of 100000 steps with a 2000-step linear warm-up schedule. Both stages share the same optimization hyperparameters unless otherwise noted.

All experiments are executed on a single NVIDIA H100 GPU with 80GB of memory, except for the One Billion Word task, which utilizes two H100 GPUs. Due to the use of the JAX framework, approximately 75% of GPU memory (around 60GB) is pre-allocated by default. The number of CPU workers used in data loading and preprocessing is kept less than or equal to 10 across all experiments. For smaller-scale tasks such as MNIST, CIFAR-10, CIFAR-100, and transfer learning from ImageNet-21k, each experiment completes in under 30 minutes. For larger-scale tasks, the fine-tuning durations are approximately 15 hours for ImageNet-1k, 3.5 hours for WikiText103, and 4 hours for One Billion Word.

## G    Experimental Results

### G.1    Verification of Linear Mode Connectivity across diverse configurations

To rigorously evaluate the robustness and generality of Linear Mode Connectivity (LMC) in expert-based Transformer architectures, we conducted systematic experiments across a wide range of Mixture-of-Experts (MoE) configurations. Our study includes three representative variants–vanilla MoE, SMoE, and DeepSeekMoE–which differ in both architectural structure and expert routing mechanisms. In all cases, we substituted the feed-forward network (FFN) of the *first* Transformer layer with an MoE module, while leaving the rest of the model architecture unchanged. All models were fine-tuned from identical random initializations to ensure consistency in comparison.

Table 5 summarizes the experimental conditions, covering diverse model depths (2, 6, 12 layers), task domains (including vision benchmarks such as MNIST, CIFAR-10, ImageNet21k→CIFAR-100, and language modeling with WikiText103), and expert configurations. SMoE employs a sparsity level of $k = 2$ active experts per token, whereas DeepSeekMoE further incorporates a shared expert ($s = 1$) to encourage parameter reuse and stability. Across nearly all configurations, LMC consistently emerges: linear interpolation between independently fine-tuned models (initialized identically) yields smooth loss trajectories, with no significant barriers. This suggests that expert-based models, even with dynamic routing and varying capacities, tend to converge to connected optima under matched training setups. The corresponding loss curves, linked in Table 5, reinforce this observation.

These results provide strong empirical evidence that LMC is a robust and general phenomenon in expert architectures, supporting the hypothesis that expert specialization and routing do not disrupt the connected geometry of the optimization landscape when alignment in initialization and architecture is preserved.

Table 5: Experimental configurations for LMC analysis across MoE, SMoE, and DeepSeekMoE variants, evaluated on multiple datasets and settings. Datasets of the form $A \rightarrow B$ denote pretraining on $A$ and finetuning on $B$. SMoE uses $k = 2$ active experts, while DeepSeekMoE uses $k = 2$ with an additional shared expert ($s = 1$). In all cases, the MLP in the *first* Transformer layer is replaced with an MoE module.

| Method | Dataset | No. layers | No. experts | Figure |
|---|---|---|---|---|
| MoE | MNIST | 1 | [2, 4] | [5, 6] |
| | | 2 | [2, 4] | [7, 8] |
| | CIFAR-10 | 2 | [2, 4, 6] | [9, 10, 11] |
| | | 6 | [2, 4, 6] | [12, 13, 14] |
| | CIFAR-100 | 6 | [2, 4, 6] | [15, 16, 17] |
| | ImageNet-21k→CIFAR-10 | 12 | [2, 4, 6] | [18, 19, 20] |
| | ImageNet-21k→CIFAR-100 | 12 | [2, 4, 6] | [21, 22, 23] |
| | ImageNet-1k | 12 | [2, 4, 6, 8] | [24, 25, 26, 27] |
| | WikiText103 | 12 | [2, 4, 6, 8] | [28, 29, 30, 31] |
| | One Billion Word | 12 | [2, 4, 6, 8] | [32, 33, 34, 35] |
| SMoE ($k = 2$) | MNIST | 1 | [4] | [36] |
| | | 2 | [4] | [37] |
| | CIFAR-10 | 2 | [4, 8] | [38, 39] |
| | | 6 | [4, 8] | [40, 41] |
| | CIFAR-100 | 6 | [4, 8] | [42, 43] |
| | ImageNet-21k→CIFAR-10 | 12 | [4, 8] | [44, 45] |
| | ImageNet-21k→CIFAR-100 | 12 | [4, 8] | [46, 47] |
| | ImageNet-1k | 12 | [4, 8, 16] | [48, 49, 50] |
| | WikiText103 | 12 | [4, 8, 16] | [51, 52, 53] |
| | One Billion Word | 12 | [4, 8, 16] | [54, 55, 56] |
| DeepSeekMoE ($k = 2, s = 1$) | MNIST | 1 | [4] | [57] |
| | | 2 | [4] | [58] |
| | CIFAR-10 | 2 | [4, 8] | [59, 60] |
| | | 6 | [4, 8] | [61, 62] |
| | CIFAR-100 | 6 | [4, 8] | [63, 64] |
| | ImageNet-21k→CIFAR-10 | 12 | [4, 8] | [65, 66] |
| | ImageNet-21k→CIFAR-100 | 12 | [4, 8] | [67, 68] |
| | ImageNet-1k | 12 | [4, 8, 16] | [69, 70, 71] |
| | WikiText103 | 12 | [4, 8, 16] | [72, 73, 74] |
| | One Billion Word | 12 | [4, 8, 16] | [75, 76, 77] |

### G.1.1 Dense Mixture-of-Experts

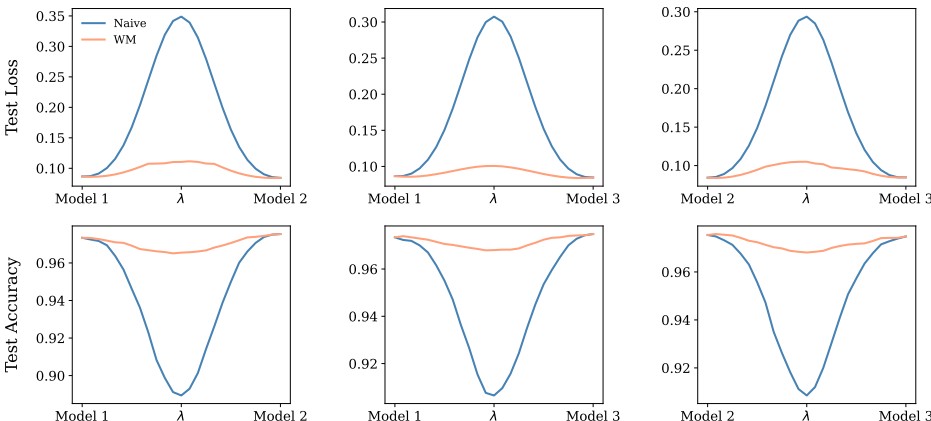

Figure 5: Linear Mode Connectivity for ViT-MoE on MNIST with 1 layer and 2 experts

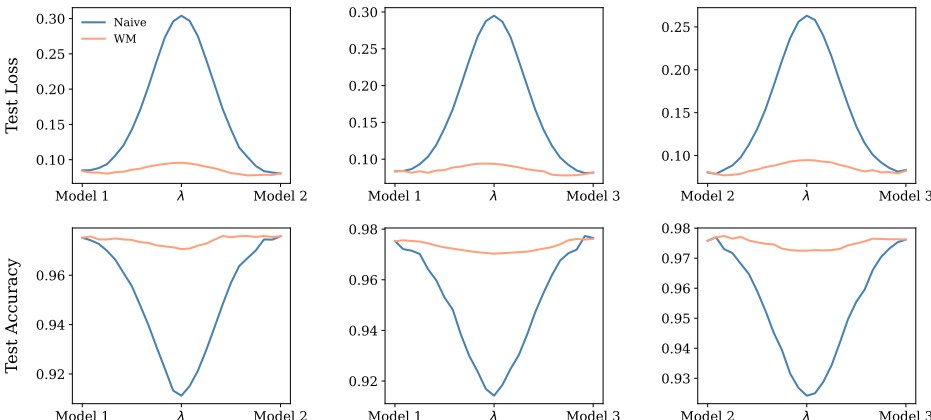

Figure 6: Linear Mode Connectivity for ViT-MoE on MNIST with 1 layer and 4 experts

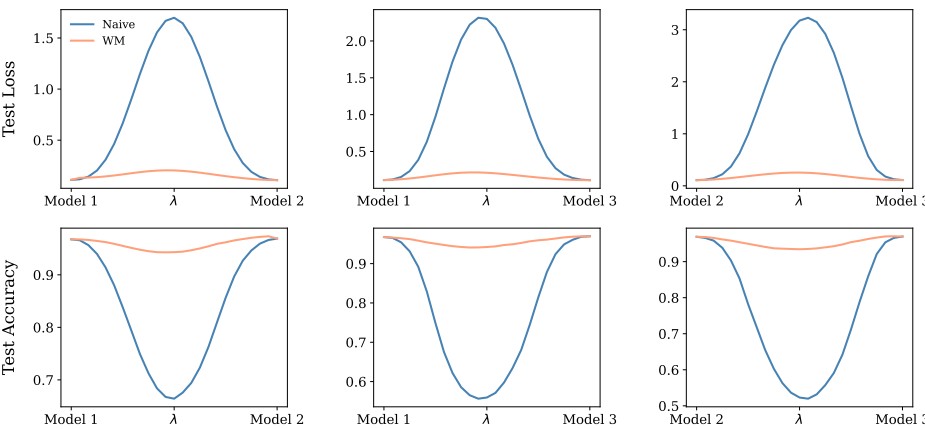

Figure 7: Linear Mode Connectivity for ViT-MoE on MNIST with 2 layers and 2 experts

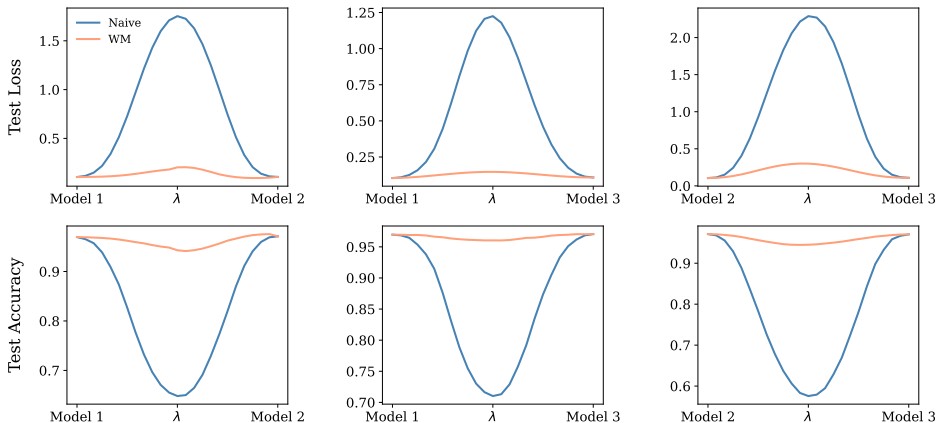

Figure 8: Linear Mode Connectivity for ViT-MoE on MNIST with 2 layers and 4 experts

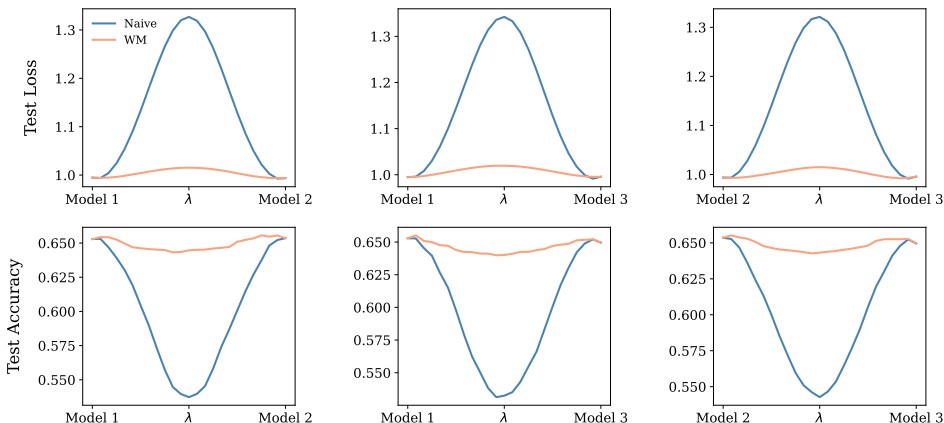

Figure 9: Linear Mode Connectivity for ViT-MoE on CIFAR-10 with 2 layers and 2 experts

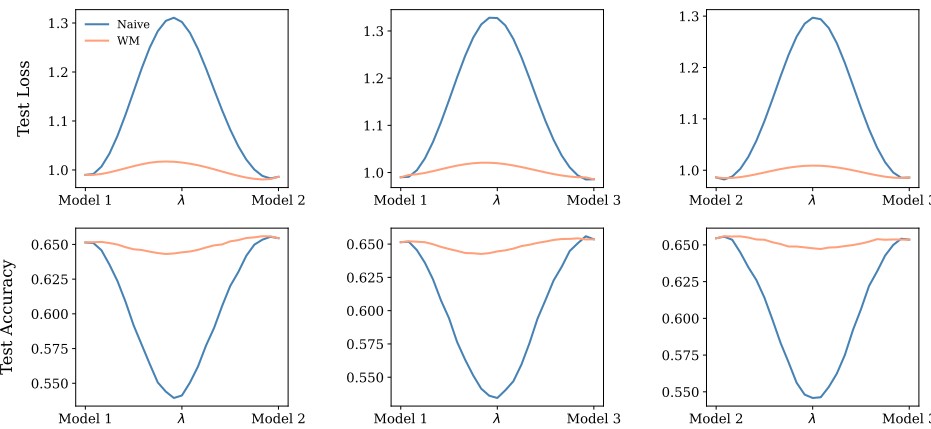

Figure 10: Linear Mode Connectivity for ViT-MoE on CIFAR-10 with 2 layers and 4 experts

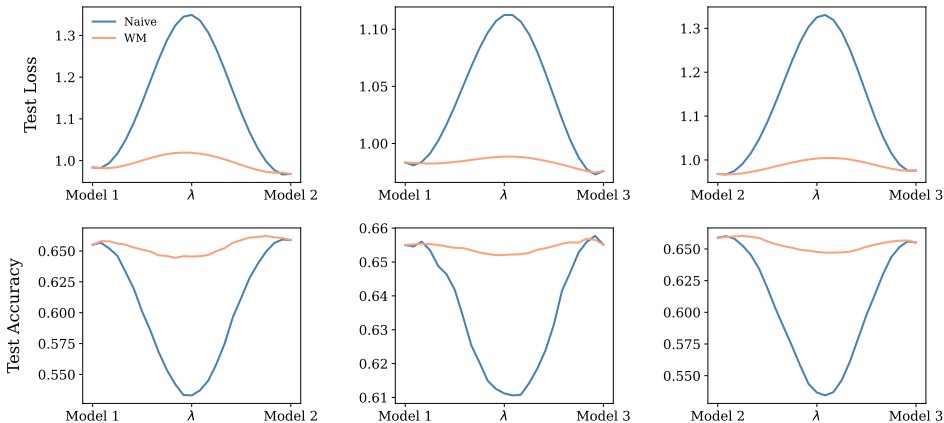

Figure 11: Linear Mode Connectivity for ViT-MoE on CIFAR-10 with 2 layers and 6 experts

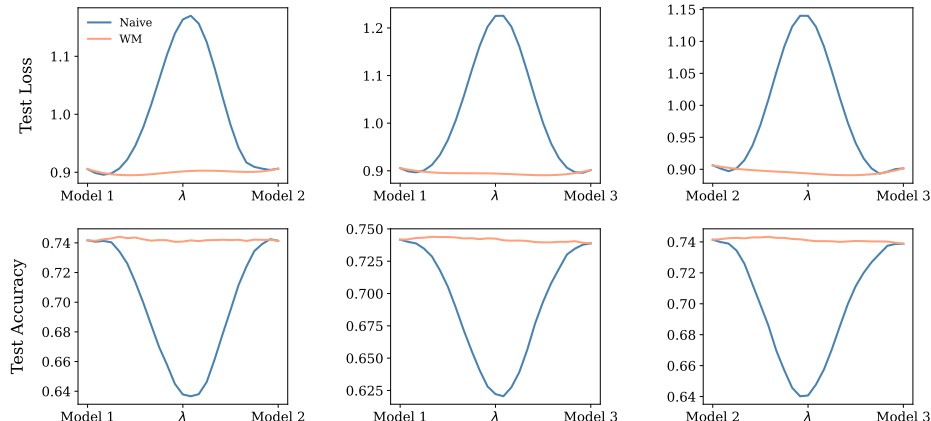

Figure 12: Linear Mode Connectivity for ViT-MoE on CIFAR-10 with 6 layers and 2 experts

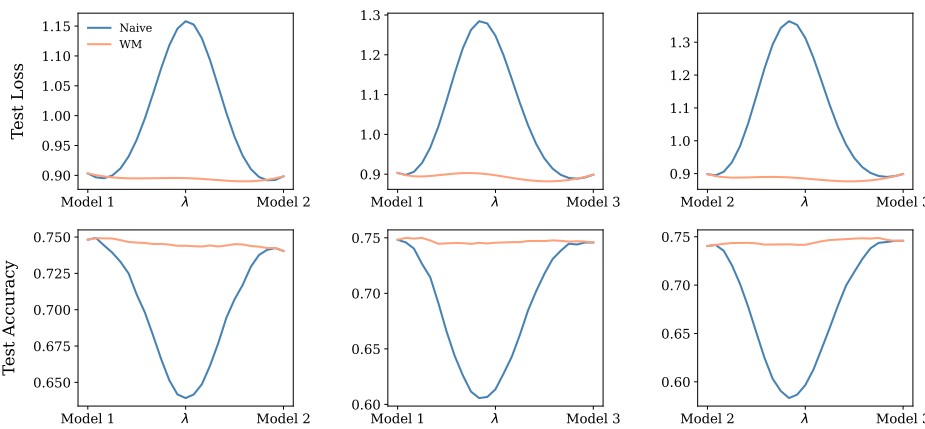

Figure 13: Linear Mode Connectivity for ViT-MoE on CIFAR-10 with 6 layers and 4 experts

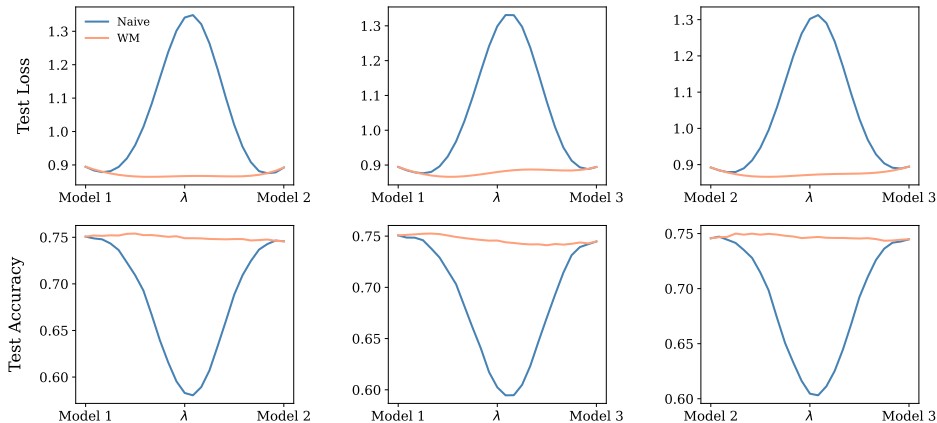

Figure 14: Linear Mode Connectivity for ViT-MoE on CIFAR-10 with 6 layers and 6 experts

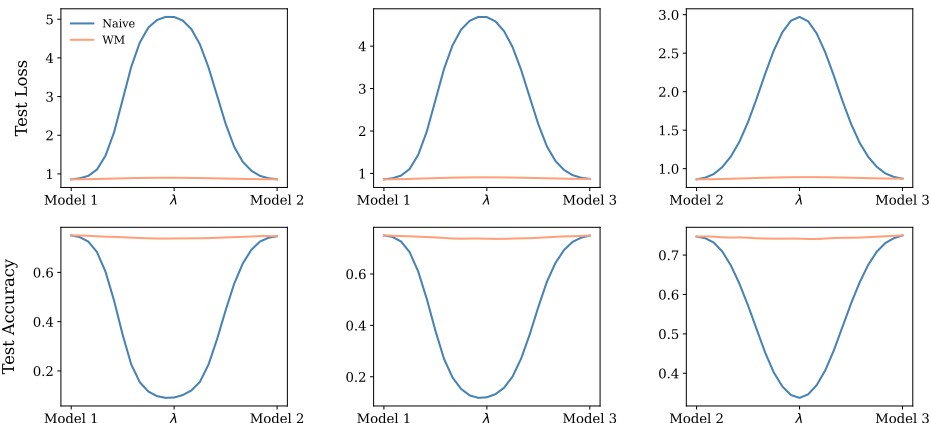

Figure 15: Linear Mode Connectivity for ViT-Moe on CIFAR-100 with 6 layers and 2 experts

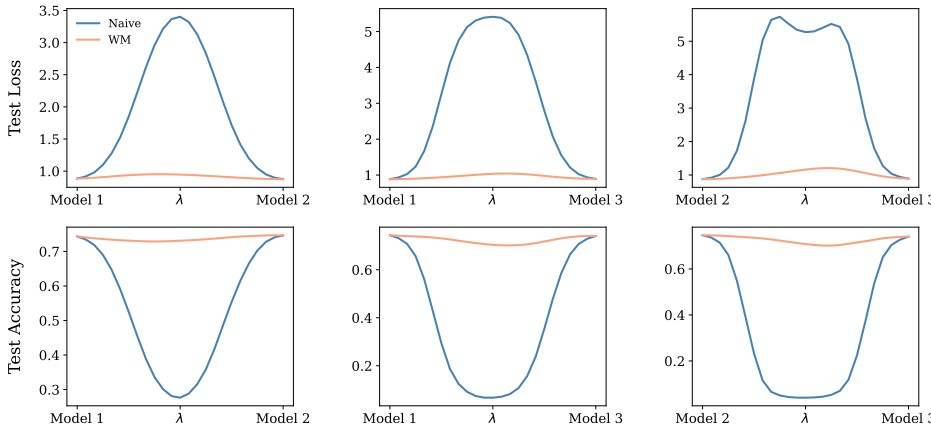

Figure 16: Linear Mode Connectivity for ViT-Moe on CIFAR-100 with 6 layers and 4 experts

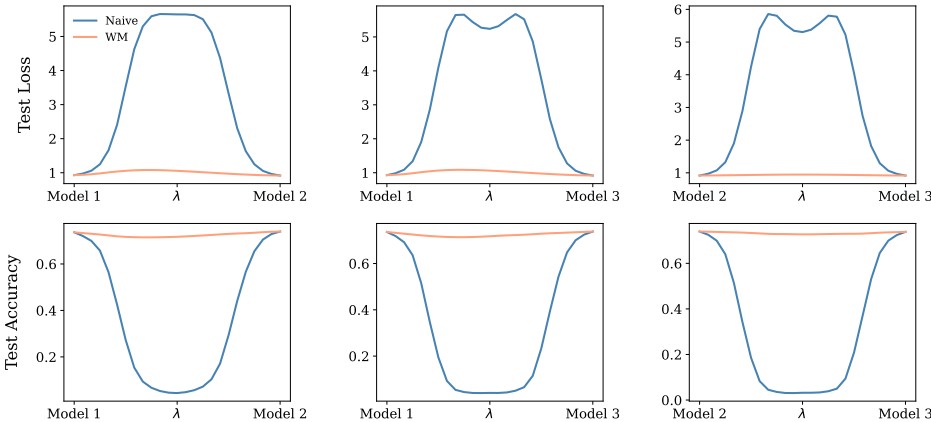

Figure 17: Linear Mode Connectivity for ViT-Moe on CIFAR-100 with 6 layers and 6 experts

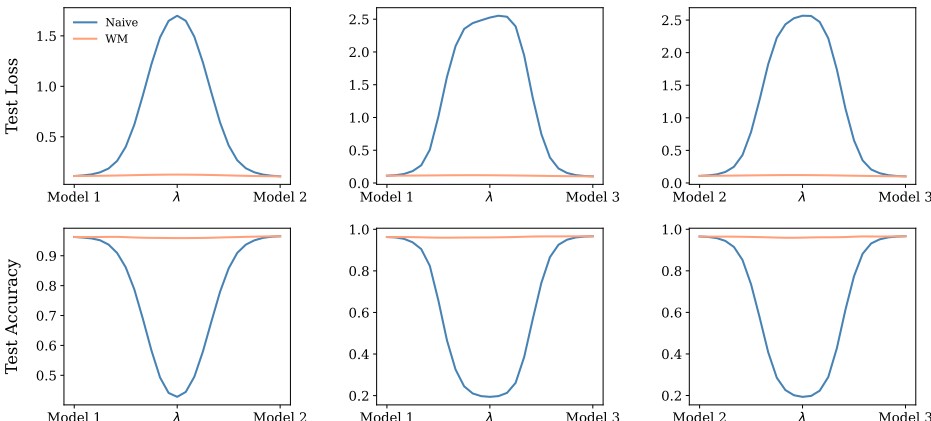

Figure 18: Linear Mode Connectivity for ViT-MoE on ImageNet-21k→CIFAR-10 with 12 layers and 2 experts

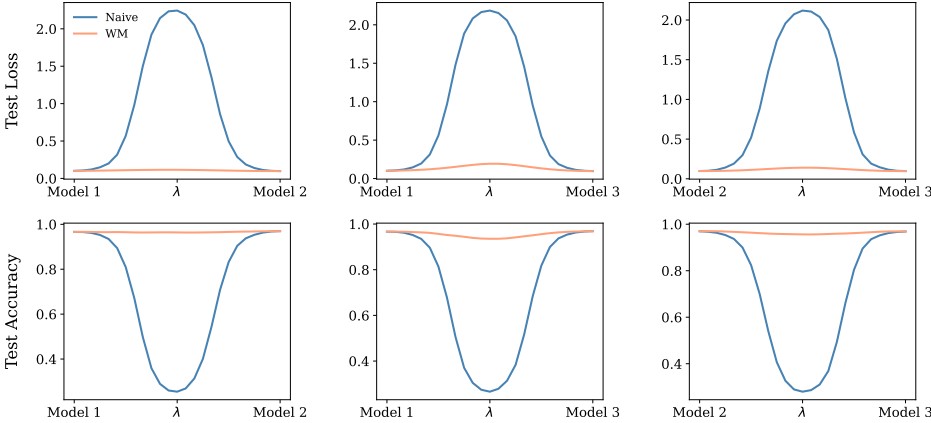

Figure 19: Linear Mode Connectivity for ViT-MoE on ImageNet-21k→CIFAR-10 with 12 layers and 4 experts

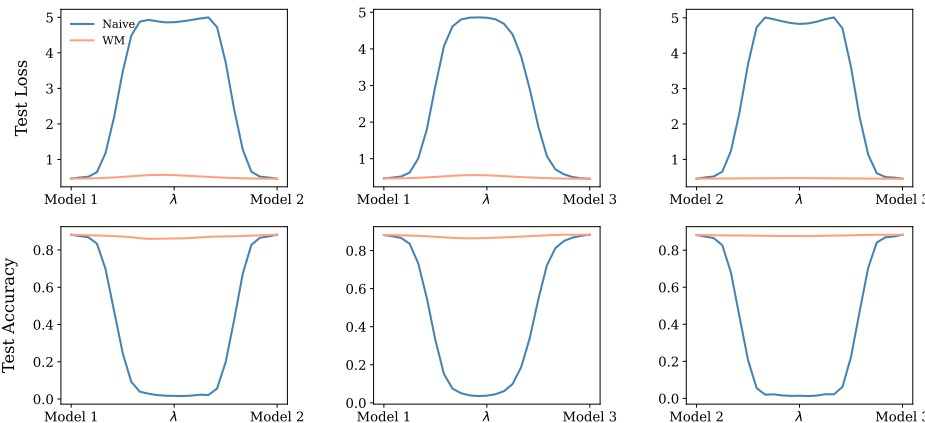

Figure 22: Linear Mode Connectivity for ViT-MoE on ImageNet-21k→CIFAR-100 with 12 layers and 4 experts

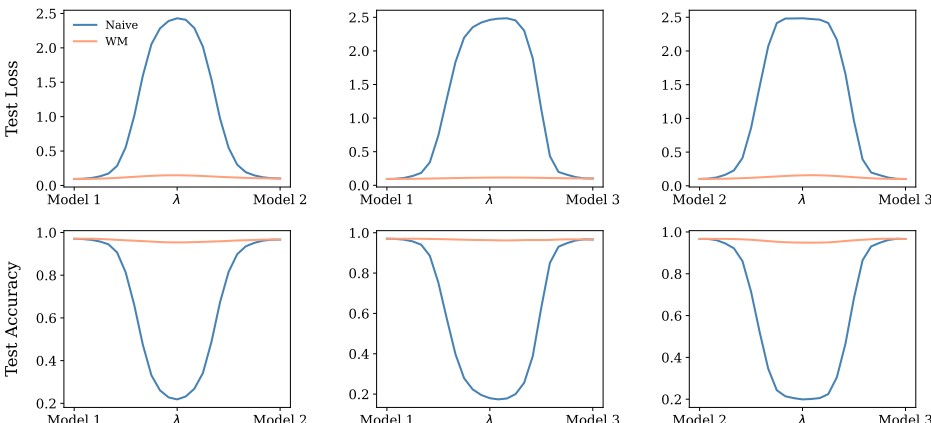

Figure 20: Linear Mode Connectivity for ViT-MoE on ImageNet-21k→CIFAR-100 with 12 layers and 6 experts

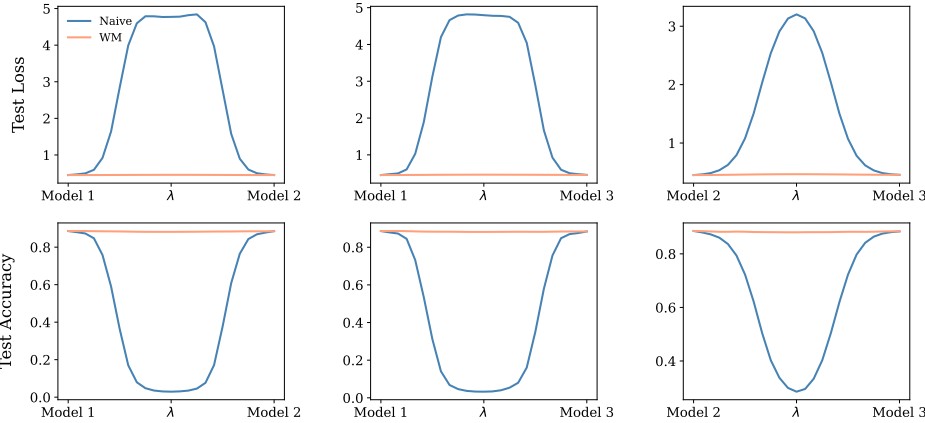

Figure 21: Linear Mode Connectivity for ViT-MoE on ImageNet-21k→CIFAR-100 with 12 layers and 2 experts

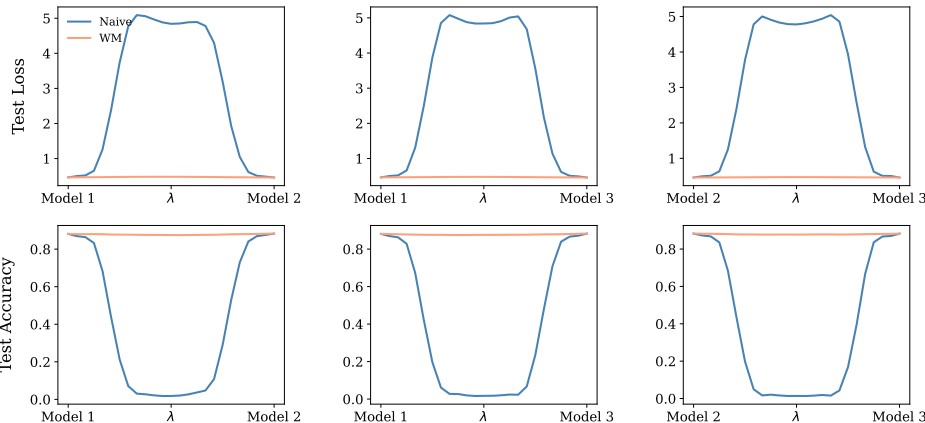

Figure 23: Linear Mode Connectivity for ViT-MoE with ImageNet-21k→CIFAR-100 with 12 layers and 6 experts

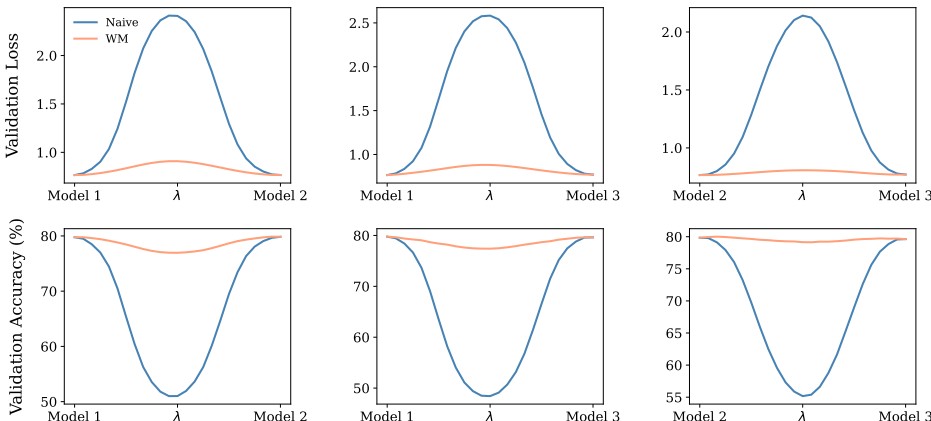

Figure 24: Linear Mode Connectivity for ViT-MoE on ImageNet-1k with 12 layers and 2 experts

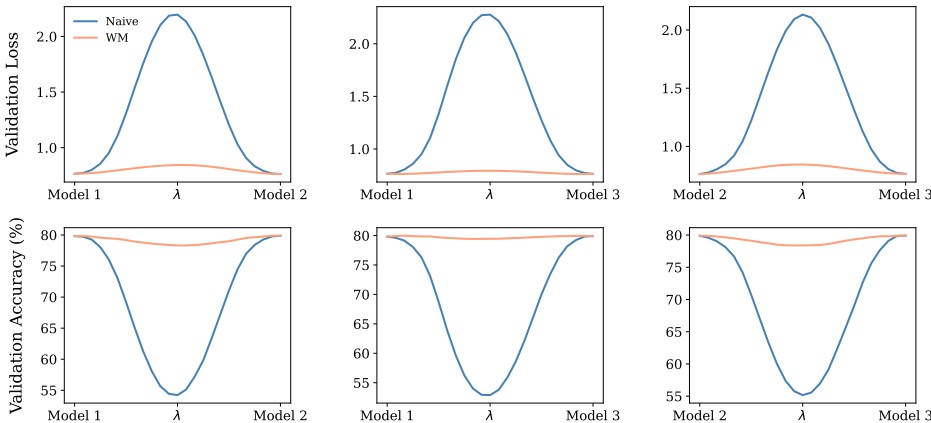

Figure 25: Linear Mode Connectivity for ViT-MoE on ImageNet-1k with 12 layers and 4 experts

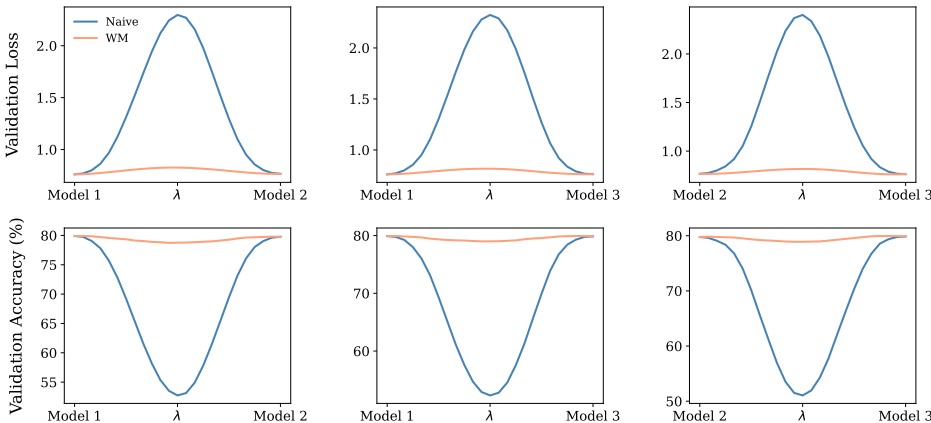

Figure 26: Linear Mode Connectivity for ViT-MoE on ImageNet-1k with 12 layers and 6 experts

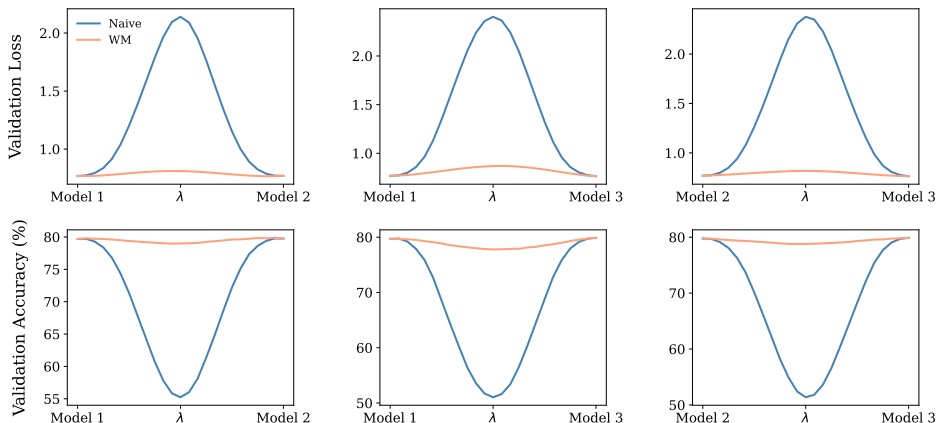

Figure 27: Linear Mode Connectivity for ViT-MoE on ImageNet-1k with 12 layers and 8 experts

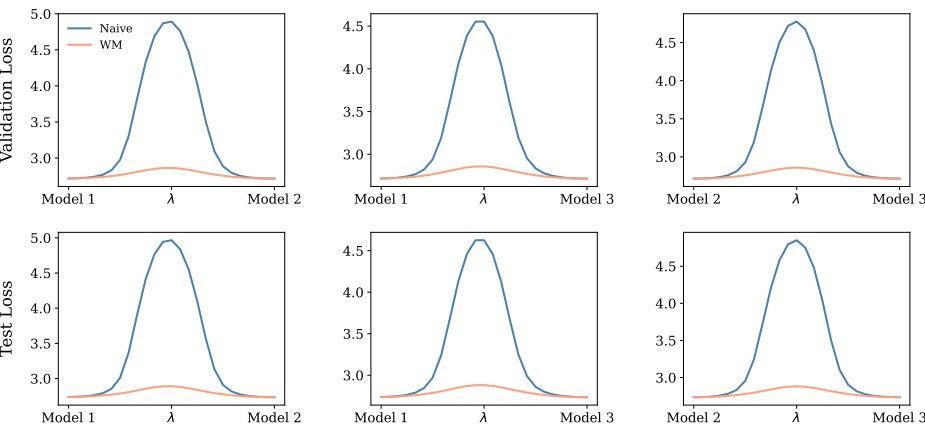

Figure 28: Linear Mode Connectivity for GPT2-MoE on Wikitext103 with 12 layers and 2 experts

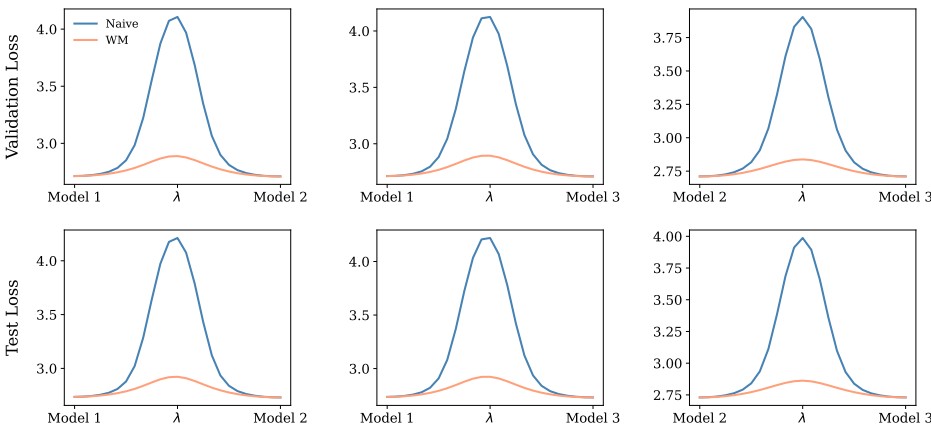

Figure 29: Linear Mode Connectivity for GPT2-MoE on Wikitext103 with 12 layers and 4 experts

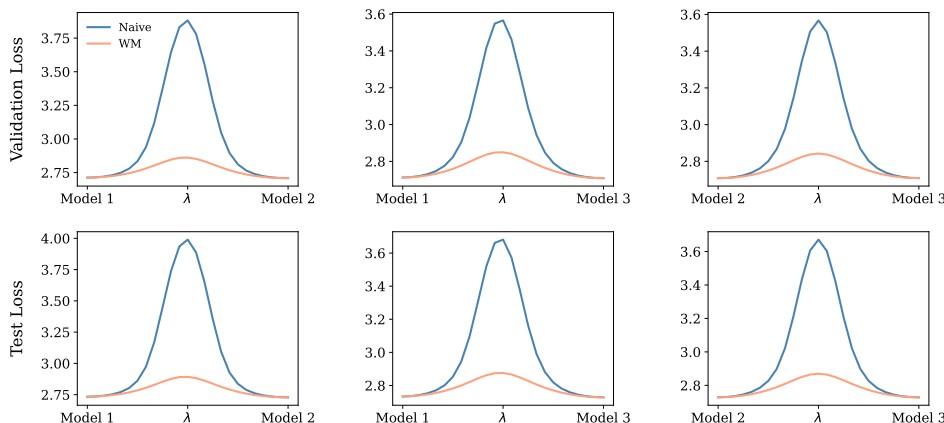

Figure 30: Linear Mode Connectivity for GPT2-MoE on Wikitext103 with 12 layers and 6 experts

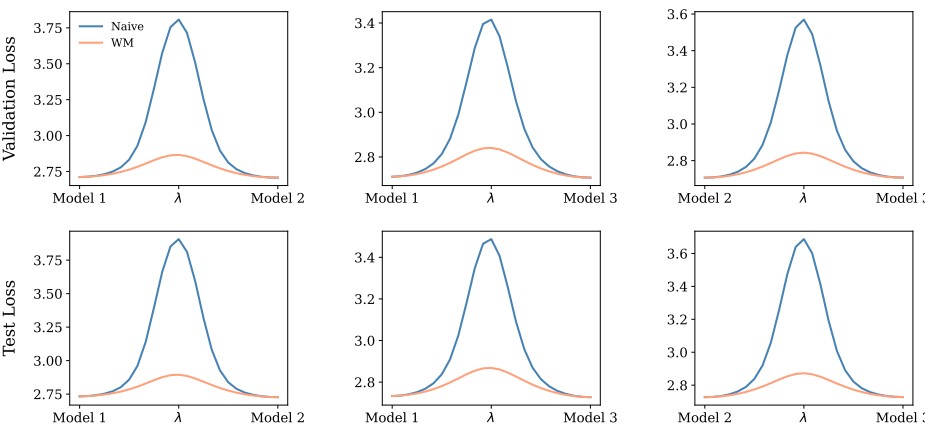

Figure 31: Linear Mode Connectivity for GPT2-MoE on Wikitext103 with 12 layers and 8 experts

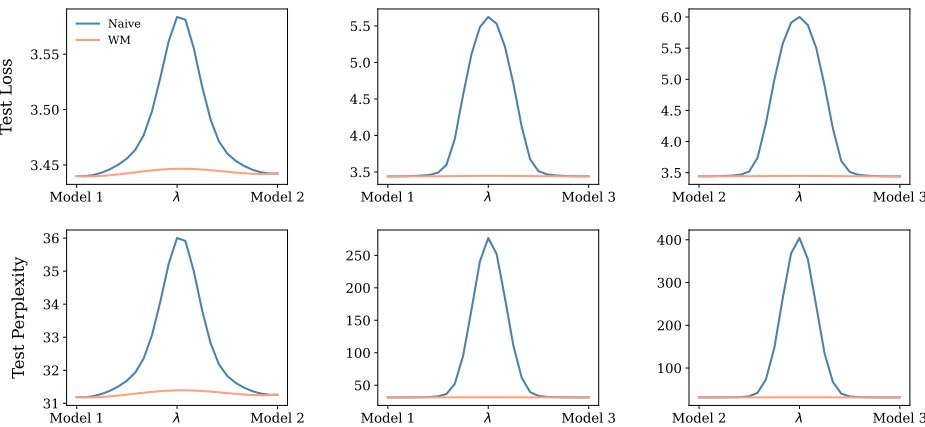

Figure 32: Linear Mode Connectivity for GPT2-MoE on One Billion Word with 12 layers and 2 experts

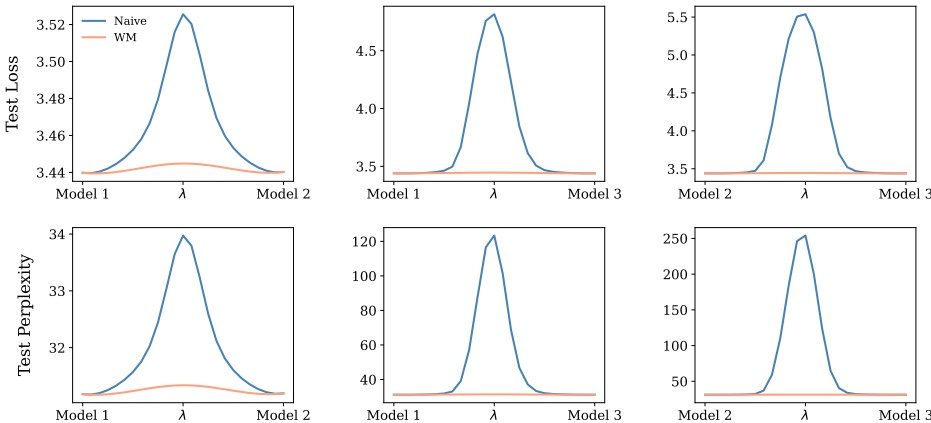

Figure 33: Linear Mode Connectivity for GPT2-MoE on One Billion Word with 12 layers and 4 experts

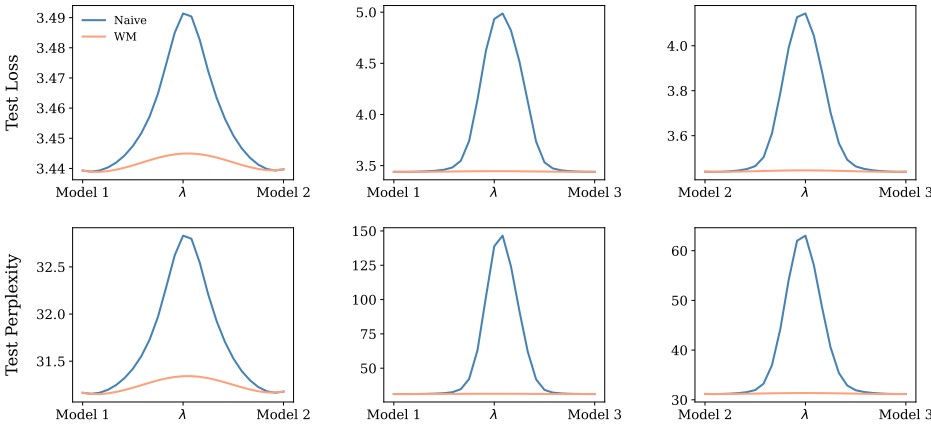

Figure 34: Linear Mode Connectivity for GPT2-MoE on One Billion Word with 12 layers and 6 experts

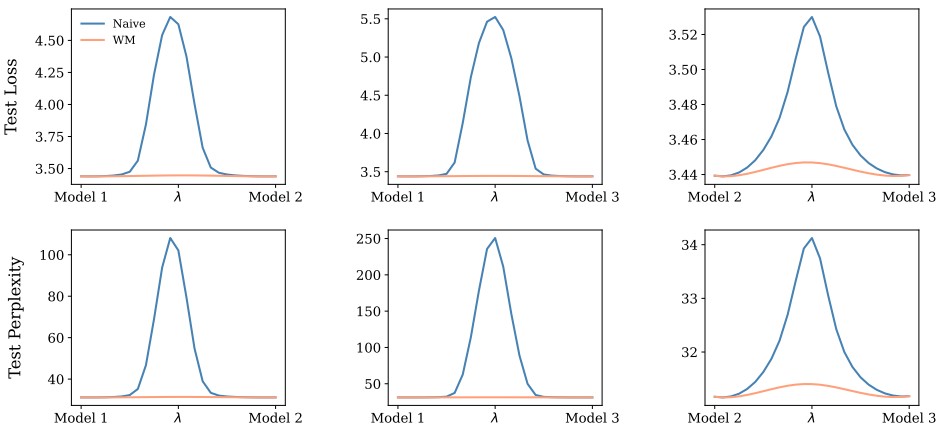

Figure 35: Linear Mode Connectivity for GPT2-MoE on One Billion Word with 12 layers and 8 experts

### G.1.2 Sparse Mixture-of-Experts

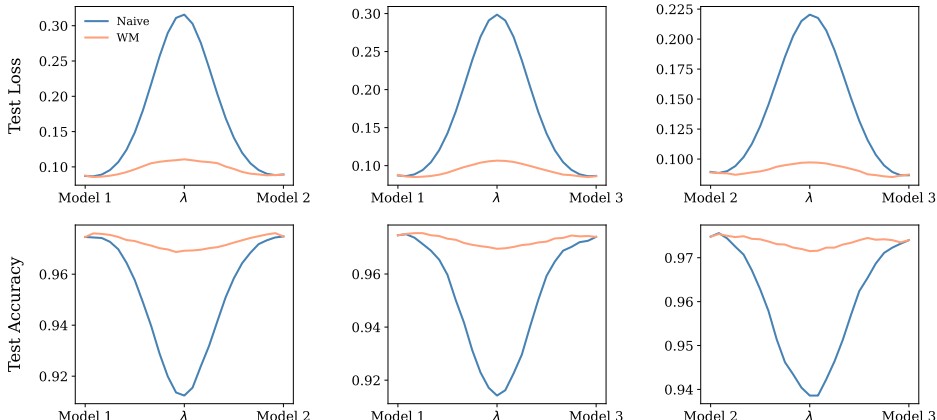

Figure 36: Linear Mode Connectivity for ViT-SMoE ($k = 2$) on MNIST with 1 layer and 4 experts

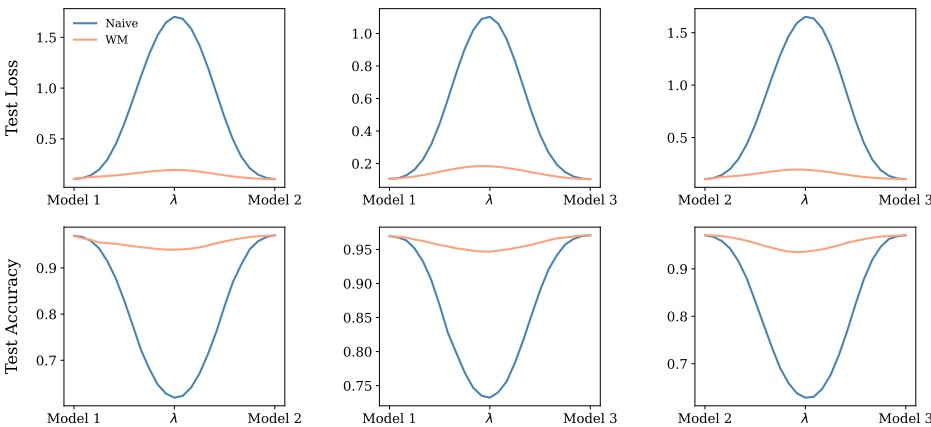

Figure 37: Linear Mode Connectivity for ViT-SMoE ($k = 2$) on MNIST with 2 layers and 4 experts

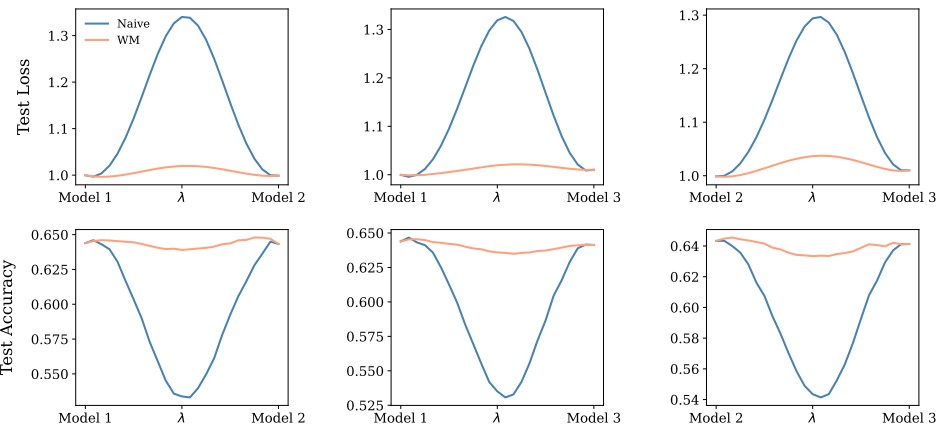

Figure 38: Linear Mode Connectivity for ViT-SMoE ($k = 2$) on CIFAR-10 with 2 layers and 4 experts

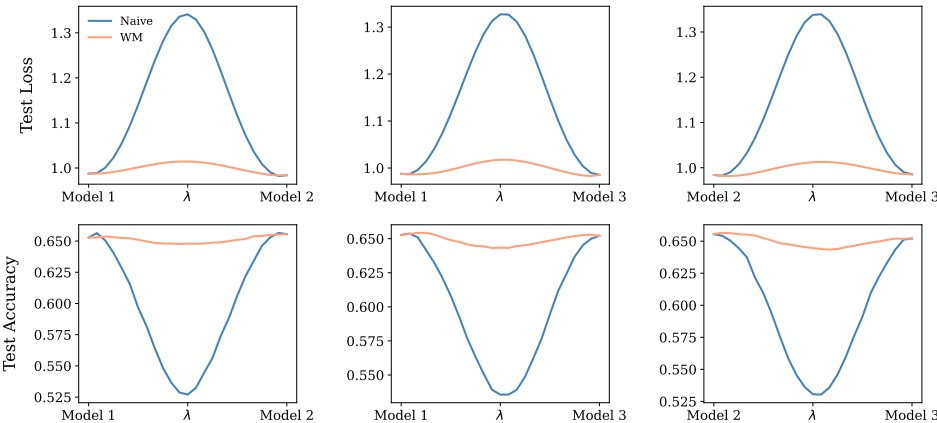

Figure 39: Linear Mode Connectivity for ViT-SMoE ($k = 2$) on CIFAR-10 with 2 layers and 8 experts

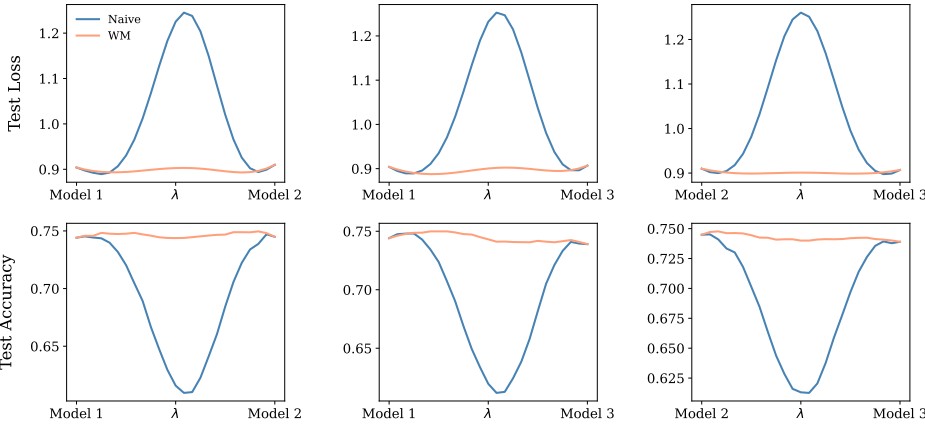

Figure 40: Linear Mode Connectivity for ViT-SMoE ($k = 2$) on CIFAR-10 with 6 layers and 4 experts

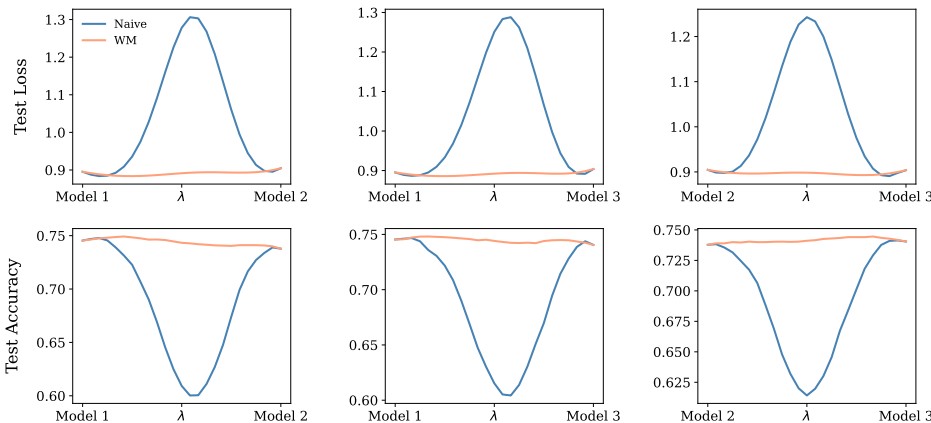

Figure 41: Linear Mode Connectivity for ViT-SMoE ($k = 2$) on CIFAR-10 with 6 layers and 8 experts

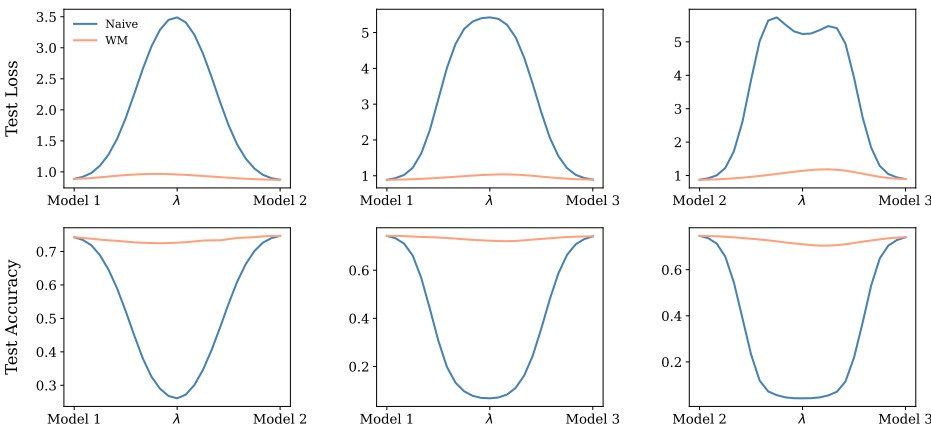

Figure 42: Linear Mode Connectivity for ViT-SMoE ($k = 2$) on CIFAR-100 with 6 layers and 4 experts

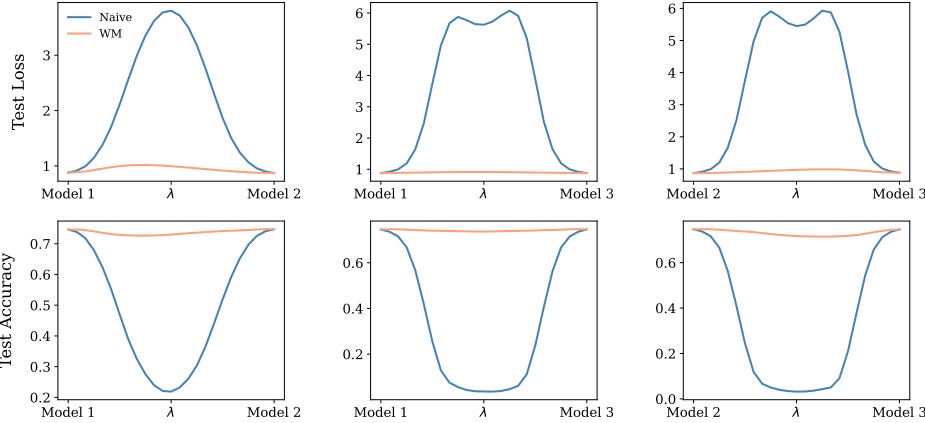

Figure 43: Linear Mode Connectivity for ViT-SMoE ($k = 2$) on CIFAR-100 with 6 layers and 8 experts

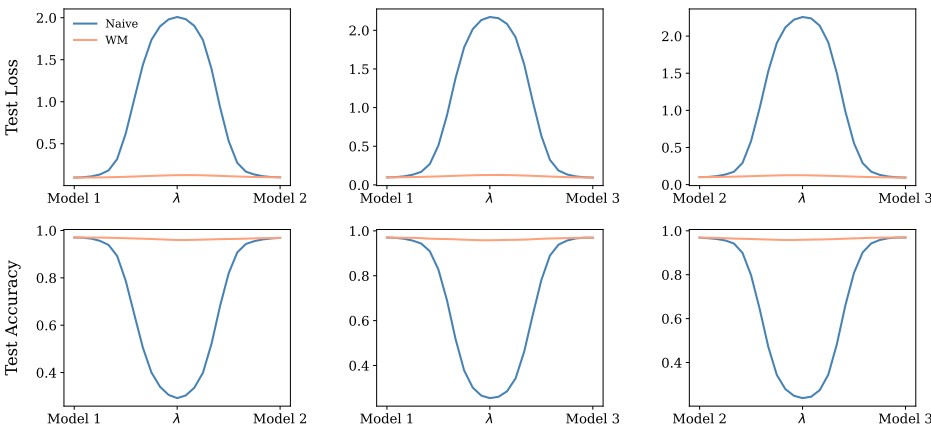

Figure 44: Linear Mode Connectivity for ViT-SMoE ($k = 2$) on ImageNet-21k→CIFAR-10 with 12 layers and 4 experts

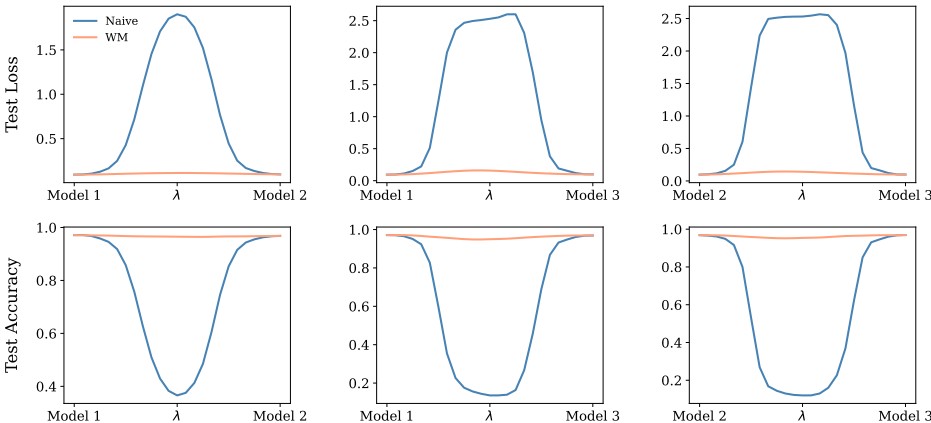

Figure 45: Linear Mode Connectivity for ViT-SMoE ($k = 2$) on ImageNet-21k→CIFAR-10 with 12 layers and 8 experts

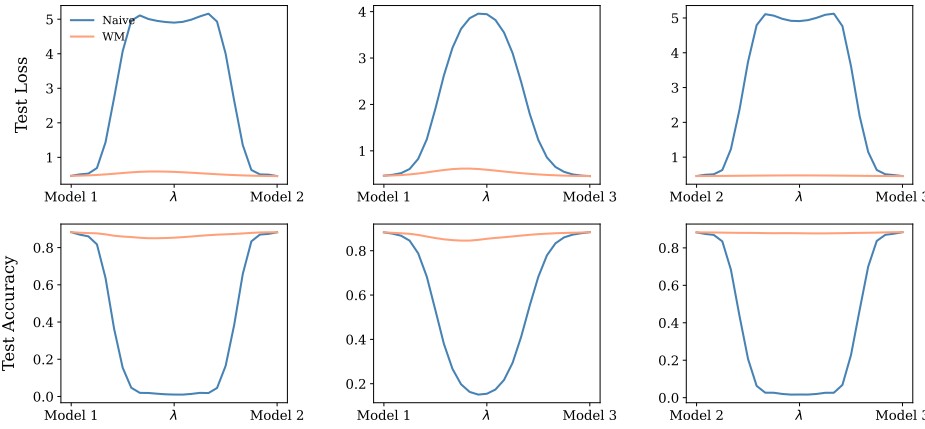

Figure 46: Linear Mode Connectivity for ViT-SMoE ($k = 2$) on ImageNet-21k→CIFAR-100 with 12 layers and 4 experts

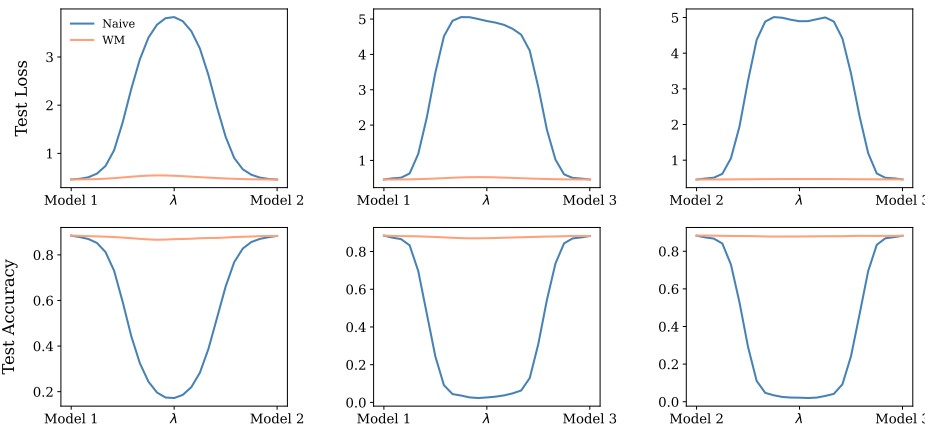

Figure 47: Linear Mode Connectivity for ViT-SMoE ($k = 2$) on ImageNet-21k→CIFAR-100 with 12 layers and 8 experts

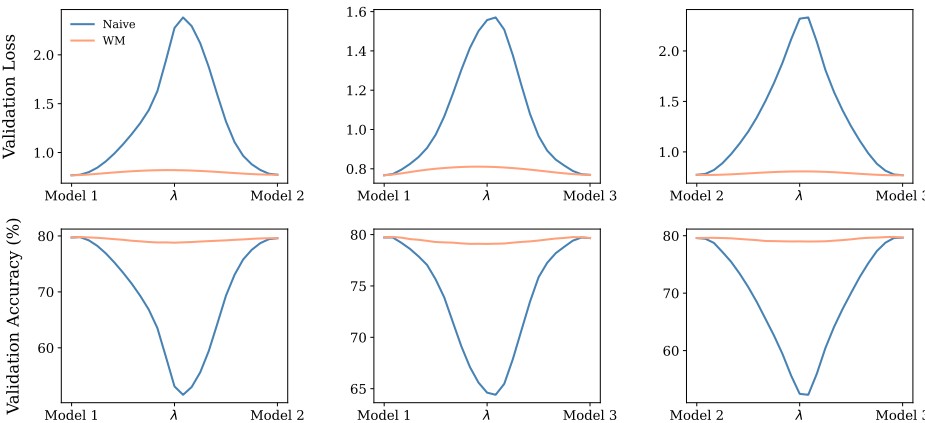

Figure 48: Linear Mode Connectivity for ViT-SMoE ($k = 2$) on ImageNet-1k with 12 layers and 4 experts

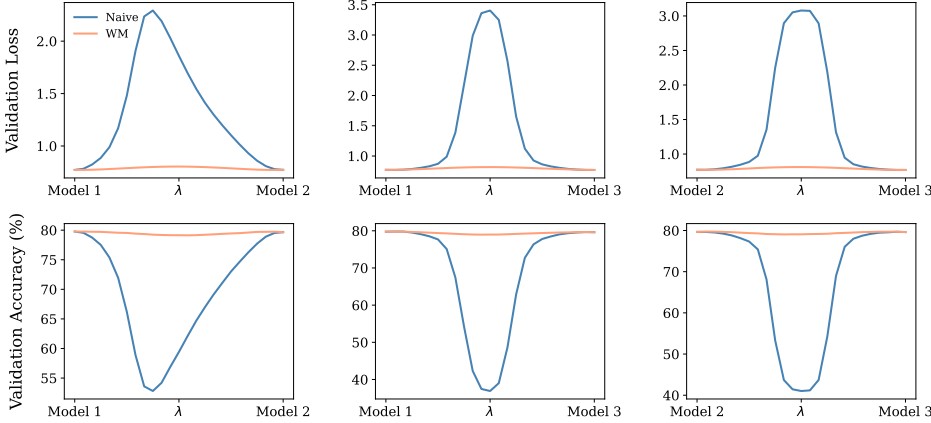

Figure 49: Linear Mode Connectivity for ViT-SMoE ($k = 2$) on ImageNet-1k with 12 layers and 8 experts

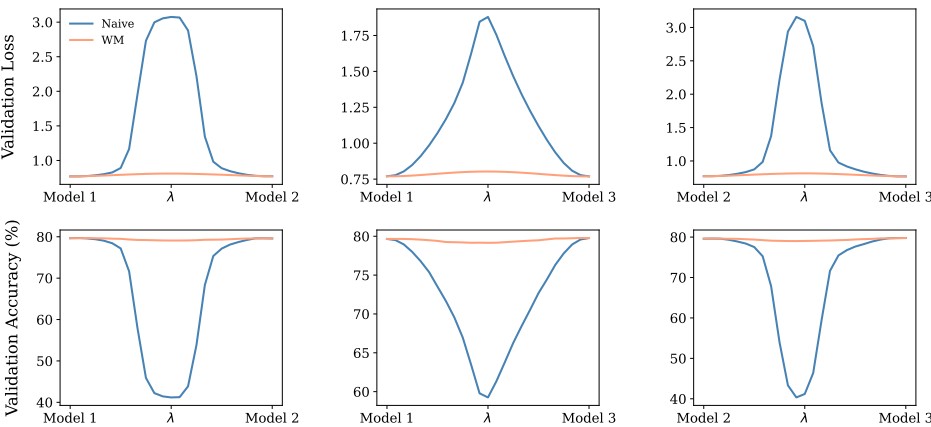

Figure 50: Linear Mode Connectivity for ViT-SMoE ($k = 2$) on ImageNet-1k with 12 layers and 16 experts

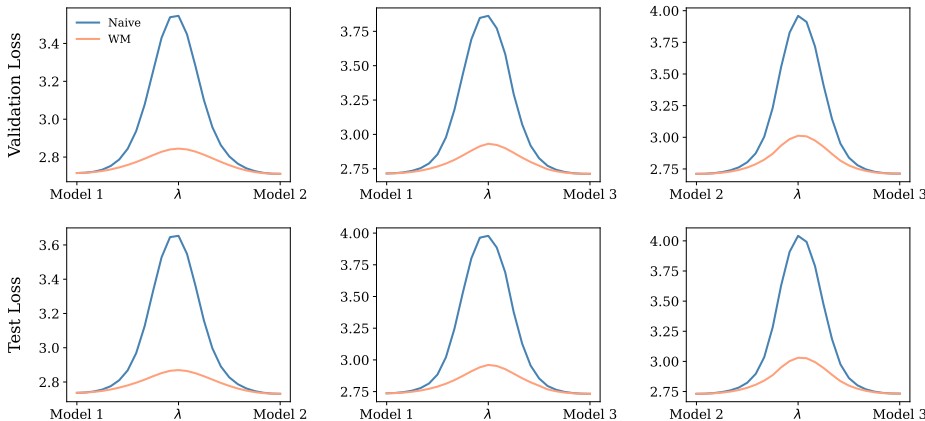

Figure 51: Linear Mode Connectivity for GPT2-SMoE ($k = 2$) on Wikitext103 with 12 layers and 4 experts

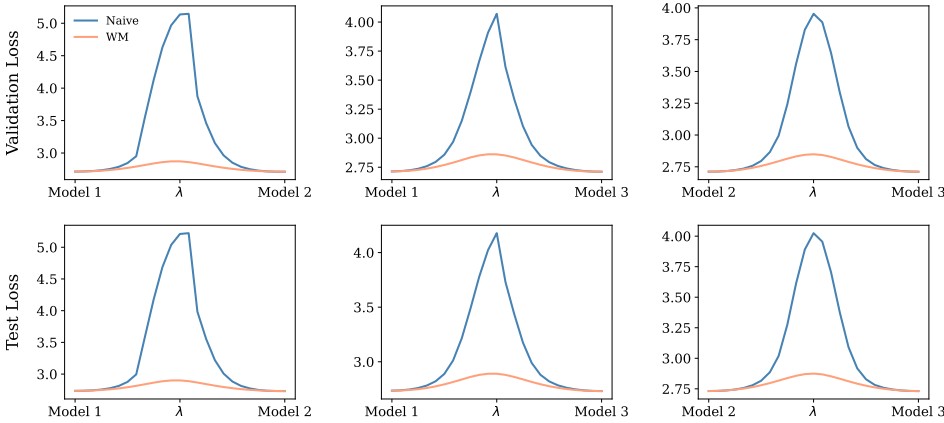

Figure 52: Linear Mode Connectivity for GPT2-SMoE ($k = 2$) on Wikitext103 with 12 layers and 8 experts

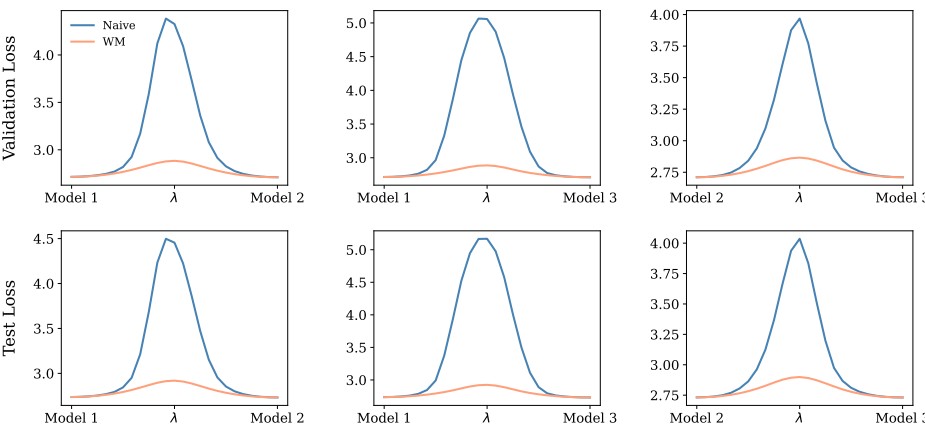

Figure 53: Linear Mode Connectivity for GPT2-SMoE ($k = 2$) on Wikitext103 with 12 layers 16 experts

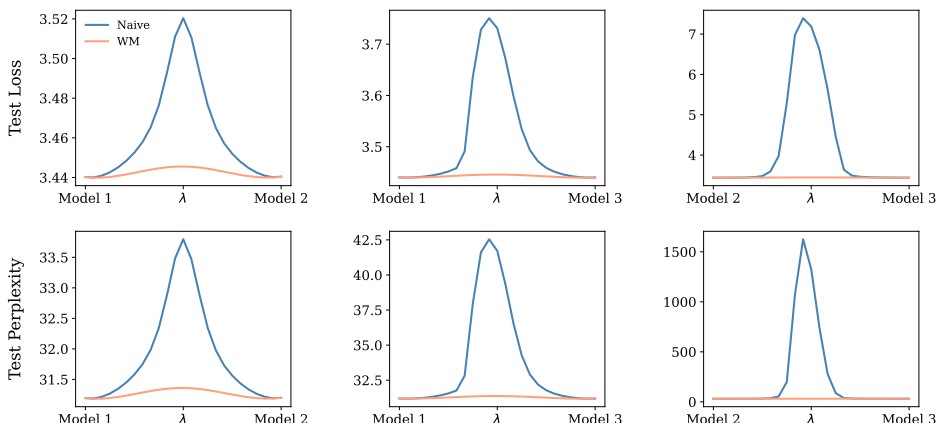

Figure 54: Linear Mode Connectivity for GPT2-SMoE ($k = 2$) on One Billion Word with 12 layers and 4 experts

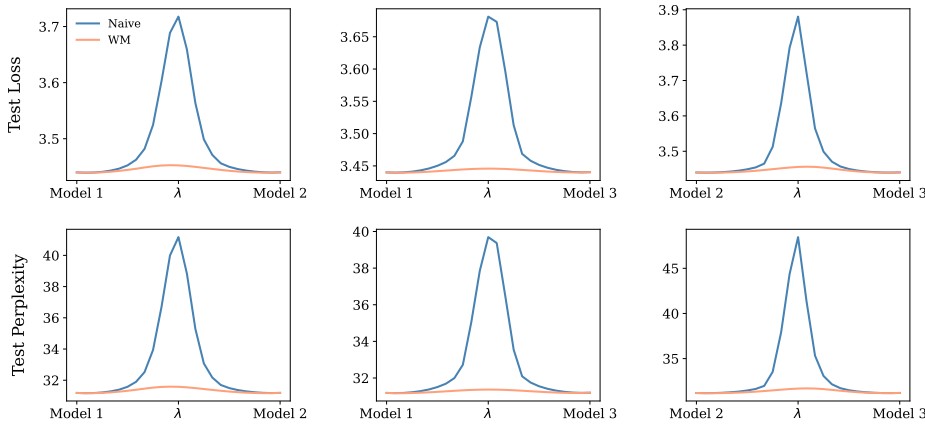

Figure 55: Linear Mode Connectivity for GPT2-SMoE ($k = 2$) on One Billion Word with 12 layers and 8 experts

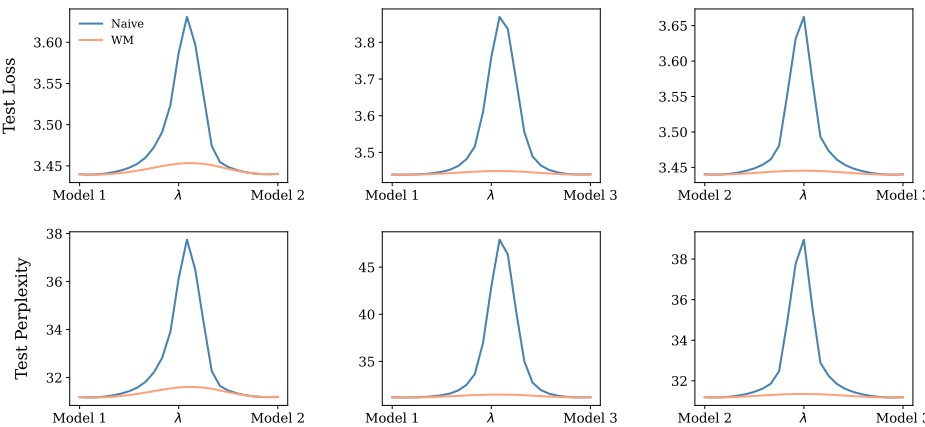

Figure 56: Linear Mode Connectivity for GPT2-SMoE ($k = 2$) on One Billion Word with 12 layers and 16 experts

### G.1.3 DeepSeek Mixture-of-Experts

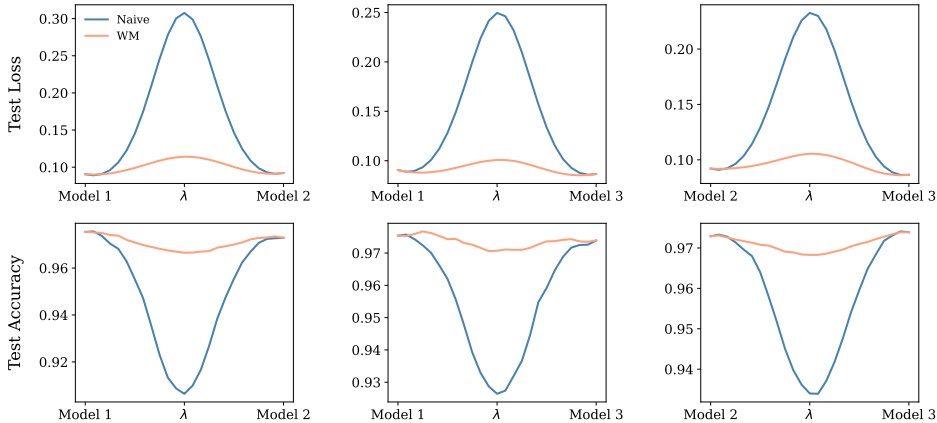

Figure 57: Linear Mode Connectivity for ViT-DeepSeekMoE ($k = 2, s = 1$) on MNIST with 1 layer and 4 experts

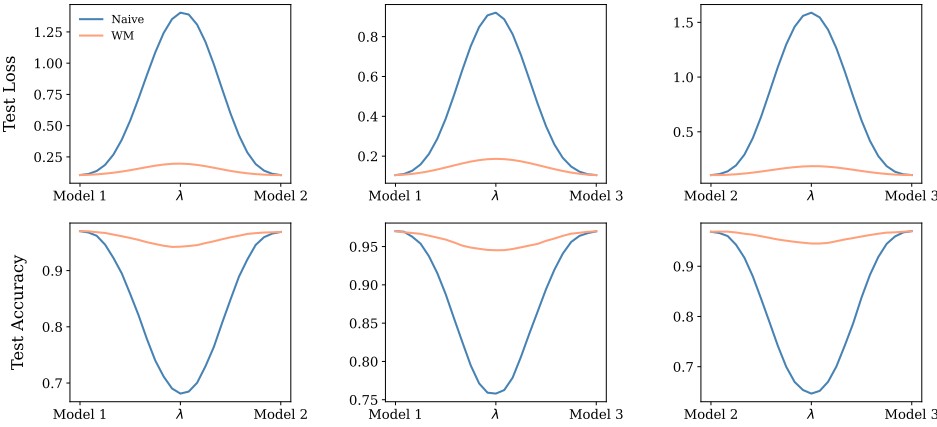

Figure 58: Linear Mode Connectivity for ViT-DeepSeekMoE ($k = 2, s = 1$) on MNIST with 2 layers and 4 experts

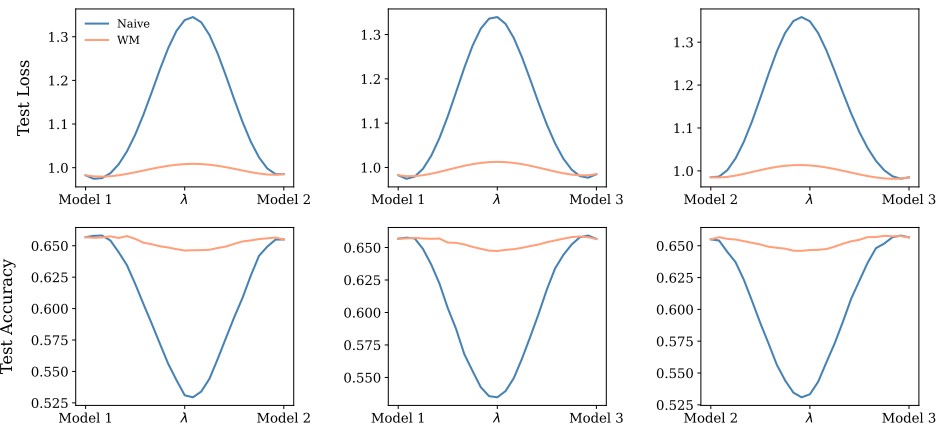

Figure 59: Linear Mode Connectivity for ViT-DeepSeekMoE ($k = 2, s = 1$) on CIFAR-10 with 2 layers and 4 experts

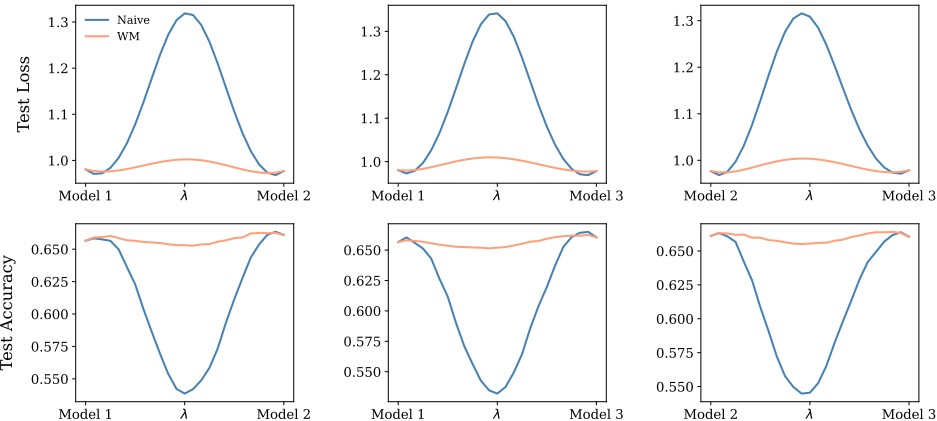

Figure 60: Linear Mode Connectivity for ViT-DeepSeekMoE ($k = 2, s = 1$) on CIFAR-10 with 2 layers and 8 experts

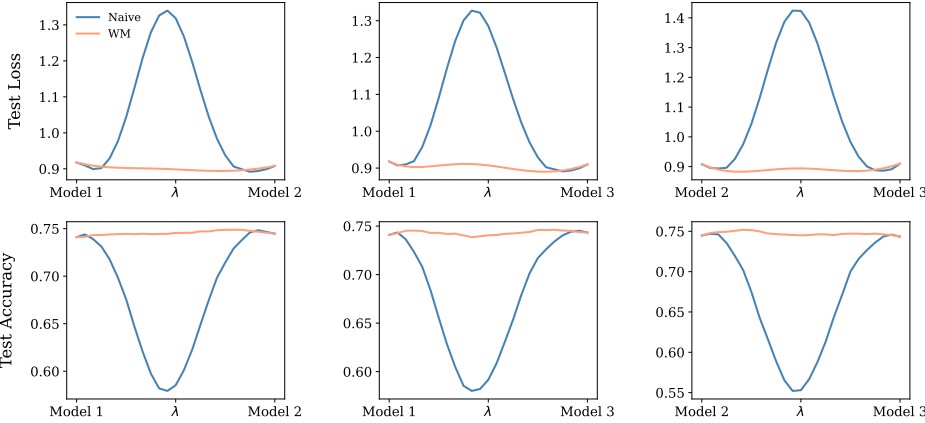

Figure 61: Linear Mode Connectivity for ViT-DeepSeekMoE ($k = 2, s = 1$) on CIFAR-10 with 6 layers and 4 experts

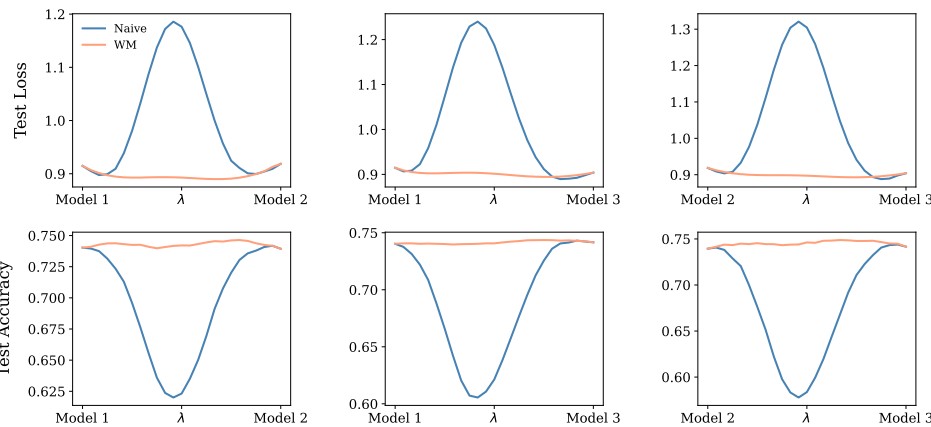

Figure 62: Linear Mode Connectivity for ViT-DeepSeekMoE ($k = 2, s = 1$) on CIFAR-10 with 6 layers and 8 experts

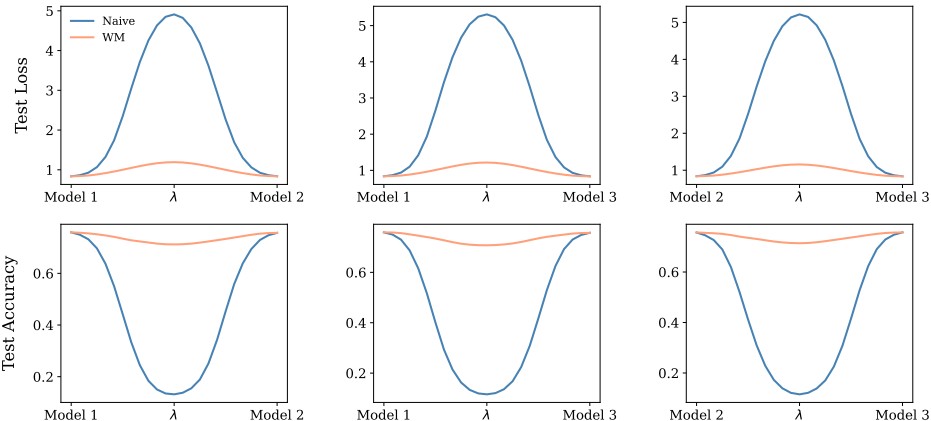

Figure 63: Linear Mode Connectivity for ViT-DeepSeekMoE ($k = 2, s = 1$) on CIFAR-100 with 6 layers and 4 experts

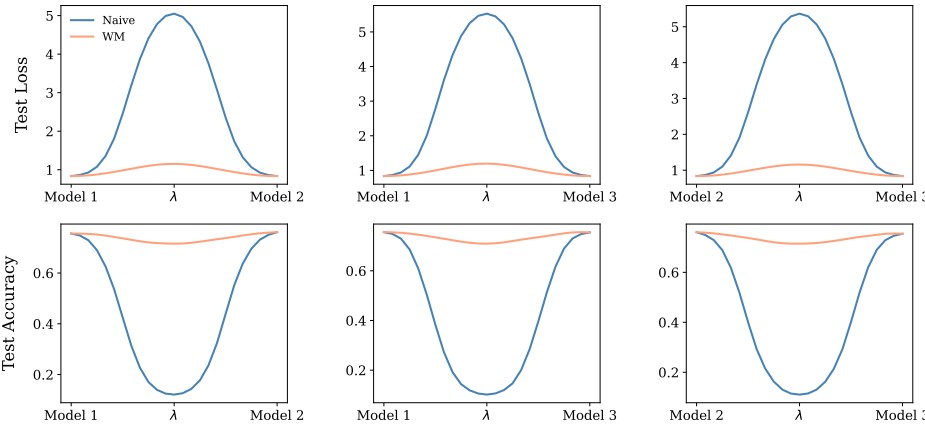

Figure 64: Linear Mode Connectivity for ViT-DeepSeekMoE ($k = 2, s = 1$) on CIFAR-100 with 6 layers and 8 experts

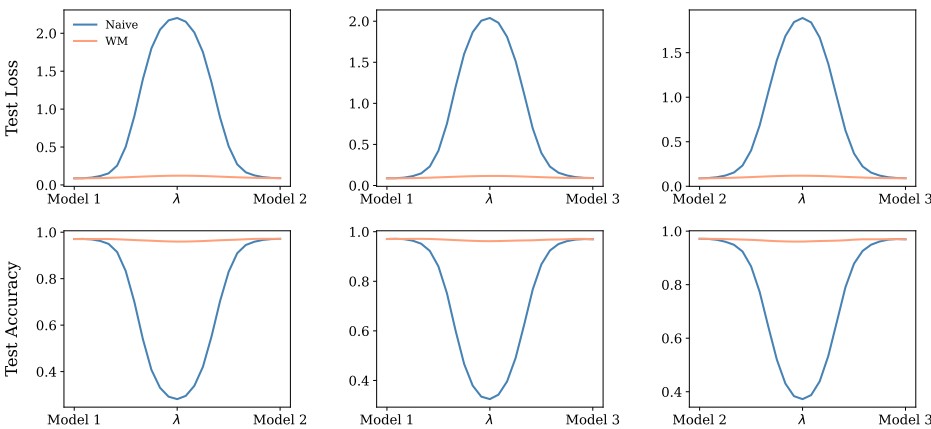

Figure 66: Linear Mode Connectivity for ViT-DeepSeekMoE ($k = 2, s = 1$) on ImageNet-21k→CIFAR-10 with 12 layers and 8 experts

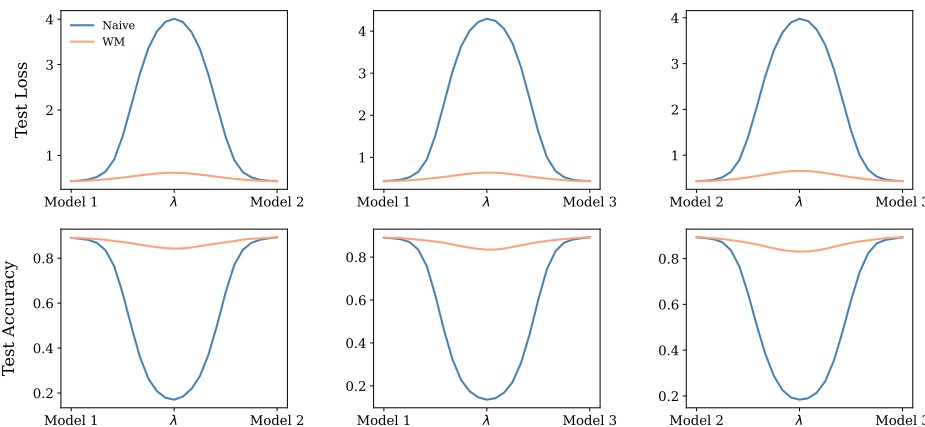

Figure 67: Linear Mode Connectivity for ViT-DeepSeekMoE ($k = 2, s = 1$) on ImageNet-21k→CIFAR-100 with 12 layers and 4 experts

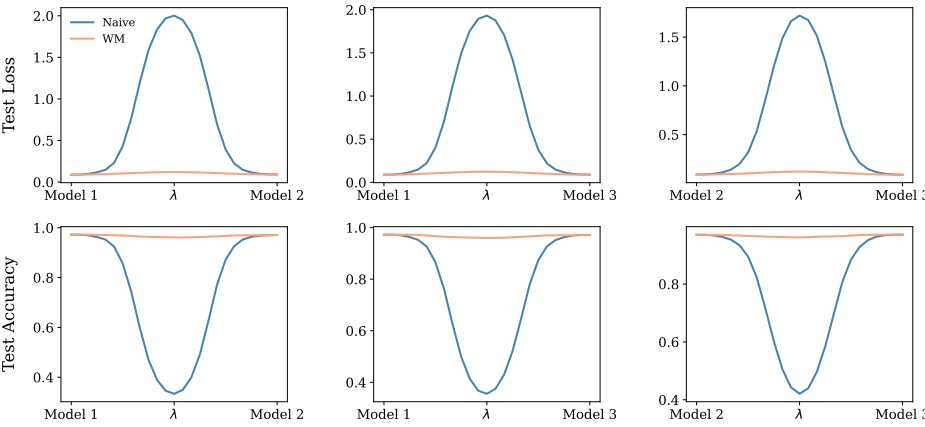

Figure 65: Linear Mode Connectivity for ViT-DeepSeekMoE ($k = 2, s = 1$) on ImageNet-21k→CIFAR-10 with 12 layers and 4 experts

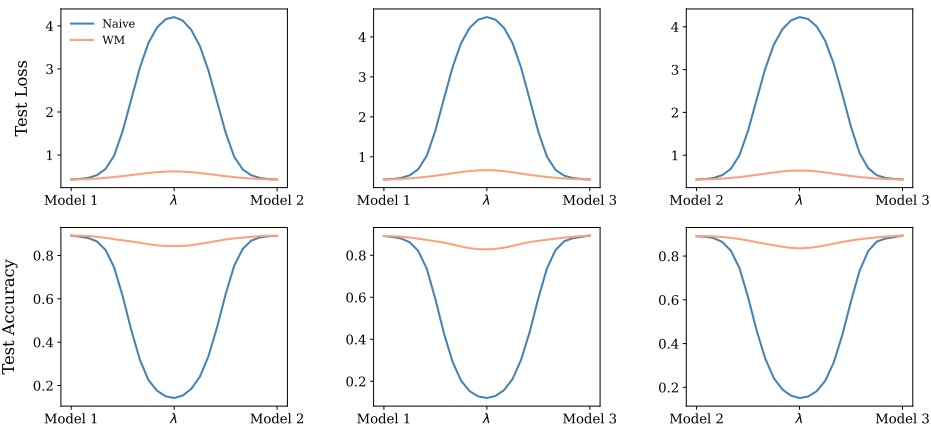

Figure 68: Linear Mode Connectivity for ViT-DeepSeekMoE ($k = 2, s = 1$) on ImageNet-21k→CIFAR-100 with 12 layers and 8 experts

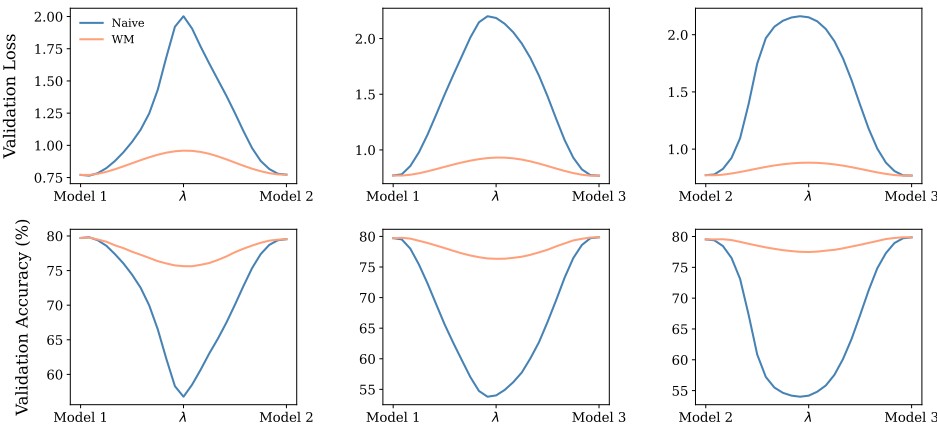

Figure 69: Linear Mode Connectivity for ViT-DeepSeekMoE ($k = 2, s = 1$) on ImageNet-1k with 12 layers and 4 experts

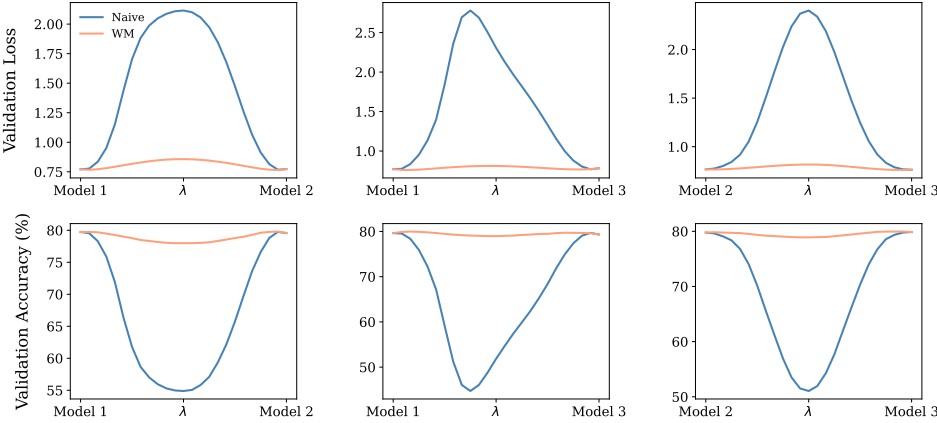

Figure 70: Linear Mode Connectivity for ViT-DeepSeekMoE ($k = 2, s = 1$) on ImageNet-1k with 12 layers and 8 experts

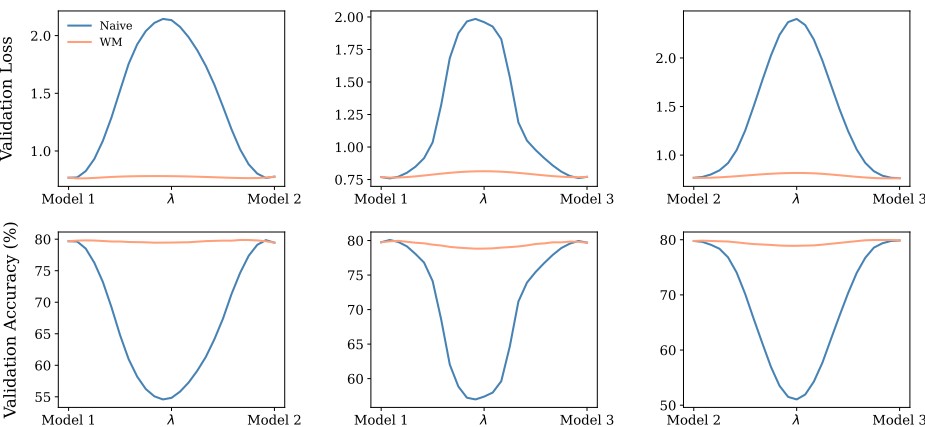

Figure 71: Linear Mode Connectivity for ViT-DeepSeekMoE ($k = 2, s = 1$) on ImageNet-1k with 12 layers and 16 experts

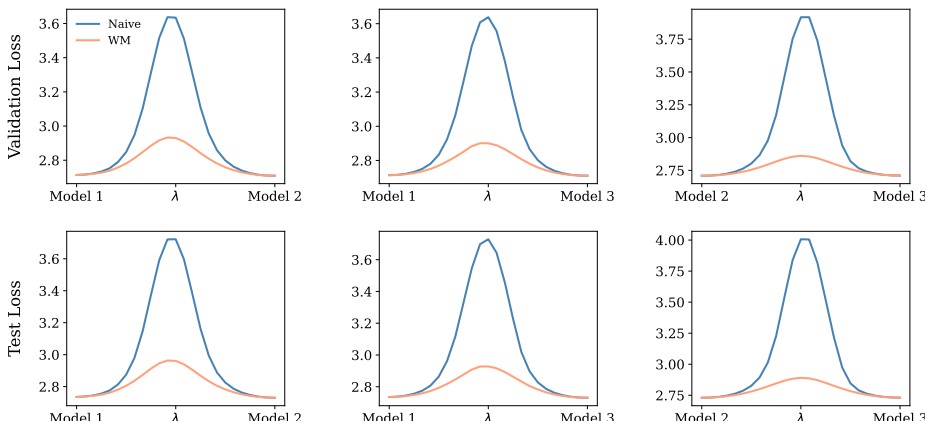

Figure 72: Linear Mode Connectivity for ViT-DeepSeekMoE ($k = 2, s = 1$) on Wikitext103 with 12 layers and 4 experts

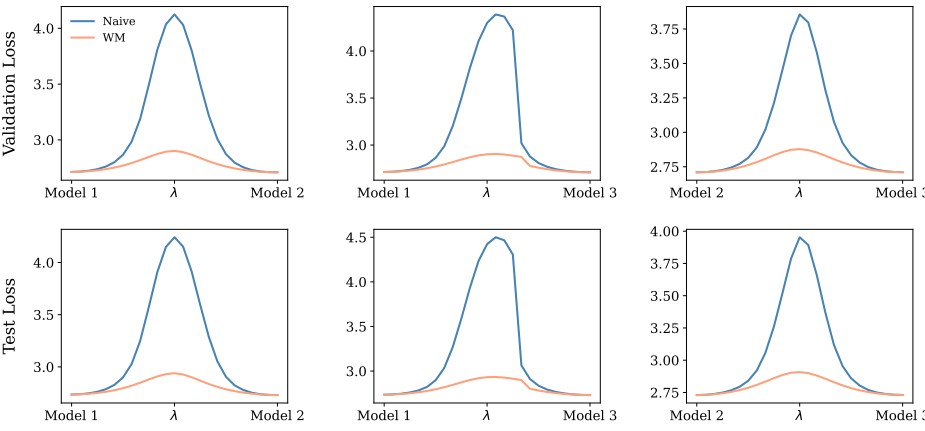

Figure 73: Linear Mode Connectivity for ViT-DeepSeekMoE ($k = 2, s = 1$) on Wikitext103 with 12 layers and 8 experts

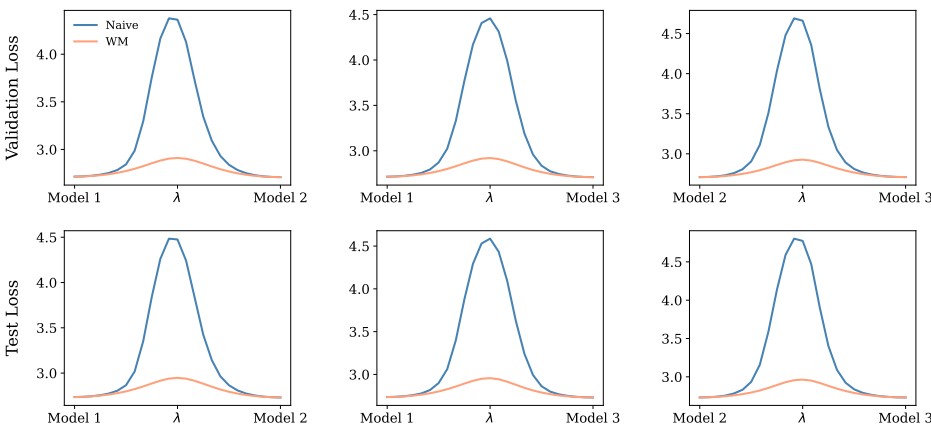

Figure 74: Linear Mode Connectivity for ViT-DeepSeekMoE ($k = 2, s = 1$) on Wikitext103 with 12 layers and 16 experts

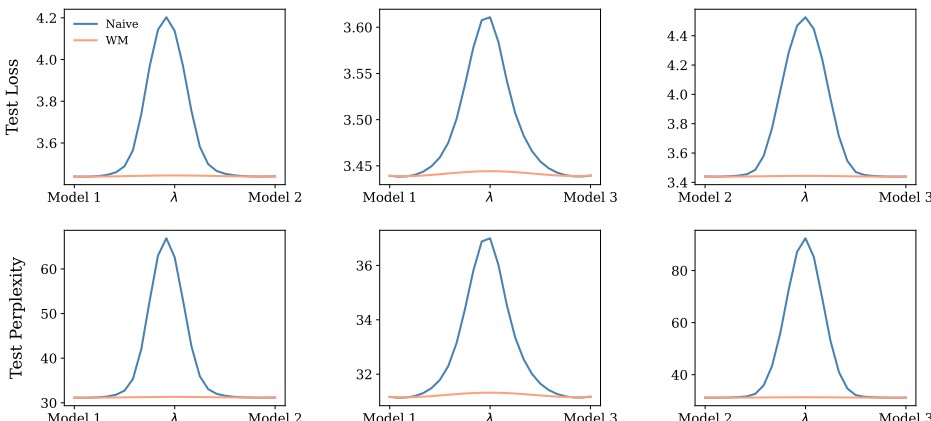

Figure 75: Linear Mode Connectivity for ViT-DeepSeekMoE ($k = 2, s = 1$) on One Billion Word with 12 layers and 4 experts

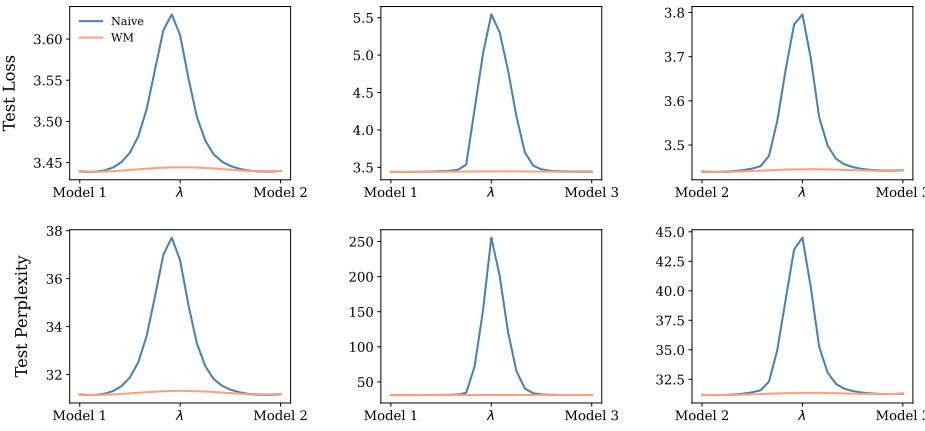

Figure 76: Linear Mode Connectivity for ViT-DeepSeekMoE ($k = 2, s = 1$) on One Billion Word with 12 layers and 8 experts

Table 6: In all configurations, the MLP component of the *last* Transformer layer is replaced by an MoE component.

| Method | Dataset | Number of layers | Number of experts | Figure |
|---|---|---|---|---|
| MoE | CIFAR-10 | 6 | [2, 4] | [78, 79] |
| | ImageNet-21k→CIFAR-10 | 12 | [2, 4] | [80, 81] |
| | ImageNet-21k→CIFAR-100 | 12 | [2, 4] | [82, 83] |
| SMoE ($k = 2$) | CIFAR-10 | 6 | [4, 8] | [84, 85] |
| | ImageNet-21k→CIFAR-10 | 12 | [4, 8] | [86, 87] |
| | ImageNet-21k→CIFAR-100 | 12 | [4, 8] | [88, 89] |
| DeepSeekMoE | CIFAR-10 | 6 | [4, 8] | [90, 91] |
| ($k = 2, s = 1$) | ImageNet-21k→CIFAR-10 | 12 | [4, 8] | [92, 93] |
| | ImageNet-21k→CIFAR-100 | 12 | [4, 8] | [94, 95] |

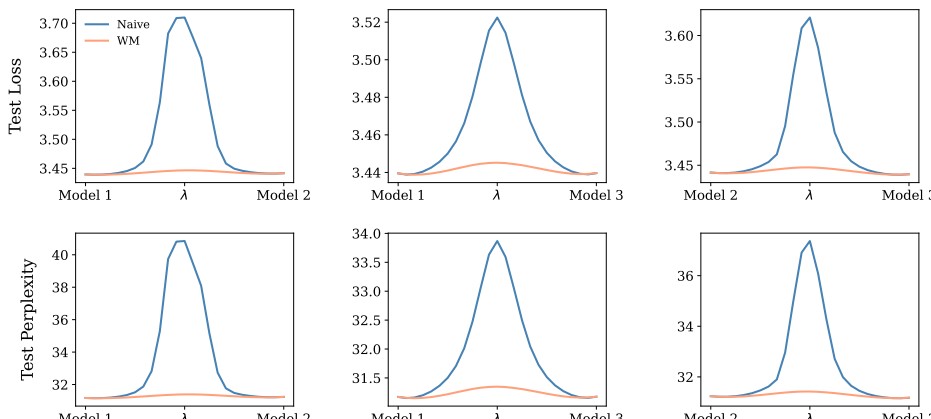

Figure 77: Linear Mode Connectivity for ViT-DeepSeekMoE ($k = 2, s = 1$) on One Billion Word with 12 layers and 16 experts

## G.2   Linear Mode Connectivity Analysis: Last Layer

We investigate Linear Mode Connectivity (LMC) in settings where the feed-forward network (FFN) of the *last* Transformer layer is replaced with a Mixture-of-Experts (MoE) module. This extends our earlier analysis of *first*-layer substitutions by examining connectivity at the terminal depth of the architecture, offering a complementary view on model plasticity and the modular structure of expert-based designs.

Following the procedure outlined in Section 6.1, all pretrained weights were frozen, and only the MoE module in the *last* layer was fine-tuned. For each configuration, three independent fine-tuning runs were conducted with different random seeds. LMC was assessed by evaluating the loss along linear interpolation paths between pairs of fine-tuned models, providing insight into the connectivity and overlap of their respective solution basins.

Table 6 summarizes the experimental settings for *last*-layer FFN replacement across dense MoE, SMoE, and DeepSeekMoE variants. Unless otherwise stated, all figure references correspond to *last*-layer experiments.

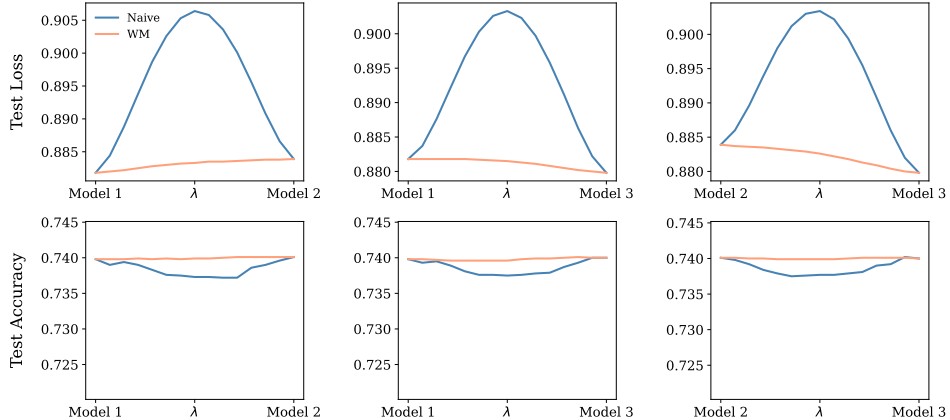

Figure 78: Linear Mode Connectivity for ViT-MoE on CIFAR-10 with 6 layers and 2 experts.

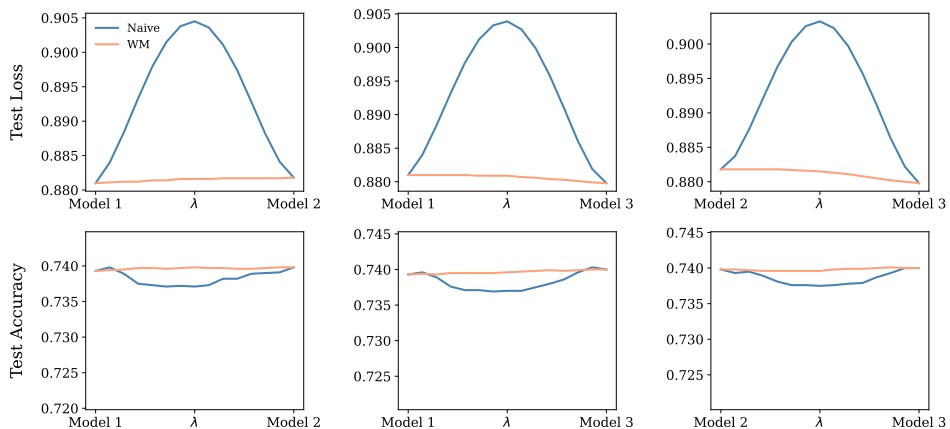

Figure 79: Linear Mode Connectivity for ViT-MoE on CIFAR-10 with 6 layers and 4 experts.

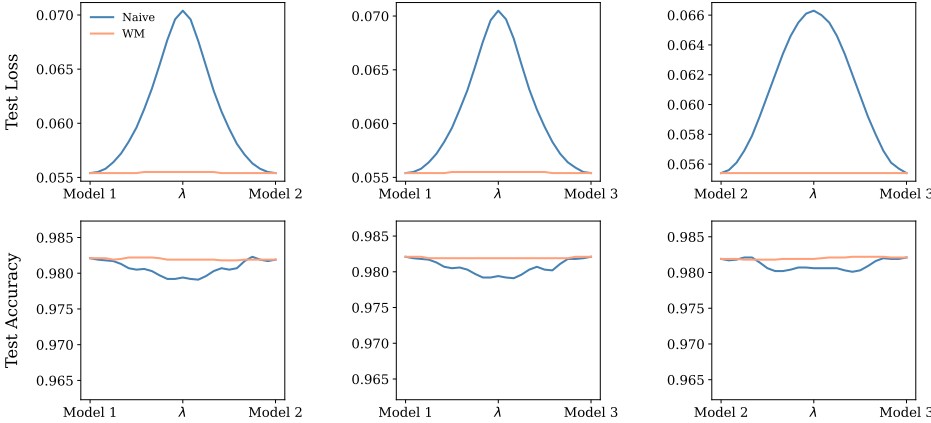

Figure 80: Linear Mode Connectivity for ViT-MoE on ImageNet-21k→CIFAR-10 with 12 layers and 2 experts.

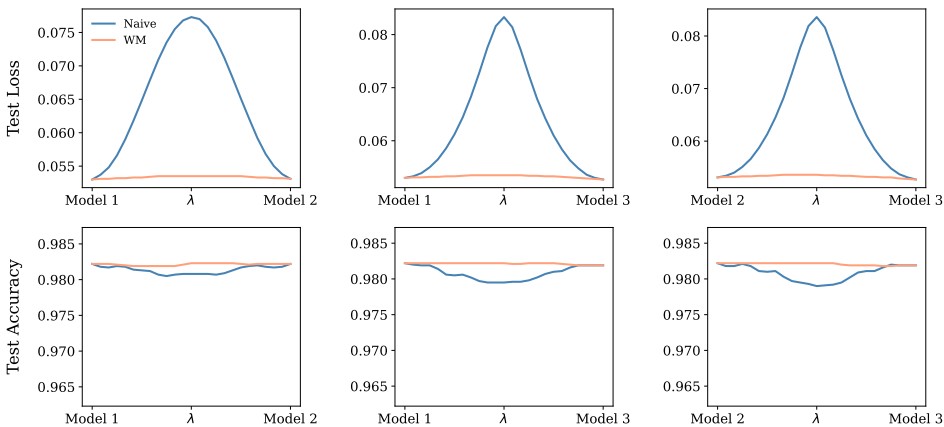

Figure 81: Linear Mode Connectivity for ViT-MoE on on ImageNet-21k→CIFAR-10 with 12 layers and 4 experts.

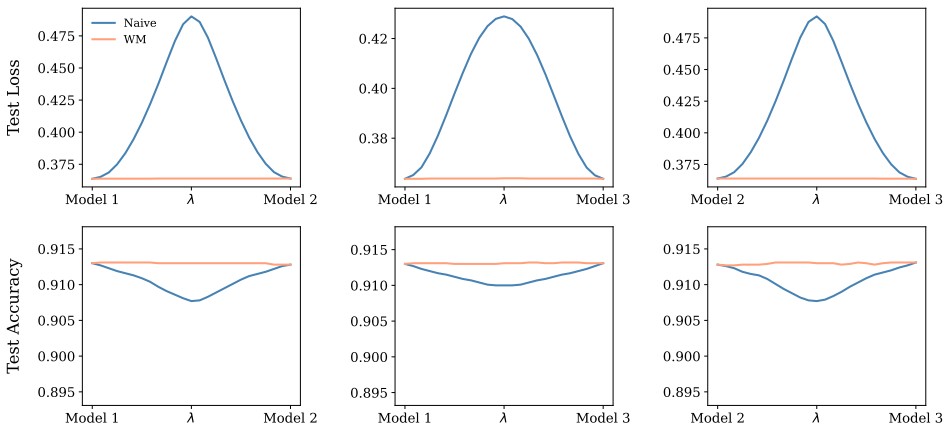

Figure 82: Linear Mode Connectivity for ViT-MoE on ImageNet-21k→CIFAR-100 with 12 layers and 2 experts.

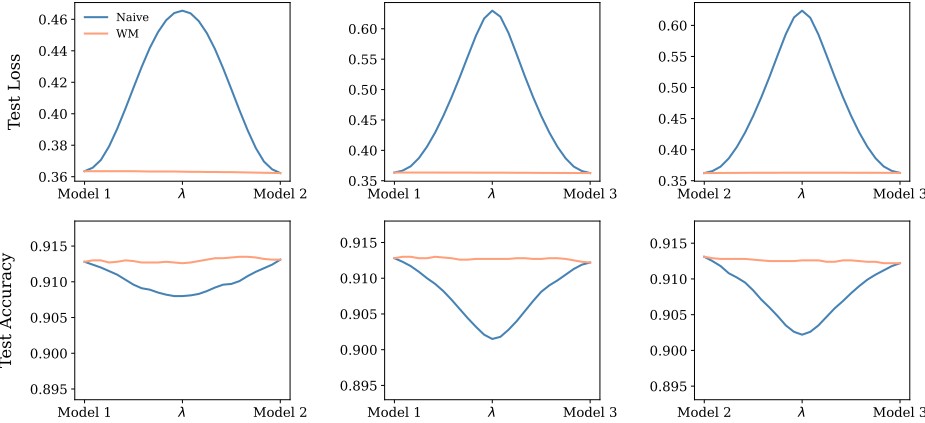

Figure 83: Linear Mode Connectivity for ViT-MoE on ImageNet-21k→CIFAR-100 with 12 layers and 4 experts.

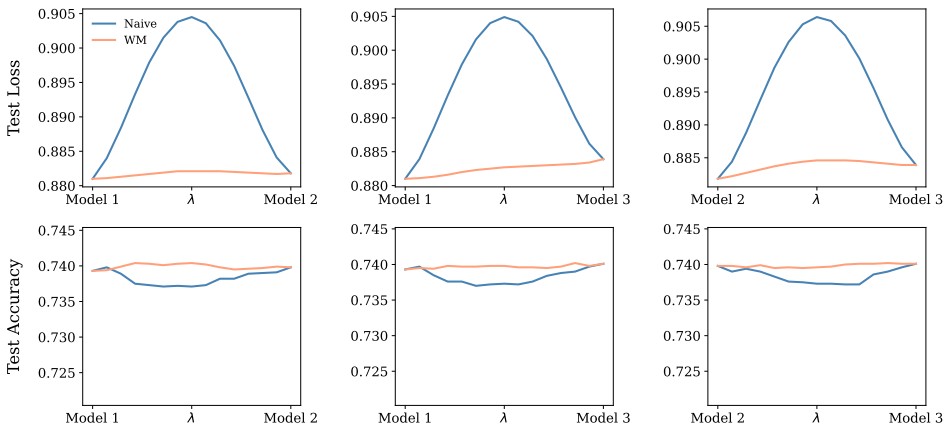

Figure 84: Linear Mode Connectivity for ViT-SMoE ($k = 2$) on CIFAR-10 with 6 layers and 4 experts.

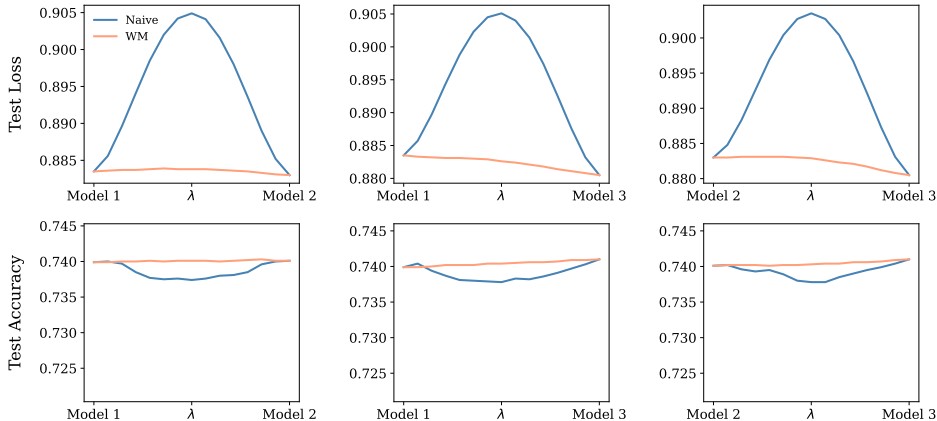

Figure 85: Linear Mode Connectivity for ViT-SMoE ($k = 2$) on CIFAR-10 with 6 layers and 8 experts.

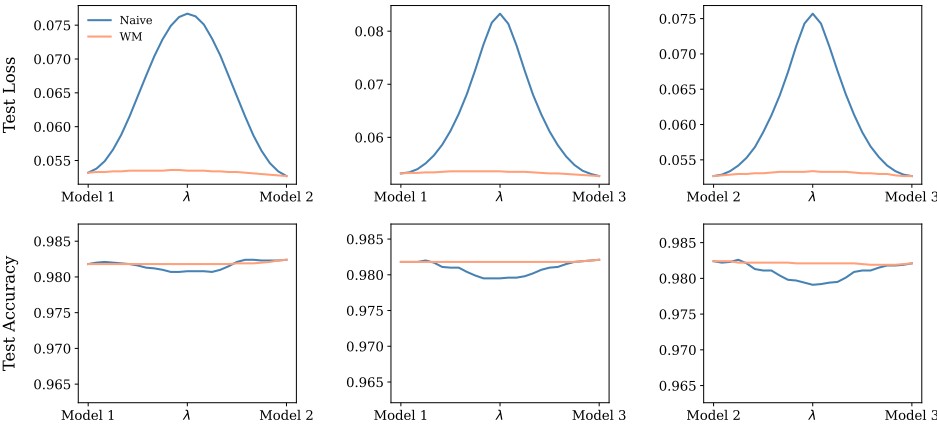

Figure 86: Linear Mode Connectivity for ViT-SMoE ($k = 2$) on ImageNet-21k→CIFAR-10 with 12 layers and 4 experts.

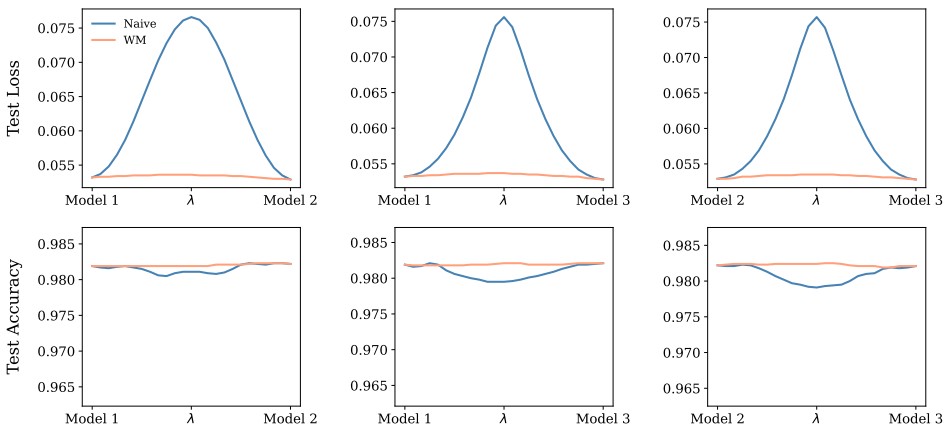

Figure 87: Linear Mode Connectivity for ViT-SMoE ($k = 2$) on ImageNet-21k→CIFAR-10 with 12 layers and 8 experts.

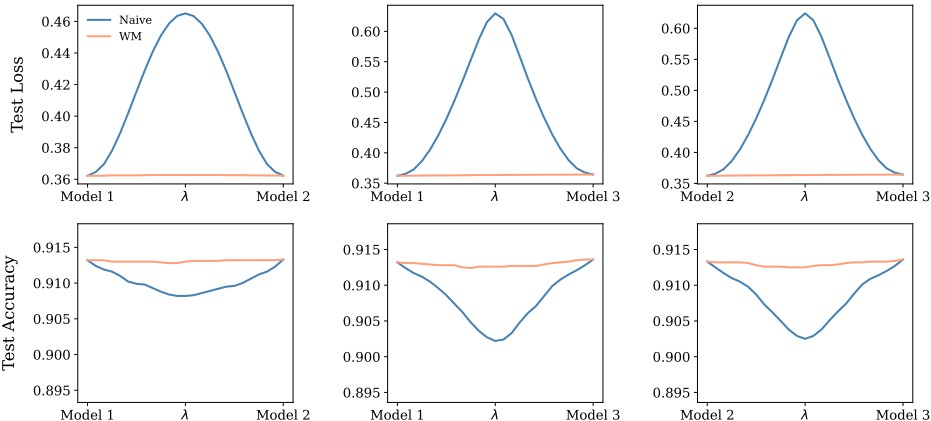

Figure 88: Linear Mode Connectivity for ViT-SMoE ($k = 2$) on ImageNet-21k→CIFAR-100 with 12 layers and 4 experts.

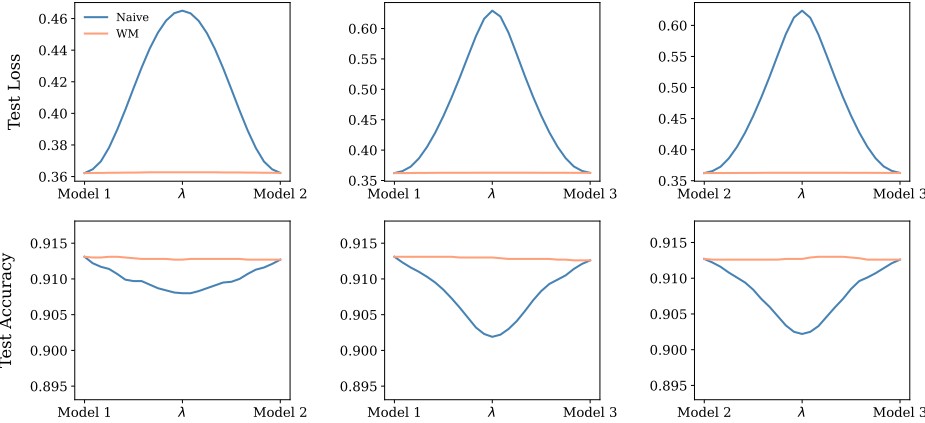

Figure 89: Linear Mode Connectivity for ViT-SMoE ($k = 2$) on ImageNet-21k→CIFAR-100 with 12 layers and 8 experts.

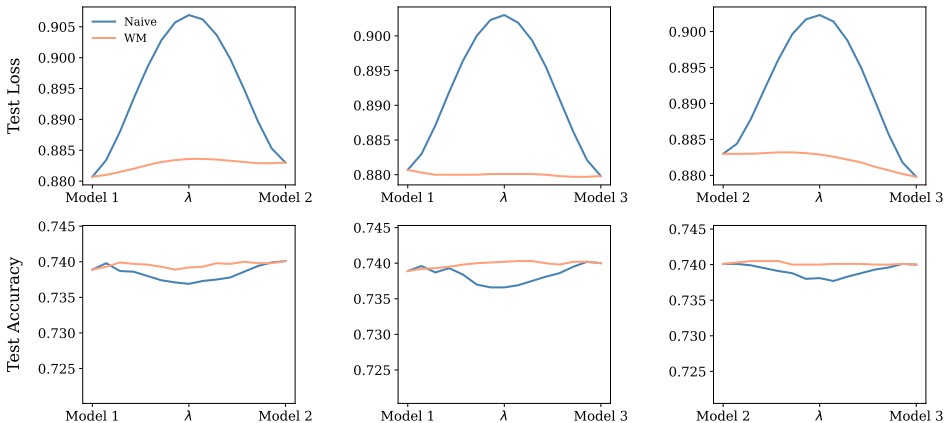

Figure 90: Linear Mode Connectivity for ViT-DeepSeekMoE ($k = 2, s = 1$) on CIFAR-10 with 6 layers and 4 experts.

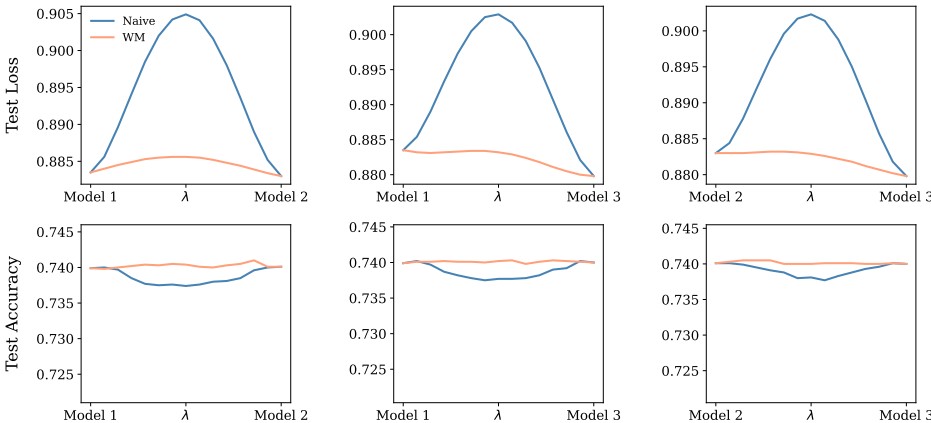

Figure 91: Linear Mode Connectivity for ViT-DeepSeekMoE ($k = 2, s = 1$) on CIFAR-10 with 6 layers and 8 experts.

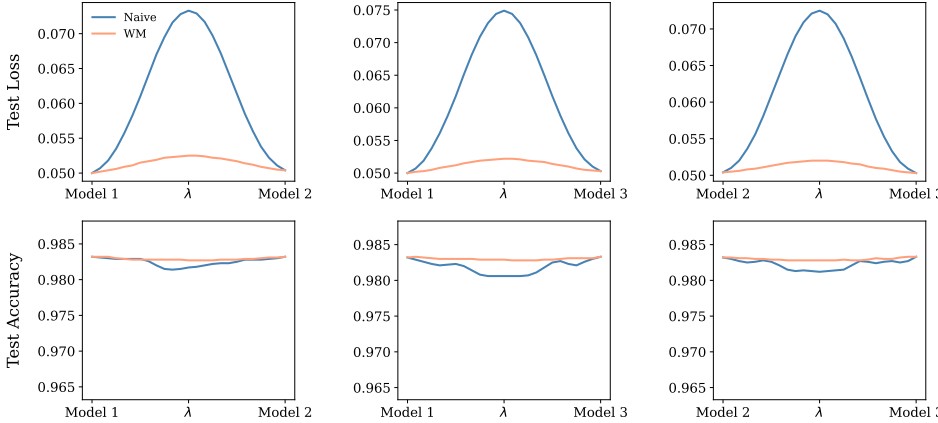

Figure 92: Linear Mode Connectivity for ViT-DeepSeekMoE ($k = 2, s = 1$) on ImageNet-21k→CIFAR-10 with 12 layers and 4 experts.

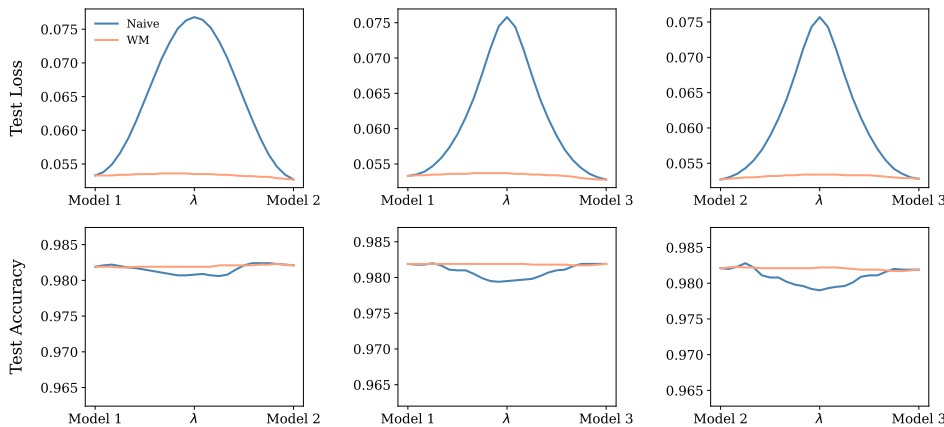

Figure 93: Linear Mode Connectivity for ViT-DeepSeekMoE ($k = 2, s = 1$) on ImageNet-21k→CIFAR-10 with 12 layers and 8 experts.

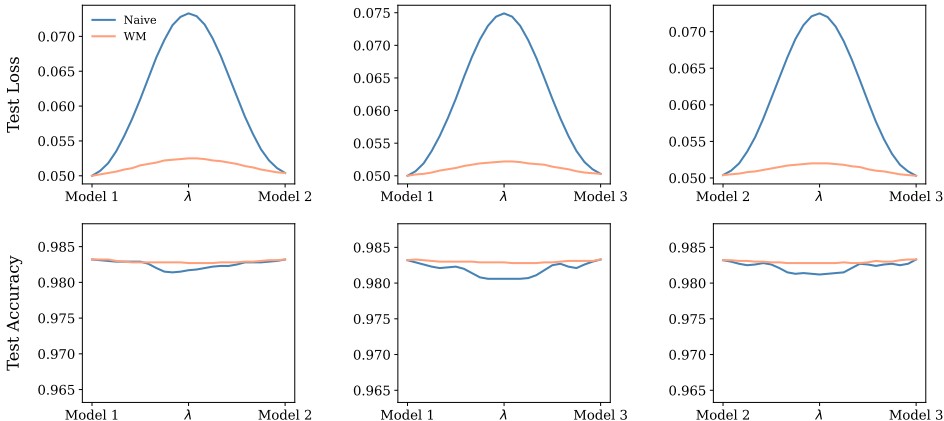

Figure 94: Linear Mode Connectivity for ViT-DeepSeekMoE ($k = 2, s = 1$) on ImageNet-21k→CIFAR-100 with 12 layers and 4 experts.

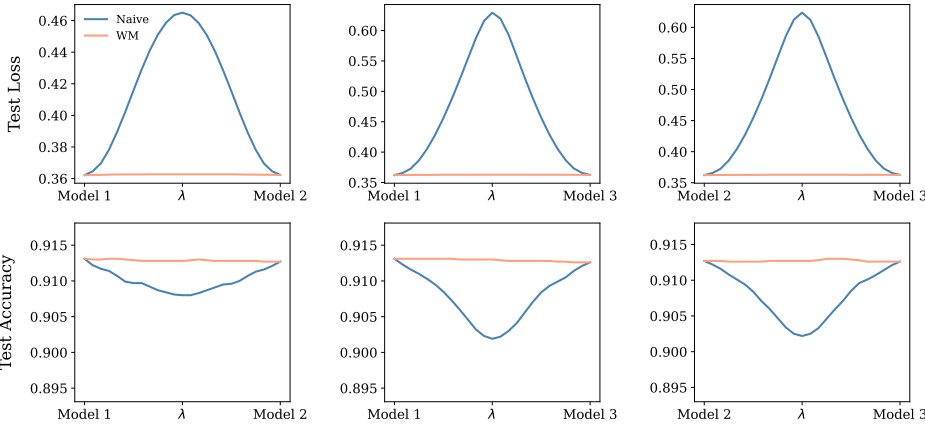

Figure 95: Linear Mode Connectivity for ViT-DeepSeekMoE ($k = 2, s = 1$) on ImageNet-21k→CIFAR-100 with 12 layers and 8 experts.

## G.3 Linear Mode Connectivity Analysis: All Layer

Table 7: Loss and accuracy barriers for weight matching interpolation versus naive interpolation, alongside test loss and accuracy values, across MoE, SMoE ($k = 2$), and DeepSeekMoE ($k = 2, s = 1$) variants of ViT models on MNIST, CIFAR-10/100 datasets, and of GPT-2 on the One Billion Word dataset. Barriers (lower is better ↓) are averaged over 6 pairs of models from a pool of 4 checkpoints, while metric values are averaged over the 4 model checkpoints. All metrics are scaled up by $10^2$ to improve readability.

| MoE variant | Dataset | No. layers | No. experts | WM Loss Barrier ↓ | Naive Loss Barrier ↓ | Loss value ↓ | WM Acc. Barrier ↓ | Naive Acc. Barrier ↓ | Acc. value ↑ |
|---|---|---|---|---|---|---|---|---|---|
| **MoE** | MNIST | 2 | 2 | $1.20 \pm 0.32$ | $11.62 \pm 2.44$ | $10.20 \pm 2.04$ | $2.12 \pm 0.21$ | $25.04 \pm 3.75$ | $97.32 \pm 0.73$ |
| | | | 4 | $1.31 \pm 0.21$ | $11.77 \pm 3.29$ | $8.45 \pm 2.53$ | $2.34 \pm 0.30$ | $26.62 \pm 4.22$ | $97.01 \pm 1.01$ |
| | CIFAR-10 | 2 | 4 | $4.24 \pm 0.72$ | $37.24 \pm 3.25$ | $95.06 \pm 0.42$ | $1.52 \pm 0.35$ | $14.21 \pm 1.42$ | $66.57 \pm 0.89$ |
| | | | 8 | $4.52 \pm 0.42$ | $36.66 \pm 4.01$ | $95.44 \pm 0.84$ | $1.74 \pm 0.50$ | $16.01 \pm 1.02$ | $65.93 \pm 1.20$ |
| | | 6 | 4 | $5.42 \pm 0.74$ | $47.47 \pm 8.02$ | $90.25 \pm 1.95$ | $2.21 \pm 0.52$ | $24.02 \pm 2.66$ | $74.61 \pm 2.13$ |
| | | | 8 | $5.27 \pm 1.26$ | $46.47 \pm 5.44$ | $91.31 \pm 2.32$ | $2.44 \pm 0.43$ | $22.45 \pm 3.21$ | $74.49 \pm 1.35$ |
| | CIFAR-100 | 6 | 4 | $2.32 \pm 0.51$ | $20.11 \pm 4.34$ | $94.04 \pm 2.02$ | $0.21 \pm 0.07$ | $3.22 \pm 0.71$ | $75.25 \pm 1.22$ |
| | | | 8 | $2.43 \pm 0.62$ | $23.33 \pm 3.33$ | $93.43 \pm 1.15$ | $0.27 \pm 0.12$ | $4.01 \pm 1.21$ | $75.66 \pm 1.30$ |
| | One Billion Word | 12 | 2 | $56.82 \pm 3.51$ | $400.11 \pm 50.54$ | $348.60 \pm 0.05$ | – | – | – |
| | | | 4 | $68.76 \pm 4.45$ | $435.65 \pm 34.92$ | $344.57 \pm 0.04$ | – | – | – |
| | | | 6 | $78.44 \pm 3.96$ | $476.65 \pm 29.58$ | $342.44 \pm 0.03$ | – | – | – |
| | | | 8 | $93.23 \pm 8.84$ | $455.82 \pm 38.10$ | $341.29 \pm 0.32$ | – | – | – |
| **SMoE** ($k = 2$) | MNIST | 2 | 2 | $1.55 \pm 0.23$ | $18.26 \pm 2.32$ | $11.21 \pm 2.02$ | $2.14 \pm 0.31$ | $24.92 \pm 3.56$ | $97.22 \pm 0.63$ |
| | | | 4 | $1.62 \pm 0.54$ | $19.19 \pm 2.27$ | $10.09 \pm 1.03$ | $2.30 \pm 0.34$ | $26.24 \pm 4.41$ | $97.06 \pm 1.00$ |
| | CIFAR-10 | 2 | 4 | $4.44 \pm 0.52$ | $37.44 \pm 3.35$ | $95.10 \pm 0.44$ | $1.50 \pm 0.37$ | $14.33 \pm 1.32$ | $66.62 \pm 0.69$ |
| | | | 8 | $4.61 \pm 0.46$ | $36.56 \pm 3.87$ | $95.74 \pm 0.92$ | $1.64 \pm 0.54$ | $17.04 \pm 1.32$ | $65.97 \pm 1.27$ |
| | | 6 | 4 | $5.57 \pm 0.84$ | $47.75 \pm 8.12$ | $90.33 \pm 1.85$ | $2.19 \pm 0.42$ | $24.12 \pm 2.44$ | $74.63 \pm 2.03$ |
| | | | 8 | $5.26 \pm 1.32$ | $46.61 \pm 5.36$ | $91.35 \pm 2.14$ | $2.31 \pm 0.41$ | $23.66 \pm 4.21$ | $74.51 \pm 1.25$ |
| | CIFAR-100 | 6 | 4 | $2.24 \pm 0.51$ | $21.25 \pm 2.05$ | $94.24 \pm 2.13$ | $0.22 \pm 0.08$ | $3.36 \pm 0.91$ | $75.35 \pm 1.11$ |
| | | | 8 | $2.67 \pm 0.52$ | $24.39 \pm 1.44$ | $93.83 \pm 4.25$ | $0.23 \pm 0.13$ | $4.12 \pm 1.02$ | $75.72 \pm 1.20$ |
| | One Billion Word | 12 | 4 | $72.31 \pm 0.28$ | $522.22 \pm 41.40$ | $345.02 \pm 0.00$ | – | – | – |
| | | | 8 | $79.86 \pm 4.81$ | $618.23 \pm 67.93$ | $342.31 \pm 0.11$ | – | – | – |
| | | | 16 | $98.68 \pm 18.59$ | $492.16 \pm 42.23$ | $340.57 \pm 0.05$ | – | – | – |
| **DeepSeekMoE** | MNIST | 2 | 2 | $1.35 \pm 0.64$ | $17.46 \pm 2.42$ | $12.32 \pm 2.02$ | $2.31 \pm 0.41$ | $25.12 \pm 3.85$ | $97.42 \pm 0.93$ |
| | | | 4 | $1.52 \pm 0.62$ | $16.72 \pm 2.33$ | $11.27 \pm 3.24$ | $2.04 \pm 0.32$ | $27.02 \pm 4.36$ | $97.17 \pm 1.31$ |
| ($k = 2, s = 1$) | CIFAR-10 | 2 | 4 | $4.22 \pm 0.42$ | $38.64 \pm 3.44$ | $95.11 \pm 0.33$ | $1.66 \pm 0.45$ | $17.31 \pm 1.63$ | $66.77 \pm 1.04$ |
| | | | 8 | $4.64 \pm 0.52$ | $38.86 \pm 4.11$ | $95.36 \pm 0.77$ | $1.68 \pm 0.52$ | $16.92 \pm 1.15$ | $65.99 \pm 1.11$ |
| | | 6 | 4 | $5.44 \pm 0.92$ | $52.27 \pm 5.32$ | $90.05 \pm 1.64$ | $2.12 \pm 0.62$ | $26.11 \pm 2.72$ | $74.63 \pm 2.33$ |
| | | | 8 | $5.47 \pm 1.22$ | $49.34 \pm 6.24$ | $91.21 \pm 2.12$ | $2.22 \pm 0.63$ | $25.25 \pm 4.21$ | $74.88 \pm 1.25$ |
| | CIFAR-100 | 6 | 4 | $2.43 \pm 0.50$ | $23.23 \pm 5.62$ | $95.22 \pm 3.15$ | $0.23 \pm 0.07$ | $3.02 \pm 0.61$ | $75.36 \pm 1.73$ |
| | | | 8 | $2.72 \pm 0.71$ | $24.55 \pm 4.07$ | $93.89 \pm 2.17$ | $0.24 \pm 0.11$ | $4.00 \pm 1.02$ | $75.76 \pm 1.21$ |
| | One Billion Word | 12 | 4 | $64.06 \pm 4.06$ | $426.11 \pm 58.07$ | $343.27 \pm 0.05$ | – | – | – |
| | | | 8 | $117.42 \pm 37.94$ | $390.59 \pm 28.63$ | $340.21 \pm 0.11$ | – | – | – |
| | | | 16 | $72.94 \pm 2.12$ | $623.63 \pm 147.42$ | $338.25 \pm 0.21$ | – | – | – |

## G.4 Expert Matching Method

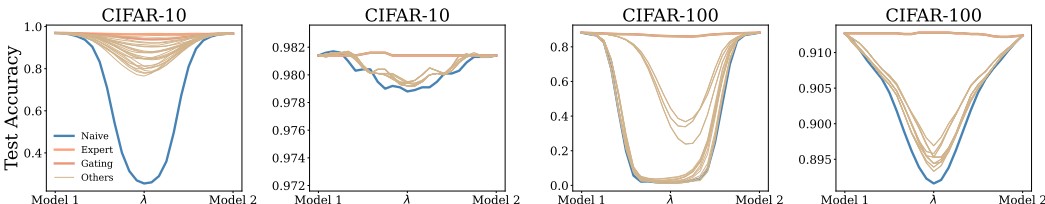

Figure 96: Accuracy curves for 12-layer ViT-MoE models with a 4-expert MoE replacement in either the first layer (subplots 1, 3) or the last layer (subplots 2, 4), on CIFAR-10 and CIFAR-100. The curves compare two Expert Order Matching methods across 24 permutations, with Weight Matching applied post-reordering to all permutations. Corresponding loss metrics are presented in Figure 2

Table 8: Comparison of expert matching methods (Expert Weight Matching and Gate Weight Matching) across three ViT-MoE model variants evaluated on the CIFAR-10 and CIFAR-100 test sets, measuring **loss**. Metrics reported include rank and $\hat{L}$, defined in Section 6.3, computed across 24 permutations for 10 checkpoint pairs. All models consist of 12 layers and 4 experts.

| Method | Dataset | Layer replaced | Expert Weight Matching | | Gate Weight Matching | |
|---|---|---|---|---|---|---|
| | | | Rank ↓ | $\hat{L}$ ↓ | Rank | $\hat{L}$ ↓ |
| MoE | CIFAR-10 | 1 | $2.50 \pm 1.50$ | $2.12 \pm 0.42$ | $3.00 \pm 1.00$ | $3.42 \pm 0.55$ |
| | | 4 | $2.10 \pm 0.50$ | $1.04 \pm 0.32$ | $2.60 \pm 0.92$ | $1.54 \pm 0.44$ |
| | | 8 | $2.70 \pm 0.46$ | $0.60 \pm 0.27$ | $2.90 \pm 0.30$ | $0.74 \pm 0.17$ |
| | | 12 | $4.60 \pm 2.00$ | $0.13 \pm 0.05$ | $3.80 \pm 1.66$ | $0.09 \pm 0.03$ |
| | CIFAR-100 | 1 | $2.80 \pm 0.40$ | $3.17 \pm 0.25$ | $2.90 \pm 0.70$ | $2.73 \pm 1.03$ |
| | | 4 | $3.60 \pm 1.20$ | $1.15 \pm 0.55$ | $2.70 \pm 1.00$ | $2.03 \pm 0.93$ |
| | | 8 | $3.30 \pm 0.78$ | $0.67 \pm 0.14$ | $3.20 \pm 0.87$ | $1.13 \pm 0.55$ |
| | | 12 | $3.40 \pm 0.92$ | $0.07 \pm 0.03$ | $4.20 \pm 0.89$ | $0.11 \pm 0.04$ |
| SMoE ($k = 2$) | CIFAR-10 | 1 | $3.00 \pm 1.00$ | $3.80 \pm 1.18$ | $3.20 \pm 0.98$ | $3.46 \pm 2.94$ |
| | | 4 | $2.80 \pm 0.98$ | $1.73 \pm 0.49$ | $2.40 \pm 1.56$ | $1.64 \pm 0.86$ |
| | | 8 | $2.60 \pm 0.49$ | $0.40 \pm 0.46$ | $2.60 \pm 0.43$ | $0.36 \pm 0.83$ |
| | | 12 | $4.00 \pm 2.45$ | $0.06 \pm 0.04$ | $2.80 \pm 1.66$ | $0.22 \pm 0.19$ |
| | CIFAR-100 | 1 | $2.80 \pm 0.40$ | $3.29 \pm 3.18$ | $2.10 \pm 0.70$ | $2.00 \pm 2.02$ |
| | | 4 | $2.60 \pm 1.20$ | $1.20 \pm 0.43$ | $3.20 \pm 1.47$ | $1.91 \pm 1.05$ |
| | | 8 | $3.10 \pm 0.83$ | $0.13 \pm 0.07$ | $2.70 \pm 0.90$ | $0.11 \pm 0.09$ |
| | | 12 | $2.40 \pm 0.92$ | $0.07 \pm 0.13$ | $2.00 \pm 0.89$ | $0.03 \pm 0.12$ |
| DeepSeekMoE ($k = 2, s = 1$) | CIFAR-10 | 1 | $3.10 \pm 2.12$ | $5.06 \pm 1.22$ | $3.60 \pm 0.83$ | $4.28 \pm 2.92$ |
| | | 4 | $2.60 \pm 0.77$ | $3.32 \pm 0.38$ | $3.10 \pm 1.52$ | $2.80 \pm 0.88$ |
| | | 8 | $2.30 \pm 0.51$ | $0.39 \pm 0.48$ | $2.40 \pm 0.37$ | $0.78 \pm 1.00$ |
| | | 12 | $3.60 \pm 3.55$ | $0.08 \pm 0.04$ | $3.60 \pm 1.48$ | $0.23 \pm 0.17$ |
| | CIFAR-100 | 1 | $4.20 \pm 0.60$ | $4.45 \pm 3.40$ | $3.70 \pm 0.45$ | $4.21 \pm 2.60$ |
| | | 4 | $2.50 \pm 0.78$ | $1.34 \pm 0.46$ | $3.10 \pm 1.91$ | $2.98 \pm 1.46$ |
| | | 8 | $1.90 \pm 0.60$ | $0.13 \pm 0.08$ | $3.10 \pm 1.02$ | $0.13 \pm 0.16$ |
| | | 12 | $3.30 \pm 1.26$ | $0.04 \pm 0.11$ | $2.70 \pm 0.89$ | $0.02 \pm 0.20$ |

Table 10: Loss barrier comparison between Total Weight Matching and Skipping-Expert-Order Matching across MoE variants on the One Billion Words dataset. Each value is averaged over 3 independent checkpoint pairs. The last column reports the interpolated loss value at the midpoint between matched models.

| MoE Variant | No. Layers | No. Experts | Loss Barrier ↓ | | Loss Value ↓ |
|---|---|---|---|---|---|
| | | | Total Weight Matching | Skipping-Expert-Order Matching | |
| MoE | 12 | 4 | $0.0053 \pm 0.0006$ | $0.1384 \pm 0.0960$ | $3.4399 \pm 0.0002$ |
| | | 8 | $0.0066 \pm 0.0008$ | $0.1728 \pm 0.1144$ | $3.4394 \pm 0.0001$ |
| SMoE ($k = 2$) | 12 | 4 | $0.0054 \pm 0.0002$ | $0.8579 \pm 1.1412$ | $3.4403 \pm 0.0001$ |
| | | 8 | $0.0051 \pm 0.0005$ | $0.2202 \pm 0.1470$ | $3.4400 \pm 0.0000$ |
| DeepSeekMoE ($k = 2, s = 1$) | 12 | 4 | $0.0047 \pm 0.0001$ | $0.3223 \pm 0.3817$ | $3.4395 \pm 0.0003$ |
| | | 8 | $0.0044 \pm 0.0004$ | $0.4005 \pm 0.3666$ | $3.4395 \pm 0.0002$ |

Table 9: Comparison of expert matching methods (Expert Weight Matching and Gate Weight Matching) across three ViT-MoE model variants evaluated on the CIFAR-10 and CIFAR-100 test sets, measuring **accuracy**. Metrics reported include rank and $\hat{L}$, defined in Section 6.3, computed across 24 permutations for 10 checkpoint pairs. All models consist of 12 layers and 4 experts.

| Method | Dataset | Layer replaced | Expert Weight Matching | | Gate Weight Matching | |
| --- | --- | --- | --- | --- | --- | --- |
| | | | Rank ↓ | $\hat{L}$ ↓ | Rank | $\hat{L}$ ↓ |
| MoE | CIFAR-10 | 1 | $4.90 \pm 2.59$ | $1.80 \pm 0.78$ | $4.40 \pm 1.80$ | $2.00 \pm 0.44$ |
| | | 4 | $3.20 \pm 0.40$ | $0.95 \pm 0.97$ | $2.90 \pm 0.70$ | $1.64 \pm 0.98$ |
| | | 8 | $3.00 \pm 1.80$ | $0.03 \pm 0.05$ | $3.80 \pm 1.83$ | $0.05 \pm 0.05$ |
| | | 12 | $4.00 \pm 1.45$ | $0.02 \pm 0.03$ | $2.80 \pm 1.40$ | $0.04 \pm 0.06$ |
| | CIFAR-100 | 1 | $3.20 \pm 1.47$ | $2.66 \pm 1.03$ | $4.20 \pm 1.00$ | $1.51 \pm 1.16$ |
| | | 4 | $2.70 \pm 0.78$ | $1.87 \pm 1.17$ | $2.90 \pm 0.83$ | $1.39 \pm 1.06$ |
| | | 8 | $4.70 \pm 1.42$ | $0.08 \pm 0.02$ | $3.80 \pm 1.47$ | $0.05 \pm 0.03$ |
| | | 12 | $2.40 \pm 0.92$ | $0.05 \pm 0.02$ | $3.20 \pm 0.87$ | $0.02 \pm 0.02$ |
| SMoE ($k = 2$) | CIFAR-10 | 1 | $2.30 \pm 2.06$ | $3.09 \pm 1.21$ | $3.20 \pm 2.75$ | $2.24 \pm 0.90$ |
| | | 4 | $2.90 \pm 0.59$ | $1.16 \pm 1.00$ | $2.80 \pm 0.83$ | $1.75 \pm 1.00$ |
| | | 8 | $3.40 \pm 1.47$ | $0.07 \pm 0.03$ | $3.00 \pm 1.74$ | $0.04 \pm 0.02$ |
| | | 12 | $3.60 \pm 1.21$ | $0.12 \pm 0.08$ | $4.70 \pm 2.05$ | $0.07 \pm 0.03$ |
| | CIFAR-100 | 1 | $2.10 \pm 1.85$ | $3.52 \pm 2.26$ | $2.40 \pm 0.93$ | $2.05 \pm 2.95$ |
| | | 4 | $2.00 \pm 0.67$ | $1.81 \pm 1.28$ | $2.90 \pm 0.73$ | $1.02 \pm 2.41$ |
| | | 8 | $4.30 \pm 1.60$ | $0.09 \pm 0.03$ | $3.10 \pm 1.02$ | $0.11 \pm 0.08$ |
| | | 12 | $2.60 \pm 1.03$ | $0.11 \pm 0.08$ | $2.70 \pm 0.89$ | $0.03 \pm 0.06$ |
| DeepSeekMoE ($k = 2, s = 1$) | CIFAR-10 | 1 | $2.80 \pm 2.20$ | $2.95 \pm 1.15$ | $3.30 \pm 2.40$ | $2.60 \pm 0.85$ |
| | | 4 | $3.10 \pm 0.80$ | $1.40 \pm 1.13$ | $2.90 \pm 0.90$ | $1.60 \pm 1.30$ |
| | | 8 | $3.60 \pm 1.32$ | $0.09 \pm 0.04$ | $3.40 \pm 1.10$ | $0.07 \pm 0.03$ |
| | | 12 | $3.20 \pm 1.00$ | $0.06 \pm 0.02$ | $3.00 \pm 0.87$ | $0.05 \pm 0.02$ |
| | CIFAR-100 | 1 | $2.54 \pm 1.90$ | $3.20 \pm 2.00$ | $2.80 \pm 2.00$ | $2.90 \pm 1.80$ |
| | | 4 | $2.90 \pm 0.70$ | $1.60 \pm 1.20$ | $3.10 \pm 0.80$ | $1.40 \pm 1.00$ |
| | | 8 | $3.80 \pm 1.43$ | $0.10 \pm 0.03$ | $3.50 \pm 1.20$ | $0.08 \pm 0.04$ |
| | | 12 | $2.70 \pm 0.90$ | $0.07 \pm 0.02$ | $2.90 \pm 0.70$ | $0.06 \pm 0.03$ |

## G.5 Ablation Study on Number of Layers

Table 11: Ablation Study on varying the number of Transformer layers and all MoE variants Integration in ViT for CIFAR-10. The ratios of metrics (loss barrier, loss AUC, accuracy barrier, and accuracy AUC) compare Algorithm 1 to naive interpolation. For full table results (non-raio), kindly refer to Tables 12 and 13

| Method | Number of layers | Layer replaced | Loss barrier ratio (%) ↓ | Loss AUC ratio (%) ↓ | Acc. barrier ratio (%) ↓ | Acc. AUC ratio (%) ↓ |
|---|---|---|---|---|---|---|
| MoE | 2 | 1 | $8.54 \pm 1.53$ | $8.73 \pm 1.68$ | $8.39 \pm 1.42$ | $8.01 \pm 1.39$ |
| | | 2 | $9.33 \pm 2.04$ | $8.94 \pm 1.92$ | $9.10 \pm 2.19$ | $8.83 \pm 2.22$ |
| | 4 | 1 | $8.29 \pm 1.63$ | $8.08 \pm 1.59$ | $7.41 \pm 1.45$ | $6.43 \pm 1.09$ |
| | | 2 | $10.19 \pm 3.43$ | $10.19 \pm 3.16$ | $10.21 \pm 5.26$ | $9.08 \pm 5.33$ |
| | | 3 | $8.50 \pm 1.82$ | $7.83 \pm 1.89$ | $9.82 \pm 2.58$ | $8.70 \pm 2.58$ |
| | | 4 | $5.12 \pm 0.67$ | $4.60 \pm 0.65$ | $4.17 \pm 1.68$ | $4.58 \pm 1.01$ |
| | 6 | 1 | $4.16 \pm 1.47$ | $4.96 \pm 2.31$ | $3.64 \pm 1.00$ | $4.13 \pm 1.55$ |
| | | 2 | $6.78 \pm 4.80$ | $7.71 \pm 3.09$ | $7.27 \pm 5.73$ | $8.98 \pm 2.94$ |
| | | 3 | $9.69 \pm 4.92$ | $8.62 \pm 5.46$ | $9.03 \pm 3.31$ | $8.14 \pm 3.57$ |
| | | 4 | $10.83 \pm 5.61$ | $12.01 \pm 6.42$ | $8.92 \pm 2.21$ | $10.00 \pm 7.30$ |
| | | 5 | $9.56 \pm 4.52$ | $11.72 \pm 5.59$ | $8.72 \pm 2.89$ | $11.01 \pm 3.83$ |
| | | 6 | $6.90 \pm 3.54$ | $9.67 \pm 6.59$ | $7.16 \pm 3.98$ | $9.69 \pm 3.11$ |
| SMoE ($k = 2$) | 2 | 1 | $8.93 \pm 0.92$ | $9.34 \pm 1.23$ | $9.36 \pm 1.33$ | $8.84 \pm 1.40$ |
| | | 2 | $7.55 \pm 1.39$ | $6.62 \pm 1.35$ | $9.10 \pm 2.88$ | $7.29 \pm 2.78$ |
| | 4 | 1 | $8.96 \pm 1.54$ | $8.83 \pm 1.64$ | $8.66 \pm 2.07$ | $8.21 \pm 1.91$ |
| | | 2 | $9.62 \pm 2.68$ | $10.07 \pm 3.50$ | $9.41 \pm 3.04$ | $9.67 \pm 3.52$ |
| | | 3 | $13.31 \pm 1.49$ | $12.44 \pm 2.06$ | $11.37 \pm 3.01$ | $13.63 \pm 3.17$ |
| | | 4 | $9.17 \pm 1.40$ | $9.36 \pm 1.48$ | $10.97 \pm 2.77$ | $12.52 \pm 2.57$ |
| | 6 | 1 | $2.37 \pm 1.92$ | $2.65 \pm 1.34$ | $2.25 \pm 2.71$ | $4.67 \pm 2.38$ |
| | | 2 | $10.09 \pm 3.11$ | $9.26 \pm 3.19$ | $8.35 \pm 3.37$ | $8.36 \pm 2.38$ |
| | | 3 | $11.22 \pm 6.25$ | $10.17 \pm 8.52$ | $12.10 \pm 7.04$ | $12.80 \pm 10.77$ |
| | | 4 | $8.76 \pm 2.06$ | $12.16 \pm 2.22$ | $13.09 \pm 5.49$ | $10.85 \pm 7.21$ |
| | | 5 | $14.39 \pm 7.82$ | $14.66 \pm 9.10$ | $11.82 \pm 9.19$ | $12.45 \pm 7.85$ |
| | | 6 | $6.38 \pm 5.01$ | $9.56 \pm 3.79$ | $11.96 \pm 11.93$ | $10.93 \pm 15.06$ |
| DeepSeekMoE ($k = 2, r = 1$) | 2 | 1 | $8.94 \pm 1.95$ | $9.12 \pm 2.10$ | $7.96 \pm 1.59$ | $7.88 \pm 1.90$ |
| | | 2 | $9.25 \pm 0.37$ | $8.27 \pm 0.44$ | $12.00 \pm 1.35$ | $11.14 \pm 1.73$ |
| | 4 | 1 | $8.65 \pm 1.01$ | $8.56 \pm 1.19$ | $7.17 \pm 0.99$ | $6.46 \pm 0.88$ |
| | | 2 | $10.80 \pm 3.77$ | $10.88 \pm 3.69$ | $10.45 \pm 4.91$ | $9.92 \pm 4.83$ |
| | | 3 | $7.90 \pm 1.79$ | $8.82 \pm 1.87$ | $10.21 \pm 2.44$ | $9.02 \pm 2.52$ |
| | | 4 | $8.35 \pm 1.20$ | $8.29 \pm 1.39$ | $9.29 \pm 2.75$ | $7.04 \pm 3.14$ |
| | 6 | 1 | $6.29 \pm 5.00$ | $6.17 \pm 3.48$ | $8.07 \pm 5.34$ | $8.35 \pm 5.65$ |
| | | 2 | $8.03 \pm 1.79$ | $9.16 \pm 2.10$ | $7.88 \pm 3.66$ | $8.54 \pm 3.93$ |
| | | 3 | $9.27 \pm 4.25$ | $9.29 \pm 3.19$ | $12.93 \pm 2.18$ | $11.56 \pm 5.58$ |
| | | 4 | $10.80 \pm 2.18$ | $12.72 \pm 1.52$ | $7.34 \pm 3.64$ | $8.99 \pm 5.10$ |
| | | 5 | $13.77 \pm 2.27$ | $13.03 \pm 7.01$ | $11.14 \pm 5.47$ | $13.28 \pm 3.32$ |
| | | 6 | $9.82 \pm 2.99$ | $8.38 \pm 4.81$ | $8.92 \pm 4.22$ | $8.14 \pm 3.48$ |

Table 12: Ablation study results on CIFAR-10 for LMC in ViT with layer replacements of all MoE variants, reporting test set **loss**. We evaluate the loss metrics for linear interpolation using our proposed Algorithm 1 against naive linear interpolation. Metrics include the loss barrier and loss Area Under the Curve (AUC), with AUC computed relative to the straight line connecting the two model endpoints. Each experiment is repeated *five* times, and LMC is performed on *ten* model pairs. Additionally, we report loss values across five models to provide enhanced context. To improve readability, all metrics are scaled up by a factor of $10^2$.

| Method | Number of layers | Layer replaced | WM loss barrier ↓ | Naive loss barrier ↓ | WM loss AUC ↓ | Naive loss AUC ↓ | Loss value ↓ |
|---|---|---|---|---|---|---|---|
| MoE | 2 | 1 | $3.14 \pm 0.55$ | $34.22 \pm 1.25$ | $1.45 \pm 0.28$ | $16.11 \pm 0.74$ | $99.04 \pm 0.92$ |
| | | 2 | $2.84 \pm 0.44$ | $33.54 \pm 1.12$ | $1.38 \pm 0.15$ | $17.53 \pm 0.57$ | $104.52 \pm 0.81$ |
| | 4 | 1 | $9.48 \pm 1.36$ | $115.56 \pm 10.19$ | $4.25 \pm 0.68$ | $53.07 \pm 4.54$ | $97.00 \pm 0.59$ |
| | | 2 | $6.59 \pm 2.12$ | $64.73 \pm 2.05$ | $3.71 \pm 1.11$ | $36.39 \pm 0.96$ | $91.74 \pm 0.87$ |
| | | 3 | $4.20 \pm 0.90$ | $48.23 \pm 0.92$ | $2.21 \pm 0.53$ | $28.23 \pm 0.50$ | $88.06 \pm 1.30$ |
| | | 4 | $2.76 \pm 0.38$ | $53.92 \pm 0.80$ | $1.42 \pm 0.22$ | $31.01 \pm 0.54$ | $86.15 \pm 0.33$ |
| | 6 | 1 | $1.42 \pm 0.44$ | $37.47 \pm 8.47$ | $0.74 \pm 0.30$ | $18.73 \pm 3.30$ | $90.35 \pm 1.15$ |
| | | 2 | $0.47 \pm 0.32$ | $7.09 \pm 0.60$ | $0.27 \pm 0.10$ | $3.53 \pm 0.30$ | $88.73 \pm 0.39$ |
| | | 3 | $0.33 \pm 0.17$ | $3.42 \pm 0.22$ | $0.13 \pm 0.08$ | $1.60 \pm 0.14$ | $88.26 \pm 0.50$ |
| | | 4 | $0.14 \pm 0.08$ | $1.36 \pm 0.33$ | $0.07 \pm 0.04$ | $0.64 \pm 0.16$ | $86.48 \pm 0.17$ |
| | | 5 | $0.14 \pm 0.06$ | $1.68 \pm 0.44$ | $0.08 \pm 0.03$ | $0.83 \pm 0.21$ | $85.92 \pm 0.46$ |
| | | 6 | $0.23 \pm 0.11$ | $2.46 \pm 0.29$ | $0.12 \pm 0.04$ | $1.26 \pm 0.21$ | $88.61 \pm 0.33$ |
| SMoE $(k=2)$ | 2 | 1 | $2.95 \pm 0.24$ | $33.17 \pm 1.01$ | $1.39 \pm 0.16$ | $14.95 \pm 0.51$ | $98.88 \pm 0.86$ |
| | | 2 | $2.52 \pm 0.46$ | $33.46 \pm 0.67$ | $1.20 \pm 0.25$ | $18.12 \pm 0.48$ | $105.59 \pm 0.37$ |
| | 4 | 1 | $10.45 \pm 1.99$ | $116.54 \pm 8.34$ | $4.75 \pm 1.01$ | $53.65 \pm 3.97$ | $96.73 \pm 0.85$ |
| | | 2 | $8.31 \pm 0.83$ | $86.88 \pm 3.25$ | $4.73 \pm 0.57$ | $47.40 \pm 1.91$ | $90.80 \pm 1.02$ |
| | | 3 | $4.73 \pm 0.62$ | $35.49 \pm 1.34$ | $2.59 \pm 0.49$ | $20.78 \pm 0.80$ | $86.29 \pm 0.24$ |
| = | | 4 | $4.48 \pm 0.70$ | $48.84 \pm 0.68$ | $2.68 \pm 0.44$ | $28.62 \pm 0.44$ | $82.33 \pm 0.37$ |
| | 6 | 1 | $2.12 \pm 0.67$ | $34.23 \pm 7.22$ | $1.28 \pm 0.35$ | $12.59 \pm 2.67$ | $90.25 \pm 1.00$ |
| | | 2 | $0.75 \pm 0.29$ | $7.28 \pm 0.83$ | $0.34 \pm 0.15$ | $3.59 \pm 0.43$ | $88.99 \pm 0.65$ |
| | | 3 | $0.34 \pm 0.19$ | $3.28 \pm 0.30$ | $0.15 \pm 0.06$ | $1.76 \pm 0.13$ | $87.69 \pm 0.52$ |
| | | 4 | $0.26 \pm 0.14$ | $2.78 \pm 0.38$ | $0.14 \pm 0.07$ | $1.35 \pm 0.19$ | $86.08 \pm 0.40$ |
| | | 5 | $0.22 \pm 0.11$ | $2.50 \pm 0.04$ | $0.11 \pm 0.05$ | $1.25 \pm 0.02$ | $86.08 \pm 0.34$ |
| | | 6 | $0.23 \pm 0.18$ | $3.63 \pm 0.17$ | $0.13 \pm 0.08$ | $1.93 \pm 0.10$ | $88.38 \pm 0.36$ |
| DeepSeekMoE $(k=2, s=1)$ | 2 | 1 | $3.00 \pm 0.61$ | $33.63 \pm 1.30$ | $1.37 \pm 0.27$ | $15.14 \pm 0.73$ | $98.41 \pm 1.06$ |
| | | 2 | $3.07 \pm 0.21$ | $33.16 \pm 1.26$ | $1.50 \pm 0.11$ | $18.18 \pm 0.72$ | $105.92 \pm 0.53$ |
| | 4 | 1 | $10.57 \pm 1.57$ | $122.13 \pm 10.50$ | $4.91 \pm 0.84$ | $57.32 \pm 4.89$ | $97.71 \pm 1.26$ |
| | | 2 | $5.85 \pm 2.14$ | $54.13 \pm 2.27$ | $2.90 \pm 1.26$ | $30.17 \pm 1.35$ | $90.89 \pm 0.52$ |
| | | 3 | $2.67 \pm 0.65$ | $33.36 \pm 1.47$ | $1.35 \pm 0.40$ | $19.48 \pm 0.89$ | $86.37 \pm 0.43$ |
| | | 4 | $4.11 \pm 0.61$ | $49.18 \pm 0.95$ | $2.37 \pm 0.41$ | $28.49 \pm 0.55$ | $82.26 \pm 0.39$ |
| | 6 | 1 | $2.25 \pm 0.82$ | $39.42 \pm 6.23$ | $1.14 \pm 0.38$ | $18.88 \pm 3.32$ | $91.32 \pm 0.25$ |
| | | 2 | $2.88 \pm 0.57$ | $28.53 \pm 1.26$ | $1.56 \pm 0.32$ | $15.97 \pm 0.69$ | $89.50 \pm 1.05$ |
| | | 3 | $0.56 \pm 0.12$ | $7.90 \pm 0.18$ | $0.26 \pm 0.06$ | $3.90 \pm 0.12$ | $88.22 \pm 0.57$ |
| | | 4 | $0.24 \pm 0.16$ | $3.07 \pm 0.34$ | $0.12 \pm 0.03$ | $1.53 \pm 0.14$ | $86.09 \pm 0.40$ |
| | | 5 | $0.32 \pm 0.17$ | $3.39 \pm 0.31$ | $0.16 \pm 0.08$ | $1.63 \pm 0.13$ | $86.10 \pm 0.51$ |
| | | 6 | $0.23 \pm 0.08$ | $2.61 \pm 0.53$ | $0.14 \pm 0.04$ | $1.43 \pm 0.32$ | $89.27 \pm 0.57$ |

Table 13: Ablation study results on CIFAR-10 for LMC in ViT with layer replacements of all MoE variants, reporting test set **accuracy**. We evaluate the accuracy metrics for linear interpolation using our proposed Algorithm 1 against naive linear interpolation. Metrics include the accuracy barrier and accuracy Area Under the Curve (AUC), with AUC computed relative to the straight line connecting the two model endpoints. Each experiment is repeated *five* times, and LMC is performed on *ten* model pairs. Additionally, we report accuracy values across five models to provide enhanced context. To improve readability, all metrics are scaled up by a factor of $10^2$.

| Method | Number of layers | Layer replaced | WM acc. barrier ↓ | Naive acc. barrier ↓ | WM acc. AUC ↓ | Naive acc. barrier ↓ | Acc. value ↑ |
|---|---|---|---|---|---|---|---|
| MoE | 2 | 1 | 0.91 ± 0.28 | 12.33 ± 1.28 | 0.44 ± 0.15 | 5.89 ± 0.54 | 66.53 ± 3.30 |
| | | 2 | 0.76 ± 0.21 | 7.85 ± 2.31 | 0.33 ± 0.08 | 3.77 ± 1.11 | 63.65 ± 0.47 |
| | 4 | 1 | 2.10 ± 0.42 | 28.36 ± 1.85 | 0.86 ± 0.16 | 13.43 ± 0.99 | 72.48 ± 0.68 |
| | | 2 | 1.41 ± 0.71 | 13.83 ± 0.53 | 0.75 ± 0.41 | 7.88 ± 0.29 | 73.43 ± 0.61 |
| | | 3 | 0.67 ± 0.17 | 6.84 ± 0.24 | 0.35 ± 0.09 | 4.00 ± 0.12 | 73.64 ± 0.20 |
| | | 4 | 0.23 ± 0.09 | 5.48 ± 0.19 | 0.14 ± 0.03 | 3.05 ± 0.14 | 73.58 ± 0.19 |
| | 6 | 1 | 0.50 ± 0.13 | 13.95 ± 2.20 | 0.24 ± 0.07 | 5.71 ± 0.93 | 74.29 ± 0.35 |
| | | 2 | 0.35 ± 0.10 | 2.49 ± 0.36 | 0.12 ± 0.04 | 1.23 ± 0.21 | 74.65 ± 0.44 |
| | | 3 | 0.14 ± 0.05 | 1.48 ± 0.28 | 0.05 ± 0.02 | 0.72 ± 0.14 | 74.72 ± 0.46 |
| | | 4 | 0.05 ± 0.08 | 1.33 ± 0.43 | 0.03 ± 0.05 | 0.68 ± 0.24 | 74.46 ± 0.12 |
| | | 5 | 0.05 ± 0.03 | 0.59 ± 0.05 | 0.02 ± 0.01 | 0.28 ± 0.02 | 74.62 ± 0.08 |
| | | 6 | 0.06 ± 0.04 | 0.61 ± 0.12 | 0.03 ± 0.02 | 0.30 ± 0.05 | 74.25 ± 0.21 |
| SMoE ($k = 2$) | 2 | 1 | 1.07 ± 0.15 | 11.42 ± 0.22 | 0.46 ± 0.07 | 5.20 ± 0.09 | 65.27 ± 0.69 |
| | | 2 | 0.58 ± 0.17 | 6.41 ± 0.24 | 0.26 ± 0.10 | 3.63 ± 0.13 | 62.58 ± 0.16 |
| | 4 | 1 | 2.48 ± 0.66 | 28.55 ± 1.55 | 1.12 ± 0.29 | 13.52 ± 0.81 | 72.26 ± 0.60 |
| | | 2 | 1.78 ± 0.23 | 19.00 ± 0.71 | 0.97 ± 0.15 | 10.16 ± 0.36 | 73.12 ± 0.40 |
| | | 3 | 0.87 ± 0.17 | 4.98 ± 0.32 | 0.38 ± 0.10 | 2.78 ± 0.22 | 73.58 ± 0.33 |
| | | 4 | 0.79 ± 0.14 | 5.29 ± 0.31 | 0.40 ± 0.09 | 3.15 ± 0.15 | 73.60 ± 0.44 |
| | 6 | 1 | 0.30 ± 0.38 | 12.61 ± 2.27 | 0.28 ± 0.11 | 5.11 ± 0.86 | 74.09 ± 0.28 |
| | | 2 | 0.29 ± 0.16 | 2.85 ± 0.37 | 0.13 ± 0.08 | 1.48 ± 0.22 | 74.81 ± 0.47 |
| | | 3 | 0.18 ± 0.14 | 1.36 ± 0.18 | 0.10 ± 0.06 | 0.95 ± 0.10 | 74.97 ± 0.30 |
| | | 4 | 0.18 ± 0.07 | 1.35 ± 0.05 | 0.08 ± 0.02 | 0.76 ± 0.03 | 74.60 ± 0.24 |
| | | 5 | 0.19 ± 0.09 | 1.45 ± 0.08 | 0.09 ± 0.04 | 0.77 ± 0.05 | 74.72 ± 0.24 |
| | | 6 | 0.19 ± 0.08 | 1.60 ± 0.14 | 0.08 ± 0.02 | 0.89 ± 0.09 | 74.15 ± 0.28 |
| DeepSeekMoE ($k = 2, s = 1$) | 2 | 1 | 0.92 ± 0.18 | 11.64 ± 0.48 | 0.42 ± 0.10 | 5.29 ± 0.25 | 65.43 ± 0.30 |
| | | 2 | 0.78 ± 0.11 | 6.45 ± 0.31 | 0.41 ± 0.08 | 3.67 ± 0.21 | 62.55 ± 0.27 |
| | 4 | 1 | 2.07 ± 0.31 | 28.85 ± 1.64 | 0.90 ± 0.13 | 13.93 ± 0.91 | 71.57 ± 0.84 |
| | | 2 | 1.77 ± 0.56 | 12.14 ± 0.50 | 0.67 ± 0.32 | 6.74 ± 0.24 | 73.70 ± 0.14 |
| | | 3 | 1.01 ± 0.14 | 9.80 ± 0.28 | 0.55 ± 0.08 | 4.91 ± 0.17 | 73.72 ± 0.20 |
| | | 4 | 0.52 ± 0.15 | 5.60 ± 0.14 | 0.23 ± 0.10 | 3.22 ± 0.11 | 73.89 ± 0.30 |
| | 6 | 1 | 2.43 ± 0.73 | 30.80 ± 2.43 | 0.99 ± 0.31 | 14.06 ± 0.90 | 74.26 ± 0.19 |
| | | 2 | 0.95 ± 0.19 | 12.63 ± 0.43 | 0.47 ± 0.10 | 7.37 ± 0.18 | 74.51 ± 0.14 |
| | | 3 | 0.39 ± 0.13 | 4.85 ± 0.14 | 0.20 ± 0.06 | 2.44 ± 0.09 | 74.98 ± 0.19 |
| | | 4 | 0.14 ± 0.07 | 2.29 ± 0.08 | 0.10 ± 0.04 | 1.13 ± 0.03 | 74.70 ± 0.16 |
| | | 5 | 0.12 ± 0.03 | 1.39 ± 0.08 | 0.08 ± 0.02 | 0.66 ± 0.04 | 74.62 ± 0.14 |
| | | 6 | 0.16 ± 0.07 | 1.74 ± 0.12 | 0.06 ± 0.04 | 0.88 ± 0.05 | 74.31 ± 0.49 |

## H Broader Impact

The investigation into Linear Mode Connectivity (LMC) within Mixture-of-Experts (MoE) architectures presents both potential positive and negative societal implications. On the positive side, this work could lead to more efficient training methods for MoE models, enhancing their scalability and computational efficiency. Such advancements may democratize access to advanced AI technologies, enabling researchers and developers with limited resources to leverage powerful models for innovative applications. Furthermore, the insights gained into the optimization dynamics and functional landscape of neural networks could contribute to the development of models that generalize more effectively, thereby improving the reliability and robustness of AI systems in critical domains such as healthcare, finance, and autonomous systems. Additionally, this research enriches the broader AI community's understanding of neural network loss landscapes, fostering further theoretical and practical advancements.

It is important to note that while these potential impacts are significant, the primary contribution of this work lies in its theoretical and foundational insights. The actual societal ramifications will largely depend on how these insights are applied and governed in practical settings. As such, responsible development and deployment practices, coupled with ongoing research into bias mitigation and ethical AI, are essential to maximizing the benefits and minimizing the risks associated with advancements in MoE architectures.

