# OpenReview forum: "On Linear Mode Connectivity of Mixture-of-Experts Architectures"
_NeurIPS.cc/2025/Conference — NeurIPS 2025 oral_

### Official Review · Reviewer_7X3K · 2025-06-22

**Clarity:** 3
**Significance:** 2
**Originality:** 2
**Rating:** 5
**Confidence:** 2

**Summary:**

This paper provides an affirmative answer to the question of whether MoE networks exhibit LMC when utilizing the symmetries inherent to MoE architectures. Specifically, it addresses not only the permutation symmetries of FNNs used in existing research, but also the translation invariance of the softmax function. The authors prove functional equivalence theorems for both dense gating and sparse gating, propose a Weight Matching algorithm, and empirically confirm the existence of LMC across diverse datasets and architectures.

**Questions:**

**Q1. Validity of theoretical assumptions**: To what extent are the strongly distinct assumptions in Theorem 4.2 satisfied in actual ReLU networks? Could partial results under weaker assumptions be obtained, or validation on real data be provided?

**Q2. Scalability and practicality**: Does the proposed method scale to larger actual MoE models (such as Switch Transformer)? How can the discovered LMC be utilized for practical model fusion applications?

**Q3. Generalization potential**: Equation (9) demonstrates that continuous symmetries (translation) do not affect loss barriers while discrete symmetries (permutation) are essential for mode connectivity in MoE architectures. This finding may represent a more general principle: continuous transformations preserve connectivity within the same basin, while discrete transformations enable connectivity between different basins. Have the authors considered whether this continuous vs. discrete framework could be extended to other architectures (e.g., attention heads in Transformers, convolutional filters)? Establishing this as a general principle could significantly broaden the theoretical impact of this work and provide a unified understanding of symmetry roles in neural network loss landscapes.

**Ethical Concerns:**

["NO or VERY MINOR ethics concerns only"]

**Final Justification:**

I acknowledge the novelty of this paper's theory that considers MoE. I believe that the extension of symmetry to MoE is a steady and important step toward identifying the symmetry of the entire transformer in the future.

**Limitations:**

The main limitations of the paper are: (1) unverified practicality of strong theoretical assumptions in sparse gating, (2) experiments limited to relatively small-scale models and datasets, and (3) unclear concrete pathways to practical applications. While the authors acknowledge some of these limitations in the conclusion, more detailed analysis and presentation of future research plans would be desirable.

**Paper Formatting Concerns:**

Nothing

**Quality:**

3

**Strengths And Weaknesses:**

## Strrengths

**S1. Novel theoretical framework**: The unified treatment of permutation symmetries and translation invariance represents a valuable theoretical contribution. The complete characterization proofs of functional equivalence for both dense gating (Theorem 4.1) and sparse gating (Theorem 4.2) appear to be correct.

**S2. Comprehensive and systematic experimental design**: The authors validate three MoE variants (Dense MoE, SMoE, DeepSeekMoE) across both vision (MNIST, CIFAR, ImageNet) and language (WikiText103, One Billion Word) domains. Systematic ablation studies varying the number of experts (2-16) and layers (1-12) demonstrate the universality of LMC.

**S3. Practical algorithmic contribution**: The Weight Matching algorithm for expert permutation alignment with O(n³ + nh³) computational complexity is efficient and implementable, utilizing permutation-invariant representations through Gram matrices.

## Weaknesses

**W1. Strong theoretical assumptions**: Sparse gating (Theorem 4.2) requires very strong assumptions such as "strongly distinct" expert functions and linear independence of {Wi−1−Wi}, but the analysis of how well these assumptions are satisfied in actual MoE models is insufficient.

**W2. Limited experimental scope**: The experiments concentrate mainly on FFN replacement, confirming LMC only for parts of actual MoE networks. Additionally, the model sizes are limited to relatively small scales.

---

> ### Author Rebuttal · Authors · 2025-07-31
>
> We address the concerns raised by the Reviewer in the Weaknesses and Questions sections as follows.
>
> ---
>
> **W1. Strong theoretical assumptions: Sparse gating (Theorem 4.2) requires very strong assumptions such as "strongly distinct" expert functions and linear independence of {Wi−1−Wi}, but the analysis of how well these assumptions are satisfied in actual MoE models is insufficient.**
>
> **Q1. Validity of theoretical assumptions: To what extent are the strongly distinct assumptions in Theorem 4.2 satisfied in actual ReLU networks? Could partial results under weaker assumptions be obtained, or validation on real data be provided?**
>
> **Answer to W1 and Q1.** Regarding the concerns related to the set of assumptions in the two main theorems of the paper, namely Theorem 4.1 and Theorem 4.2, we would like to note that we have already provided clarifications on both the necessity and the implications of these assumptions in Section 4.2. These are further elaborated in Remarks B.6 and C.8 in the Appendix. In particular, we explain why these assumptions are typically satisfied in practice and provide examples of scenarios where they fail - referred to as singular cases - under which the theorems may no longer hold. We kindly invite the Reviewer to revisit those sections, as we believe they contain a clear and thorough explanation.
>
> Additionally, we would like to elaborate more on the “strongly distinct” condition in Remark C.8, which may initially appear to be a restrictive or unrealistic assumption. In fact, this condition is almost surely satisfied in practice. Specifically, if one randomly selects a set of ReLU experts, they will, with probability one, be strongly distinct. While a full proof of this claim may be lengthy, the intuition is straightforward: each ReLU expert can be viewed as a locally affine map, and within each affine region, it is parameterized by a vector in $\mathbb{R}^d$ for some fixed $d$. Under this perspective, the strongly distinct condition reduces to a statement akin to: “A finite set of randomly sampled vectors in $\mathbb{R}^d$ are almost surely pairwise distinct.” We emphasize that this observation is not intended to serve as a formal proof; however, due to its intuitive nature and practical relevance, we believe it provides a satisfactory justification to address the Reviewer’s concern.
>
>
> **W2. The experiments concentrate mainly on FFN replacement, confirming LMC only for parts of actual MoE networks. Additionally, the model sizes are limited to relatively small scales.**
>
> **Q2.  Scalability and practicality: Does the proposed method scale to larger actual MoE models (such as Switch Transformer)? How can the discovered LMC be utilized for practical model fusion applications?**
>
> **Answer to W2 and Q2.** We appreciate the reviewer's suggestion to explore model fusion. To address this, we experimented with **model soups** [Wortsman et al., 2022], which leverages LMC to average unaligned model weights for improved performance.
>
> In our experiments, we applied model soups within our experimental setup using the LMC pipeline. Specifically, we aligned MoE layers from fine-tuned checkpoints of the same pretrained Transformer, testing configurations with MoE replacement at the first layer as well as across all layers. However, these efforts yielded only insignificant improvements over the best original checkpoint in the pool. A key insight from our visualizations is that the midpoint loss along both naive and aligned loss barrier curves remains the highest (though significantly reduced post-alignment), suggesting limited gains from simple averaging in this MoE-only setup.
>
> We hypothesize that model soups would be more effective in flexible, generalized frameworks extending beyond MoE layers alone. Our current algorithm aligns only MoE components, which renders full-model fusion less meaningful. Extending alignment to the entire Transformer architecture introduces complex symmetries, such as those in Multi-Head Attention (MHA) layers (e.g., symmetries under invertible transformations on query/key matrices: $Q \cdot K^\top = (QA^\top)(KA^{-1})^\top$). To provide intuition on how symmetries can emerge in composite layers, consider two parameterized layers, $f(\cdot;\alpha) : \mathbb{R}^{d_0} \to \mathbb{R}^{d_1}$ and $g(\cdot;\beta) : \mathbb{R}^{d_1} \to \mathbb{R}^{d_2}$, composed as $g \circ f: \mathbb{R}^{d_0} \to \mathbb{R}^{d_2}$. In practice, these layers may exhibit symmetries. For instance, let $P$ be a $d_1 \times d_1$ permutation matrix representing a reordering of indices in $\mathbb{R}^{d_1}$. Suppose there exist modified parameters $\alpha'$ and $\beta'$ such that $f(\cdot;\alpha') = P \circ f(\cdot;\alpha)$ and $g(\cdot;\beta') = g(\cdot;\beta) \circ P^{-1}$. Then, the composite function remains invariant: $g(\cdot;\beta) \circ f(\cdot;\alpha) = g(\cdot;\beta') \circ f(\cdot;\alpha')$. Typically, $\alpha'$ involves permuting the coefficients of $\alpha$ according to $P$, and similarly for $\beta'$. For example, one may think of $f$ and $g$ as feedforward layers, with $P$ permuting neurons at the output of $f$ and input of $g$. Depending on the layers, $P$ can generalize beyond permutations to include generalized permutation matrices, orthogonal matrices, or arbitrary invertible matrices. This illustrates one form of symmetry that arises when stacking layers.
>
> Characterizing these global symmetries across heterogeneous Transformer stacks is challenging, as no general method yet exists for aligning such composites. Due to scope and time constraints, we focused on MoE alignment in this work and deferred full-model experiments. We are actively developing a comprehensive alignment algorithm for entire Transformer blocks, including MoEs, and plan to demonstrate model fusion applications in future work. This could enable practical benefits, such as efficient merging of large models for enhanced generalization or resource savings.
>
> **Q3.  Generalization potential: Equation (9) demonstrates that continuous symmetries (translation) do not affect loss barriers while discrete symmetries (permutation) are essential for mode connectivity in MoE architectures. This finding may represent a more general principle: continuous transformations preserve connectivity within the same basin, while discrete transformations enable connectivity between different basins. Have the authors considered whether this continuous vs. discrete framework could be extended to other architectures (e.g., attention heads in Transformers, convolutional filters)? Establishing this as a general principle could significantly broaden the theoretical impact of this work and provide a unified understanding of symmetry roles in neural network loss landscapes.**
>
> **Answer to Q3.** In the context of MoE, we agree that continuous symmetries do not influence the loss barrier, and hence do not affect the observation of LMC. However, we believe this is largely due to the specific structure of MoE, considered in Equation (2) and (3). In general, various forms of continuous symmetry can be found in other architectures. A well-known example arises in the Attention mechanism, where there exists a continuous symmetry induced by the general linear group acting on the query and key matrices. Specifically, let $Q$ and $K$ denote the query and key matrices; for any invertible matrix $A$, we have $Q \cdot K^{\top} = (QA^\top)(KA^{-1})^\top$ demonstrating a continuous symmetry under the action of the general linear group on $Q$ and $K$.
>
> Building on this, our experiments to explore LMC in this setting reveal that considering only permutation matrices for $A$ was insufficient to observe LMC in Attention layers. This suggests that focusing solely on discrete symmetries may not be adequate for identifying LMC in architectures that admit richer symmetry structures.
>
> The problems of characterizing functional equivalence in neural architectures and identifying LMC are both promising and foundational areas of research. A complete understanding of symmetry in deep learning models will likely require a general, conceptual, and rigorous framework - one that we believe holds great potential for advancing our understanding of network behavior through the lens of symmetry.
>
>
> ---
> We thank the Reviewer for their constructive feedback, thoughtful evaluation, and probing questions, which have significantly strengthened our paper by prompting deeper clarifications and highlighting opportunities for broader impact. We particularly appreciate the suggestions on validating assumptions, scaling to larger models, and exploring practical applications like model fusion. We will incorporate the recommended enhancements, including expanded discussions on the practicality of assumptions (e.g., via additional intuitions and remarks), scalability considerations, and future research directions for full-model alignment and generalization to other architectures, to further improve the paper's rigor and applicability. If the Reviewer is satisfied with our responses to the weaknesses and questions raised, we kindly hope that the evaluation may be adjusted to reflect this. We remain open to further discussion and would be happy to engage in the next phase of the review process.

---

> > ### Author Response · Authors · 2025-08-07
> >
> > Dear Reviewer,
> >
> > As we approach the final days of the discussion phase, we would like to kindly follow up regarding the concerns you raised during the review process. We sincerely hope that our responses have addressed your questions and clarified the key aspects of our work.
> >
> > If you find our clarifications satisfactory, we would be grateful if you could consider updating your evaluation to reflect this. Of course, if there remain any unresolved points or further questions, we would be more than happy to continue the discussion.
> >
> > We truly value the thoughtful feedback we have received throughout the review process. Engaging with experts across different areas has greatly contributed to strengthening our work, and we are thankful for the opportunity to benefit from your insights.
> >
> > Warm regards,
> >
> > The Authors

---

> > > ### Comment · Reviewer_7X3K · 2025-08-08
> > >
> > > Thank you for your response. My concerns regarding Q1 have been resolved.
> > > Concerns regarding W2 remain unresolved. As HzV1 also points out, the alignment in this research is limited to the FFN of MoE, and considering that existing research has already succeeded in FFN alignment, the impact of these experimental results is minimal.Considering the response to Q3, I speculate that while the authors are examining symmetry that includes attention, the current status is that it is difficult to experimentally demonstrate that LMC holds.
> > >
> > > On the other hand, I acknowledge the novelty of this paper's theory that considers MoE. I believe that the extension of symmetry to MoE is a steady and important step toward identifying the symmetry of the entire transformer in the future. Therefore, I would like to maintain my score.

---

> ### Author Response · Authors · 2025-08-08
>
> We sincerely thank the Reviewer for engaging with us during the discussion phase. Below, we address your additional concerns.
>
> ---
>
> > Concerns regarding W2 remain unresolved. As HzV1 also points out, the alignment in this research is limited to the FFN of MoE, and considering that existing research has already succeeded in FFN alignment, the impact of these experimental results is minimal.Considering the response to Q3, I speculate that while the authors are examining symmetry that includes attention, the current status is that it is difficult to experimentally demonstrate that LMC holds.
>
> **Answer.** We argee with the Reviewer that, given the established presence of LMC in FFNs, one might naturally anticipate similar behavior in MoE  architectures. However, we argue that MoEs introduce fundamentally distinct challenges. In particular, the inclusion of a non-linear softmax gating mechanism distinguishes MoEs from classical FFNs. This gating induces conditional computation and sparse expert activation, which in turn leads to highly non-uniform training signals and pronounced expert specialization. As a result, the optimization landscape becomes considerably more fragmented, thereby complicating alignment across independently trained models.
>
> Our ablation studies on the expert order matching method, presented in Section 6.3, **highlight the necessity of each stage in our alignment procedure for successfully revealing LMC in MoEs**. Notably, the first stage - expert order alignment - proves to be a key innovation, enabling meaningful correspondence between experts. As illustrated in Section 6.3, the four plots in Figure 2 and the corresponding four plots in Figure 96 reveal that the majority of the 24 possible expert order permutations yield poor alignment. Due to permutation invariance, any such permutation is equally likely between two checkpoints without explicit expert order matching. The second stage - weight matching 0 further aligns the FFN components, whose importance has been well established in prior literature. Empirically, we observe that omitting or altering any of these stages - such as bypassing expert permutation alignment or removing weight matching - substantially degrades the quality of the interpolated models, frequently resulting in significant loss barriers.
>
> We also provide additional experimental results, as shown in the table below.
>
> |MoE variant|No. layers|No. experts|Total Weight Matching Loss Barrier ↓|Skipping-Expert-Order Matching Loss Barrier ↓|
> |-|-|-|-|-|
> |MoE|12|4|0.53 ± 0.06|13.84 ± 9.60|
> |||8|0.66 ± 0.08|17.28 ± 11.44|
> |SMoE ($k=2$)|12|4|0.54 ± 0.02|85.79 ± 114.12|
> |||8|0.51 ± 0.05|22.02 ± 14.70|
> |DeepSeekMoE ($k=2, s=1$)|12|4|0.47 ± 0.01|32.23 ± 38.17|
> |||8|0.44 ± 0.04|40.05 ± 36.66|
>
> The table reports the loss barrier (averaged over 3 pairs of independently trained checkpoints on the One Billion Words dataset and scaled by a factor of $10^2$ for readability) under two different matching strategies: Total Weight Matching - our proposed method which aligns both expert order and expert weights - and Skipping-Expert-Order Matching, which matches expert weights without resolving permutation alignment.
>
> Our results indicate that Total Weight Matching consistently yields a significantly lower loss barrier across different MoE variants and expert counts, while the Skipping-Expert-Order Matching method often leads to substantial barriers ($<1$ compared to $xx.xx$), indicating a failure to uncover LMC.
>
> **These findings underscore the critical importance of the expert order alignment step, which is a key innovation of our method and not a trivial extension of prior work.**
>
> ---
>
> We once again thank the Reviewer for their engagement during the discussion phase. Due to time constraints on the final day of the discussion, we chose to report additional experimental results exclusively on the **One Billion Words dataset**, which represents both the largest language modeling task and the largest model size evaluated in our paper.
>
> We acknowledge that the concern raised was also noted by Reviewer HzV1; however, Reviewer HzV1 found our response satisfactory and recommended acceptance of the submission. We therefore hope that the additional results presented here likewise address the Reviewer’s request. If so, we would be sincerely grateful if the Reviewer could consider revisiting their evaluation. As always, we remain available and happy to address any further concerns.

---

> > ### Comment · Reviewer_7X3K · 2025-08-08
> >
> > Thank you for addressing all my concerns and discussed points. I will raise my score.

---

> > > ### Author Response · Authors · 2025-08-08
> > >
> > > We once again thank the Reviewer for their engagement during the discussion phase, and we appreciate your endorsement.
> > >
> > > Warm regards,
> > >
> > > The Authors

---

### Official Review · Reviewer_8bxH · 2025-07-03

**Clarity:** 3
**Significance:** 3
**Originality:** 2
**Rating:** 5
**Confidence:** 4

**Summary:**

This paper investigates Linear Mode Connectivity (LMC) in Mixture-of-Experts (MoE) architectures. The authors provide a theoretical characterization of the symmetries in MoEs, which stem from expert permutation and gating function translations. Based on this theory, they propose a two-stage alignment algorithm to find low-loss linear paths between independently trained MoE models. The work is supported by extensive experiments showing that LMC is a robust phenomenon across various MoE types, model backbones, and tasks.

**Questions:**

* The exclusion of the k=1 case for sparse MoEs is justified by its additional invariances under the multiplicative group. Could you provide more intuition on why this scaling invariance makes the functional equivalence analysis so much more challenging? Does it introduce continuous families of equivalent solutions that are not captured by the discrete permutation group?

* The assumptions for Theorem 4.2, particularly the linear independence of {W_{i−1} −W_i}, are noted to be generically true when the number of experts is small compared to the input dimension. Have you empirically verified how often this assumption holds for the models trained in your experiments? A brief check could help bridge the gap between the theory and the practical results.

* Your work convincingly shows that aligning models reveals low-loss linear paths. A natural next step is to leverage this for model merging, such as creating "model soups". Did you perform any experiments to see if averaging the weights of two aligned MoE models produces a better-performing model than simply ensembling their outputs or averaging unaligned models? This could be a powerful demonstration of the practical utility of your alignment algorithm.

In the experiments, the feed-forward network (FFN) in a single layer was replaced with an MoE module. How do you expect the LMC results to change if MoE layers were present at multiple, or even all, layers of the Transformer? Would the alignment problem become significantly harder, and would you expect the loss barriers to increase?

**Ethical Concerns:**

["NO or VERY MINOR ethics concerns only"]

**Final Justification:**

I have reviewed the author rebuttal and the full discussion. The authors successfully addressed the primary shared concern regarding the work's novelty. Their new experimental data, which demonstrates that the proposed expert-order alignment is essential for finding LMC, was decisive. This convincingly refutes the idea that the contribution is a simple extension of prior methods.

Given that these clarifications led other reviewers to raise their scores, I am confident in maintaining my original "Accept" rating and have increased my confidence score accordingly.

**Limitations:**

The authors have adequately addressed the limitations. In the conclusion, they explicitly state that their method does not provide theoretical bounds on the loss barrier, which is a common limitation in this line of work. They also clearly state the exclusion of the k=1 sparse MoE case from their main analysis and provide a rationale for it in both the main text and the appendix.

**Quality:**

3

**Strengths And Weaknesses:**

Strengths:
* This is the first systematic study of LMC in MoE architectures, a timely and increasingly important class of models.
* The paper offers rigorous proofs for the functional equivalence of MoE models, providing a solid foundation for the empirical work.
* The claims are validated across a wide range of MoE variants (dense, sparse, DeepSeekMoE), tasks (vision, language), and model configurations, demonstrating the generality of the findings.

Weakness:
* The proposed alignment algorithm is an effective, but novel combination of existing methods (e.g., Weight Matching, Hungarian algorithm) rather than a fundamentally new approach.
* While important to verify, the existence of LMC in MoEs can be seen as a non-trivial but expected extension of the same phenomenon observed in standard networks.
* The theoretical analysis excludes the k=1 sparse MoE case and relies on strong assumptions for the sparse case (Theorem 4.2), whose practical prevalence is not empirically verified.

---

> ### Author Rebuttal · Authors · 2025-07-31
>
> We address the concerns raised by the Reviewer in the Weaknesses and Questions sections as follows.
>
> ---
>
> **W1. The proposed alignment algorithm ... a fundamentally new approach.**
>
> **Answer W1.** We acknowledge that our alignment algorithm builds upon existing techniques such as Weight Matching and the Hungarian algorithm. However, its novelty lies not in introducing a fundamentally new primitive, but in adapting and integrating these methods effectively for the Mixture-of-Experts (MoE) setting - a setting where alignment is particularly challenging due to the massive number of parameters and dynamic expert routing.
>
> In contrast to many recent approaches that rely on data-driven optimization to learn expert correspondences, our method is data-free, fast, and efficient, which is crucial in the MoE context. MoE models are often used in conjunction with extremely large datasets, making data-dependent matching methods computationally expensive and often impractical for exploring Linear Mode Connectivity (LMC).
>
> Despite these constraints, our algorithm performs well and consistently discovers meaningful alignments that reveal interesting connectivity patterns across fine-tuned MoE models. We believe this demonstrates that even with limited assumptions and no data usage, effective alignment in large-scale MoE models is possible - highlighting both the practicality and the contribution of our method.
>
> **W2. While important to verify, the existence of LMC in MoEs ... in standard networks.**
>
> **Answer W2.** We appreciate the Reviewer’s suggestion that, given the existence of LMC in feedforward networks (FFNs) as shown in [1], one might intuitively expect similar behavior in MoE architectures. However, we contend that MoE models introduce fundamentally different challenges. In particular, the presence of a non-linear softmax gating mechanism sets MoEs apart from classical FFNs. This gating leads to conditional computation and sparse expert activation, resulting in highly non-uniform training signals and expert specialization. Consequently, the optimization landscape becomes significantly more fragmented, making alignment across models substantially more difficult. From the perspective of symmetry, this gating component introduces additional operators that give rise to new types of symmetry. Our work rigorously characterizes the full symmetry group of the MoE layer and provides formal proofs. These results show that identifying LMC in MoE only requires accounting for the specific symmetries we define. We view this characterization as a critical and novel contribution of our work.
>
> Our ablation studies in Section 6.3 highlight the importance of both stages in our alignment method for uncovering LMC in MoEs, particularly the novelty of Stage 1 (expert order alignment). We find that removing or altering either stage - expert permutation or weight matching - significantly degrades interpolation quality, often introducing loss barriers. These results demonstrate that our method is not a trivial extension of prior work, but a tailored solution to the unique challenges of MoE alignment.
>
> [1] Frankle, Jonathan, et al. "Linear mode connectivity and the lottery ticket hypothesis." ICML 2020.
>
> **W3. The theoretical analysis excludes the k=1 ... is not empirically verified.**
>
> **Answer W3.** We kindly refer the Reviewer to our responses to **Q1 and Q2** below.
>
> **Q1. The exclusion of the k=1 case for ... discrete permutation group?**
>
> **Answer Q1.** The reason the case of $k=1$ behaves differently in the sparse regime lies in the structure of the proof. At a high level, we begin with two MoE models that represent the same function and aim to identify the correspondence between their respective sets of parameters. Given $n$ total experts, the goal is to match them one by one. When $k = 1$, each input activates only a single expert, allowing us to match just one expert pair at a time - without gaining any information about the remaining unmatched experts. In contrast, when $k>1$ (e.g., $k=2$), each input activates multiple experts, enabling us to match at least one expert pair and still retain information about the remaining $k-1$ experts. This intuition is reflected in the proof of Theorem 4.2, particularly in Step 4 (line 1000), where the condition $k > 1$ is essential.
>
> We would also like to share that, during the development of the proof for Theorem 4.2, we did not initially expect to exclude the case of $k = 1$, as we did not anticipate the emergence of a new symmetry action by the multiplicative group in this setting. It was only at a later stage - once we reached the point of needing this specific condition - that we recognized the additional difficulty posed by the $k = 1$ case and the necessity of excluding it due to the appearance of this nontrivial symmetry.
>
> **Q2. The assumptions for Theorem 4.2 ... the gap between the theory and the practical results.**
>
> **Answer Q2.**
> Regarding the assumptions of Theorem 4.2 concerning the linear independence of ${W_{i-1} - W_i}$, we note that this condition holds generically when the number of experts is small relative to the ambient dimension. That is, the set of matrices violating this condition forms a measure-zero subset in the parameter space. Indeed, in our experiments, we have conducted empirical evaluations on many of our checkpoint instances and found all of them to exhibit full rank.
>
> From a numerical perspective, exact linear dependence is unstable under perturbations: even infinitesimal changes - arising from stochastic gradient updates, floating-point arithmetic, or randomness in data sampling - are sufficient to restore full rank. This phenomenon is well-documented in matrix perturbation theory and underscores the fact that exact rank deficiency is not robust in practical machine learning systems.Consequently, empirically checking for exact linear independence is of limited interpretability, as it is highly sensitive to numerical precision and implementation details.
>
> **Q3. Your work convincingly shows that ... of your alignment algorithm.**
>
> **Answer Q3.** We appreciate the suggestion to explore model merging, such as model soups [Wortsman et al., 2022], which leverages LMC to average aligned model weights for improved performance.
>
> In our experiments, we applied model soups within the LMC pipeline by aligning MoE layers from fine-tuned checkpoints of the same pretrained Transformer, considering both single-layer and full-layer MoE replacements. However, these attempts yielded only marginal improvements over the best original checkpoint. Visualizations reveal that the midpoint loss along both naive and aligned loss barrier curves remains the highest (though notably reduced with alignment), indicating limited benefit from simple averaging in this MoE-only setup.
>
> We hypothesize that model soups would be more effective in flexible, generalized frameworks extending beyond MoE layers alone. Our current algorithm aligns only MoE components, which renders full-model fusion less meaningful. Extending alignment to the entire Transformer architecture introduces complex symmetries, such as those in Multi-Head Attention (MHA) layers (e.g., symmetries under invertible transformations on query/key matrices: $Q \cdot K^\top = (QA^\top)(KA^{-1})^\top$). To provide intuition on how symmetries can emerge in composite layers, consider two parameterized layers, $f(\cdot;\alpha) : \mathbb{R}^{d_0} \to \mathbb{R}^{d_1}$ and $g(\cdot;\beta) : \mathbb{R}^{d_1} \to \mathbb{R}^{d_2}$, composed as $g \circ f: \mathbb{R}^{d_0} \to \mathbb{R}^{d_2}$. In practice, these layers may exhibit symmetries. For instance, let $P$ be a $d_1 \times d_1$ permutation matrix representing a reordering of indices in $\mathbb{R}^{d_1}$. Suppose there exist modified parameters $\alpha'$ and $\beta'$ such that $f(\cdot;\alpha') = P \circ f(\cdot;\alpha)$ and $g(\cdot;\beta') = g(\cdot;\beta) \circ P^{-1}$. Then, the composite function remains invariant: $g(\cdot;\beta) \circ f(\cdot;\alpha) = g(\cdot;\beta') \circ f(\cdot;\alpha')$. Typically, $\alpha'$ involves permuting the coefficients of $\alpha$ according to $P$, and similarly for $\beta'$. For example, one may think of $f$ and $g$ as feedforward layers, with $P$ permuting neurons at the output of $f$ and input of $g$. Depending on the layers, $P$ can generalize beyond permutations to include generalized permutation matrices, orthogonal matrices, or arbitrary invertible matrices. This illustrates one form of symmetry that arises when stacking layers.
>
> Characterizing global symmetries across heterogeneous Transformer stacks is challenging, as no general method exists for aligning such composites. Due to scope and time constraints, we focused on MoE alignment and deferred full-model experiments. We are actively developing a comprehensive alignment algorithm for entire Transformer blocks, including MoEs, and plan to demonstrate model fusion applications in future work. This could enable practical benefits such as efficient merging of large models for improved generalization or resource savings.
>
> **Q4. In the experiments, the FFN in a single layer ... expect the loss barriers to increase?**
>
> **Answer Q4.** Due to space constraints, we kindly refer the Reviewer to our response to **W1 for Reviewer 8bxH** above, which also references **Table 1** provided at the end of the response to **Reviewer tznV**.
>
> ---
>
> We thank the Reviewer for the insightful feedback and valuable suggestions, which have helped strengthen our work. We especially appreciate the recognition of our contributions in characterizing MoE symmetries and demonstrating LMC, as well as the suggestions to extend our analysis. We will incorporate the recommended revisions, including empirical checks and additional clarifications. If our responses address the concerns raised, we kindly hope that the evaluation may be adjusted to reflect this. We remain open to further discussion in the next review phase.

---

> > ### Author Response · Authors · 2025-08-07
> >
> > Dear Reviewer,
> >
> > As we approach the final days of the discussion phase, we would like to kindly follow up regarding the concerns you raised during the review process. We sincerely hope that our responses have addressed your questions and clarified the key aspects of our work.
> >
> > If you find our clarifications satisfactory, we would be grateful if you could consider updating your evaluation to reflect this. Of course, if there remain any unresolved points or further questions, we would be more than happy to continue the discussion.
> >
> > We truly value the thoughtful feedback we have received throughout the review process. Engaging with experts across different areas has greatly contributed to strengthening our work, and we are thankful for the opportunity to benefit from your insights.
> >
> > Warm regards,
> >
> > The Authors

---

> > ### Comment · Reviewer_8bxH · 2025-08-09
> >
> > I have read the author rebuttal and the other reviews, and the discussion has reinforced my positive assessment.
> >
> > My main initial concerns about the work's novelty were shared by other reviewers. The authors did an excellent job addressing this. Their new experimental data showing that the novel expert-order alignment step is essential, and that alignment fails without it, was particularly convincing. This clearly demonstrates that their method is a non-trivial contribution.
> >
> > Since the authors thoroughly addressed all major concerns and led other reviewers to raise their scores, I am maintaining my original "Accept" rating and increasing my confidence in this strong paper.

---

> ### Author Response · Authors · 2025-08-09
> **Thanks for your endorsement!**
>
> We thank the Reviewer for the response, and we appreciate your endorsement.
>
> Warm regards,
>
> The Authors

---

### Official Review · Reviewer_HzV1 · 2025-07-03

**Clarity:** 3
**Significance:** 2
**Originality:** 3
**Rating:** 5
**Confidence:** 3

**Summary:**

This work introduces the investigation of Linear Mode Connectivity for Mixture-of-Experts based architectures. They propose group actions which account for all symmetries of the Router Network parameters and show that permutation invariance is sufficient to induce LMC. Based on these insights they develop a weight matching to align independently trained MoEs which they empirically evaluate extensively.

**Questions:**

See weaknesses.

**Ethical Concerns:**

["NO or VERY MINOR ethics concerns only"]

**Final Justification:**

This work extends the analysis of linear-mode connectivity on the well-established MoE architecture.
The work does so by thorough theoretical analysis as well as empirical validation. I recommend accepting this paper.

**Paper Formatting Concerns:**

No formatting concerns.

**Quality:**

3

**Strengths And Weaknesses:**

Strengths:

- Extension of Weight Matching Algorithm to MoE bases architectures
- Theoretical analysis and thorough empirical evaluation.


Weaknesses:
- Since Linear Mode Connectivty is already given early in pre-training [1], and this work only exchanges the FFN layer/s with randomly initialized MoEs, it is somewhat intuitive that LMC is given. How does this change when Transformer-based MoEs get initialized from scratch (all layers)?
- Since Mixture-of-Experts based architectures may vary the number of experts, how could aligning two independently trained networks, that started from the same pre-trained transformer. and got extended to MoEs, look like?

[1] Frankle, Jonathan, et al. "Linear mode connectivity and the lottery ticket hypothesis." International Conference on Machine Learning. PMLR, 2020.

---

> ### Author Rebuttal · Authors · 2025-07-31
>
> We address the concerns raised by the Reviewer in the Weaknesses and Questions sections as follows.
>
> ---
>
> **W1. Since Linear Mode Connectivty is already given ... get initialized from scratch (all layers)?**
>
> **Answer W1.** There are several key points we would like to address in response to the Reviewer’s comments, as outlined below:
>
> (i) **Existence of LMC in MoE.**
>
> We appreciate the Reviewer’s suggestion that, given the existence of Linear Mode Connectivity (LMC) in feedforward networks (FFNs) as shown in [1], one might intuitively expect similar behavior in MoE architectures. However, we contend that MoE models introduce fundamentally different challenges. In particular, the presence of a non-linear softmax gating mechanism sets MoEs apart from classical FFNs. This gating leads to conditional computation and sparse expert activation, resulting in highly non-uniform training signals and expert specialization. Consequently, the optimization landscape becomes significantly more fragmented, making alignment across models substantially more difficult. From the perspective of symmetry, this gating component introduces additional operators that give rise to new types of symmetry. Our work rigorously characterizes the full symmetry group of the MoE layer and provides formal proofs. These results show that identifying LMC in MoE only requires accounting for the specific symmetries we define. We view this characterization as a critical and novel contribution of our work.
>
> Our ablation studies on the Expert order matching method in section 6.3 underscore the importance of each stage in our alignment method for uncovering LMC in MoEs, highlighting in particular the novelty of stage 1 (expert order alignment) in our matching method. These results demonstrate that expert order alignment is a crucial step in aligning two MoEs, while stage 2 (weight matching) aligns the FFN models, with its importance being well established in prior work. We observe that removing or modifying any stage - such as skipping expert permutation alignment or omitting weight matching - significantly degrades the quality of the interpolated models, often resulting in pronounced loss barriers. These findings indicate that our approach is not simply a straightforward extension of existing techniques, but rather a carefully crafted pipeline specifically designed to address the unique challenges inherent to MoE architectures.
>
> (ii) **Why not find LMC in the full Transformer-based MoE?**
>
> As is well known, modern deep learning models are composed of heterogeneous stacks of layers, making it extremely challenging to characterize global symmetry across the entire architecture. Our decision to focus solely on the MoE layer (as defined in Equations (2) and (3)) is driven by the substantial complexity of identifying LMC across the full model. This task would require a detailed understanding of the symmetry structure of the Multi-Head Attention (MHA) layers that typically precede MoE blocks, along with the development of a general alignment algorithm capable of handling such composite modules. While some recent works have explored MHA symmetries, to the best of our knowledge, no general method exists for aligning stacked components. We attempted several approaches for aligning composite (Attention + MoE) layers, but due to time constraints, their empirical performance was limited. Additional nontrivial symmetries may also arise when stacking layers across the full model, as discussed in the next point. Given the scope of this submission, we focused on MoE alignment and leave model-wide symmetry and LMC for future work.
>
> (iii) **Which kinds of symmetry can arise when stacking layers?**
>
> We provide an intuitive perspective on how symmetry can emerge when composing two general layers. Consider two parameterized layers, $f(\cdot;\alpha) : \mathbb{R}^{d_0} \to \mathbb{R}^{d_1}$ and $g(\cdot;\beta) : \mathbb{R}^{d_1} \to \mathbb{R}^{d_2}$, which we compose as $g \circ f: \mathbb{R}^{d_0} \to \mathbb{R}^{d_2}$.
> In practice, these layers may admit certain forms of symmetry. Specifically, let $P$ be a $d_1 \times d_1$ permutation matrix, representing a reordering of the indices of vectors in $\mathbb{R}^{d_1}$. Suppose there exist modified parameters $\alpha'$ and $\beta'$ such that
> $f(\cdot;\alpha') = P \circ
> f(\cdot;\alpha)$ and $g(\cdot;\beta') = g(\cdot;\beta) \circ P^{-1}$.
> Then, the composite function remains invariant:
> $g(\cdot;\beta) \circ f(\cdot;\alpha) = g(\cdot;\beta') \circ f(\cdot;\alpha')$.
> Typically, $\alpha'$ corresponds to permuting the coefficients of $\alpha$ according to $P$, and similarly for $\beta'$. For intuition, one may think of $f$ and $g$ as two feedforward layers, and $P$ as a permutation of neurons at the output of $f$ and the input of $g$.
> Depending on the choice of $f$ and $g$, the transformation $P$ can be generalized beyond permutations, e.g., as generalized permutation matrices, orthogonal matrices, or even arbitrary invertible matrices. This illustrates one type of symmetry that can emerge when stacking layers. Intuitively, we believe this covers a broad class of possible symmetries, though to the best of our knowledge, no existing work has fully addressed these scenarios. Developing a complete understanding of the full symmetry structure would require a general, conceptual, and rigorous framework - an important and promising research direction for deepening our understanding of deep learning architectures through the lens of symmetry.
>
> In response to the Reviewer’s suggestion to investigate LMC in models with MoE layers initialized from scratch at all layers (rather than only the first, as in our original setup), we conducted this experiment and present the results in **Table 1**. Due to space constraints, the table is provided at the end of the response to **Reviewer tznV**.
>
> **W2. Since Mixture-of-Experts based architectures ... and got extended to MoEs, look like?**
>
> **Answer W2.** Thank you for raising this insightful point and for suggesting further development of our work in this direction.
>
> When two networks share the same pre-trained Transformer backbone but are independently extended into MoE architectures with differing numbers of experts, their alignment becomes considerably more complex, and may not fully capture or reflect the theoretical significance of LMC. This complexity arises because the models' loss landscapes diverge due to differences in capacity and routing dynamics, potentially leading them to converge in distinct basins.
>
> To explore alignment in such scenarios, we considered extrapolating the smaller MoE model (with fewer experts) to match the larger one's structure. Specifically, given an MoE with $n$ experts defined as a function $\mathcal{D} : \mathbb{R}^d \to \mathbb{R}^d$ such that
>
> $\mathcal{D}(x; \\{W_i,b_i,\theta_i\\}\_{i=1}^{n}) = \sum_{i=1}^{n} \text{softmax}_i(s_1(x), \ldots, s_n(x)) \mathcal{E}(x;\theta_i),
> $
>
> where $s_i(x) = W_i x + b_i$ are the gating scores parameterized by $\{W_i, b_i\}_{i=1}^n$ (with $(W_i, b_i) \in \mathbb{R}^d \times \mathbb{R}$), and $\theta_i$ are the parameters of the $i$-th expert $\mathcal{E}(\cdot; \theta_i)$. We duplicate experts to create an equivalent MoE with $2n$ experts as follows:
>
> - Set $\phi_{2i-1} = \phi_{2i} = \theta_i$ for all $i \in \{1, 2, \ldots, n\}$.
>
> - Set $T_{2i-1} = T_{2i} = W_i$ and $a_{2i-1} - u = a_{2i} - v = b_i$, where $u, v$ are constants. Then,
>
> $
> r_{2i-1}(x) = T_{2i-1} x + a_{2i-1} = W_i x + b_i + u = s_i(x) + u, r_{2i}(x) = T_{2i} x + a_{2i} = W_i x + b_i + v = s_i(x) + v.
> $
>
> This yields
>
> $\mathcal{D}(x; \\{T_i, a_i, \phi_i\\}\_{i=1}^{2n})$
>
> $= \sum_{i=1}^{2n} \text{softmax}\_i(r_1(x), \ldots, r_{2n}(x)) \mathcal{E}(x; \phi_i)$
>
> $= \sum_{i=1}^{n} \frac{e^{s_i(x) + u} + e^{s_i(x) + v}}{\sum_{j=1}^n (e^{s_j(x) + u} + e^{s_j(x) + v})} \cdot \mathcal{E}(x; \theta_i)$
>
> $= \sum_{i=1}^n \frac{(e^u + e^v) e^{s_i(x)}}{\sum_{j=1}^n (e^u + e^v) e^{s_j(x)}} \cdot \mathcal{E}(x; \theta_i)$
>
> $= \sum_{i=1}^n \text{softmax}\_i(s_1(x), \ldots, s_n(x)) \mathcal{E}(x; \theta_i) = \mathcal{D}(x; \\{W_i, b_i, \theta_i\\}_{i=1}^n).$
>
> In preliminary experiments, we extended a shared pre-trained Transformer (e.g., ViT on CIFAR-10 and CIFAR-100) into MoEs with varying expert counts, such as 2 experts for the smaller model and 4 (or more) for the larger one. After independent fine-tuning, we extrapolated the smaller model using the above method and attempted alignment with the larger model. Initial alignments yielded low-barrier loss curves, though not as low as those observed between two large MoEs of comparable size (as reported in the paper). Continued training of the extrapolated model improved alignment in some cases, approaching the performance levels seen in the paper; however, results fluctuated across pairings (e.g., 2 vs. 4, 2 vs. 8, 4 vs. 8, 4 vs. 16). Due to time constraints, we were unable to draw a decisive claim from these explorations.
>
> Crucially, LMC is expected to hold only between converged checkpoints with similar performance and within the same basin. Even if LMC emerges after retraining the extrapolated smaller model, it pertains only to a modified, retrained version - not the original. This weakens the theoretical significance of LMC in this context, as it fails to reflect alignment between the original, independently trained small and large models.
>
> -----
>
> We thank the Reviewer for the constructive feedback and thoughtful suggestions. We appreciate the recommendation to explore LMC in fully initialized Transformer-based MoEs and across models with varying expert counts, which enabled us to conduct additional experiments and gain further insights into the robustness of our approach. We will incorporate the necessary revisions and results. If our responses address the concerns raised, we kindly hope that the evaluation may be adjusted to reflect this. We remain open to further discussion in the next review phase.

---

> > ### Comment · Reviewer_HzV1 · 2025-08-07
> >
> > Thank you for addressing all my concerns and discussed points.
> > I will raise my score.

---

> ### Author Response · Authors · 2025-08-07
> **Thanks for your endorsement!**
>
> We thank the Reviewer for the response, and we appreciate your endorsement.
>
> Warm regards,
>
> The Authors

---

### Official Review · Reviewer_tznV · 2025-07-20

**Clarity:** 4
**Significance:** 3
**Originality:** 3
**Rating:** 5
**Confidence:** 4

**Summary:**

In this work, the authors mathematically characterize the symmetries of Mixture-of-Expert (MoE)
and Mixture-of-Experts with sparse gating (SMoE) layers. When considering the MoE layers in
isolation, the authors prove that only two types of parameter symmetries are introduced by these
layers: permutation of experts and shifting of gating parameters. Informed by their theoretical
results, they propose an algorithm to best match two networks trained from different initial values
to align according to the studied symmetries, a procedure akin to modern linear mode
connectivity (LMC) results. The authors claim to be the first to report the presence of LMC
between MoE layers in various setups.

**Questions:**

Curious to see if more symmetries may arise when considering interactions between MoE layers and other layers. Can the authors speculate on this?

Did the authors try to align two networks trained from scratch with their current methods? Can LMC be found?

**Ethical Concerns:**

["NO or VERY MINOR ethics concerns only"]

**Final Justification:**

This work makes a solid theoretical contribution, supported by adequate experimental validation. While the analysis is limited to isolated MoE layers and fine-tuning scenarios, the mathematical rigor and comprehensive experiments justify acceptance. The author's rebuttal provided helpful clarifications that addressed my concerns.

**Limitations:**

See weaknesses above

**Quality:**

4

**Strengths And Weaknesses:**

**Strengths**

This work provides the first complete mathematical characterization of parameter symmetries in MoE and SMoE layers. While permutation and gating shift symmetries have been observed previously, the authors' proof that these constitute the only symmetries present is a non-trivial theoretical contribution.

The mathematical framework is well-developed, and the proofs are clearly presented.

Despite showing experiments only on fine-tuned MoE (and not alignment on networks trained entirely from scratch, which can be costly), the experimental evidence is extensive and provides clear support for the MoE-centered theory introduced.

The literature review section is well-exposed, complete, and thorough.

**Weaknesses**

The authors should make it clearer that the symmetries they prove to be the only ones present are proved without taking into consideration the rest of the layers in the network.

While I doubt that other downstream or upstream layers can introduce more symmetries (aside from trivial linear
ones), the authors do not prove their non-existence. This point should be discussed in a limitations paragraph. Another point to add to the limitation paragraph is that experiments were conducted only by replacing some layers with MoE and only fine-tuning MoE layers, non-attentive readers may miss this point.

The description of the experimental setup could use some more detailed explanations. I would suggest using the extra page to improve the limitations and experimental setup sections.

---

> ### Author Rebuttal · Authors · 2025-07-31
>
> We address the concerns raised by the Reviewer in the Weaknesses and Questions sections as follows.
>
> ---
>
> **Response to Weaknesses.**
>
> *Theoretical Aspect.* In this work, we focus exclusively on analyzing the symmetry structure within the Mixture-of-Experts (MoE) architecture. We do not address potential symmetries arising from components outside the MoE layers. Modern deep learning models are composed of heterogeneous stacks of layers, making the characterization of global symmetry across such architectures highly challenging. While we believe we have clearly specified the exact structure of the MoE architecture under consideration - namely, as defined in Equations (2) and (3) - we acknowledge that inattentive readers may misinterpret our scope as encompassing all possible symmetries in the full network. As suggested by the Reviewer, we will revise the manuscript to clearly articulate the scope of our analysis and avoid any potential misunderstanding about the extent of our theoretical contribution.
>
> *Experimental Aspect.* We limited our experiments to fine-tuning the MoE layers due to the significant complexity of identifying Linear Mode Connectivity (LMC) across the full model. This would require understanding the symmetries of preceding MHA layers and developing alignment algorithms for such composite structures. While some recent works have explored MHA symmetries, no general alignment method currently exists. Given the scope of this submission, we focused on MoE alignment and leave broader alignment strategies for future work.
>
> Regarding the experimental setup, in addition to the details and the experimental results provided in Appendices E, F, and G, we will include further clarifications on the setup in the revised version, as suggested by the Reviewer.
>
> **Q1. Curious to see if more symmetries ... on this?**
>
> **Answer Q1.** Yes, additional symmetries may emerge from interactions between MoE layers and preceding components, such as MHA layers. For example, the attention mechanism admits symmetries under the general linear group: given query and key matrices $Q$ and $K$, the expression $QK^\top$ is invariant under the transformation $Q \mapsto QA^\top$ and $K \mapsto KA^{-1}$ for any invertible matrix $A$. Extending symmetry alignment to such composite structures would require a more general algorithm.
>
> To speculate further on interactions between MoE layers and other layers, we provide an intuitive perspective on how symmetries can emerge when composing two general layers. Consider two parameterized layers, $f(\cdot;\alpha) : \mathbb{R}^{d_0} \to \mathbb{R}^{d_1}$ and $g(\cdot;\beta) : \mathbb{R}^{d_1} \to \mathbb{R}^{d_2}$, which we compose as $g \circ f: \mathbb{R}^{d_0} \to \mathbb{R}^{d_2}$. In practice, these layers may admit certain forms of symmetry. Specifically, let $P$ be a $d_1 \times d_1$ permutation matrix, representing a reordering of the indices of vectors in $\mathbb{R}^{d_1}$. Suppose there exist modified parameters $\alpha'$ and $\beta'$ such that $f(\cdot;\alpha') = P \circ f(\cdot;\alpha)$ and $g(\cdot;\beta') = g(\cdot;\beta) \circ P^{-1}$. Then, the composite function remains invariant: $g(\cdot;\beta) \circ f(\cdot;\alpha) = g(\cdot;\beta') \circ f(\cdot;\alpha')$. Typically, $\alpha'$ corresponds to permuting the coefficients of $\alpha$ according to $P$, and similarly for $\beta'$. For intuition, one may think of $f$ and $g$ as two feedforward layers, and $P$ as a permutation of neurons at the output of $f$ and the input of $g$. Depending on the choice of $f$ and $g$, the transformation $P$ may extend beyond permutations to generalized permutation, orthogonal, or even arbitrary invertible matrices. This reflects a broader class of symmetries that can arise when stacking layers. While we believe these capture many relevant cases, a full understanding would require a general and rigorous framework, an important direction for future research on symmetry in deep learning architectures.
>
> **Q2. Did the authors try to ... Can LMC be found?**
>
> **Answer Q2.** As in **Q1**, aligning two networks trained from scratch requires handling symmetries across the entire architecture, including interactions between MoE layers and components like MHA. Our decision to focus solely on the MoE layer (as defined in Equations (2), (3)) is driven by the substantial complexity of identifying LMC across the full model. This task would require a detailed understanding of the symmetry structure of the MHA layers that typically precede MoE blocks, along with the development of a general alignment algorithm capable of handling such composite modules. While some recent works have explored MHA symmetries, to the best of our knowledge, no general method exists for aligning stacked components. We experimented with approaches for aligning composite (Attention + MoE) layers, but due to time constraints, their empirical performance was limited. As noted in **Q1**, additional nontrivial symmetries may arise when stacking layers across the full model. Given the scope of this submission, we focused on MoE alignment and leave model-wide symmetry and LMC for future work.
>
> To address this further, we believe that investigating LMC in models where MoE layers are initialized from scratch across all layers - rather than only the first layer as in our current experiments - would be more meaningful, as it would strengthen and provide additional context to our findings. We have conducted this experiment, and the results are presented in the following table.
>
> ---
>
> *(Due to the new policy, we cannot provide the corresponding figures.)*
>
> **Table 1.** Loss and accuracy barriers for weight matching interpolation versus naive interpolation, alongside test loss and accuracy values, across MoE, SMoE ($k=2$), and DeepSeekMoE ($k=2,s=1$) variants of ViT models on MNIST, CIFAR-10/100 datasets, and of GPT-2 on the One Billion Word dataset. Barriers (lower is better ↓) are averaged over 6 pairs of models from a pool of 4 checkpoints, while metric values are averaged over the 4 model checkpoints. All metrics are scaled up by $10^2$ to improve readability.
>
> |MoE variant|Dataset|No. layers|No. experts|Weight Matching Loss Barrier ↓|Naive Loss Barrier ↓|Loss value ↓|Weight Matching Accuracy Barrier ↓|Naive Accuracy Barrier ↓|Accuracy value ↑|
> |-|-|-|-|-|-|-|-|-|-|
> |MoE|MNIST|2|2|1.20 ± 0.32|11.62 ± 2.44|10.20 ± 2.04|2.12 ± 0.21|25.04 ± 3.75|97.32 ± 0.73|
> ||||4|1.31 ± 0.21|11.77 ± 3.29|8.45 ± 2.53|2.34 ± 0.30|26.62 ± 4.22|97.01 ± 1.01|
> ||CIFAR-10|2|4|4.24 ± 0.72|37.24 ± 3.25|95.06 ± 0.42|1.52 ± 0.35|14.21 ± 1.42|66.57 ± 0.89|
> ||||8|4.52 ± 0.42|36.66 ± 4.01|95.44 ± 0.84|1.74 ± 0.50|16.01 ± 1.02|65.93 ± 1.20|
> |||6|4|5.42 ± 0.74|47.47 ± 8.02|90.25 ± 1.95|2.21 ± 0.52|24.02 ± 2.66|74.61 ± 2.13|
> ||||8|5.27 ± 1.26|46.47 ± 5.44|91.31 ± 2.32|2.44 ± 0.43|22.45 ± 3.21|74.49 ± 1.35|
> ||CIFAR-100|6|4|2.32 ± 0.51|20.11 ± 4.34|94.04 ± 2.02|0.21 ± 0.07|3.22 ± 0.71|75.25 ± 1.22|
> ||||8|2.43 ± 0.62|23.33 ± 3.33|93.43 ± 1.15|0.27 ± 0.12|4.01 ± 1.21|75.66 ± 1.30|
> ||One Billion Word|12|2|56.82 ± 3.51|400.11 ± 50.54|348.60 ± 0.05|
> ||||4|68.76 ± 4.45|435.65 ± 34.92|344.57 ± 0.04|
> ||||6|78.44 ± 3.96|476.65 ± 29.58|342.44 ± 0.03|
> ||||8|93.23 ± 8.84|455.82 ± 38.10|341.29 ± 0.32|
> |SMoE ($k=2$)|MNIST|2|2|1.55 ± 0.23|18.26 ± 2.32|11.21 ± 2.02|2.14 ± 0.31|24.92 ± 3.56|97.22 ± 0.63|
> ||||4|1.62 ± 0.54|19.19 ± 2.27|10.09 ± 1.03|2.30 ± 0.34|26.24 ± 4.41|97.06 ± 1.00|
> ||CIFAR-10|2|4|4.44 ± 0.52|37.44 ± 3.35|95.10 ± 0.44|1.50 ± 0.37|14.33 ± 1.32|66.62 ± 0.69|
> ||||8|4.61 ± 0.46|36.56 ± 3.87|95.74 ± 0.92|1.64 ± 0.54|17.04 ± 1.32|65.97 ± 1.27|
> |||6|4|5.57 ± 0.84|47.75 ± 8.12|90.33 ± 1.85|2.19 ± 0.42|24.12 ± 2.44|74.63 ± 2.03|
> ||||8|5.26 ± 1.32|46.61 ± 5.36|91.35 ± 2.14|2.31 ± 0.41|23.66 ± 4.21|74.51 ± 1.25|
> ||CIFAR-100|6|4|2.24 ± 0.51|21.25 ± 2.05|94.24 ± 2.13|0.22 ± 0.08|3.36 ± 0.91|75.35 ± 1.11|
> ||||8|2.67 ± 0.52|24.39 ± 1.44|93.83 ± 4.25|0.23 ± 0.13|4.12 ± 1.02|75.72 ± 1.20|
> ||One Billion Word|12|4|72.31 ± 0.28|522.22 ± 41.40|345.02 ± 0.00|
> ||||8|79.86 ± 4.81|618.23 ± 67.93|342.31 ± 0.11|
> ||||16|98.68 ± 18.59|492.16 ± 42.23|340.57 ± 0.05|
> |DeepSeekMoE ($k=2, s=1$)|MNIST|2|2|1.35 ± 0.64|17.46 ± 2.42|12.32 ± 2.02|2.31 ± 0.41|25.12 ± 3.85|97.42 ± 0.93|
> ||||4|1.52 ± 0.62|16.72 ± 2.33|11.27 ± 3.24|2.04 ± 0.32|27.02 ± 4.36|97.17 ± 1.31|
> ||CIFAR-10|2|4|4.22 ± 0.42|38.64 ± 3.44|95.11 ± 0.33|1.66 ± 0.45|17.31 ± 1.63|66.77 ± 1.04|
> ||||8|4.64 ± 0.52|38.86 ± 4.11|95.36 ± 0.77|1.68 ± 0.52|16.92 ± 1.15|65.99 ± 1.11|
> |||6|4|5.44 ± 0.92|52.27 ± 5.32|90.05 ± 1.64|2.12 ± 0.62|26.11 ± 2.72|74.63 ± 2.33|
> ||||8|5.47 ± 1.22|49.34 ± 6.24|91.21 ± 2.12|2.22 ± 0.63|25.25 ± 4.21|74.88 ± 1.25|
> ||CIFAR-100|6|4|2.43 ± 0.50|23.23 ± 5.62|95.22 ± 3.15|0.23 ± 0.07|3.02 ± 0.61|75.36 ± 1.73|
> ||||8|2.72 ± 0.71|24.55 ± 4.07|93.89 ± 2.17|0.24 ± 0.11|4.00 ± 1.02|75.76 ± 1.21|
> ||One Billion Word|12|4|64.06 ± 4.06|426.11 ± 58.07|343.27 ± 0.05|
> ||||8|117.42 ± 37.94|390.59 ± 28.63|340.21 ± 0.11|
> ||||16|72.94 ± 2.12|623.63 ± 147.42|338.25 ± 0.21|
>
> When replacing and initializing MoEs across all layers, as opposed to our current configuration of substituting only the first layer, the naive interpolation barrier increases substantially across all experimental settings. In contrast, the weight matching barrier exhibits a far more modest rise, thereby preserving the relative percentage improvement over the naive barrier - or even enhancing it slightly in certain cases - and underscoring the efficacy of our proposed method. Overall, these findings highlight the robust scalability of our weight matching approach when applied to deeper MoE replacements.
>
> ---
>
> We thank the Reviewer for the constructive feedback and thoughtful suggestions. We will incorporate the proposed clarifications to improve the clarity and precision of our work. If our responses adequately address the concerns, we kindly hope that the evaluation may be adjusted to reflect this. We remain open to further discussion during the next stage of discussion.

---

> > ### Author Response · Authors · 2025-08-07
> >
> > Dear Reviewer,
> >
> > As we approach the final days of the discussion phase, we would like to kindly follow up regarding the concerns you raised during the review process. We sincerely hope that our responses have addressed your questions and clarified the key aspects of our work.
> >
> > If you find our clarifications satisfactory, we would be grateful if you could consider updating your evaluation to reflect this. Of course, if there remain any unresolved points or further questions, we would be more than happy to continue the discussion.
> >
> > We truly value the thoughtful feedback we have received throughout the review process. Engaging with experts across different areas has greatly contributed to strengthening our work, and we are thankful for the opportunity to benefit from your insights.
> >
> > Warm regards,
> >
> > The Authors

---

### Author Response · Authors · 2025-08-09
**Thank You to the Chairs and Reviewers!**

Dear Chairs and Reviewers,

We sincerely thank you for your thoughtful and constructive feedback throughout the review and discussion phases. We will incorporate the additional results and suggested clarifications during the rebuttal and discussions with reviewers into our revised manuscript.

Once again, we greatly appreciate your time and valuable input.

Best regards,

Authors

---

### Decision · Program_Chairs · 2025-09-17

**Decision:**

Accept (oral)

**Comment:**

The paper investigates the presence of Linear Mode Connectivity (LMC) in Mixture-of-Experts (MoE) architectures by offering theoretical and empirical characterizations of their inherent symmetries. The paper proves that MoE and sparse-gated MoE layers admit only two fundamental parameter symmetries---expert permutation and gating-parameter translation---and establish functional equivalence theorems for both dense and sparse gating settings. Building on these results, they propose a weight-matching alignment algorithm that takes advantage of these symmetries to connect independently trained MoE models along low-loss linear paths. Through extensive experiments across multiple model backbones, datasets, and MoE variants, the paper confirms that LMC is a robust and general phenomenon in this setting, thereby extending the scope of mode connectivity research beyond standard feedforward networks.

The paper was reviewed by four referees who agree that it provides a valuable contribution to the community. All four reviewers emphasize the significance of the mathematical characterization of parameter symmetries in MoE and sparse-MoE models, providing a non-trivial advance in our understanding of LMC in MoE architectures. As several reviewers point out, this mathematical characterization is well developed and clearly presented. As noted by each reviewer, a key strength of the paper is its extensive experimental evidence that clearly supports the theory that is introduced. The reviewers initially raised a few relatively minor questions/concerns with the paper, which the authors thoroughly addressed in during the author-reviewer discussion phase, as the reviewers acknowledge in their follow-up comments.